# High-altitude balloon-launched uncrewed aircraft system measurements of atmospheric turbulence and qualitative comparison with infrasound microphone response

Anisa N. Haghighi[1], Ryan D. Nolin[1], Gary D. Pundsack[2], Nick Craine[2], Aliaksei Stratsilatau[3], and Sean C. C. Bailey[1]

[1]Department of Mechanical and Aerospace Engineering, University of Kentucky, Lexington, Kentucky 40506, USA
[2]Stratodynamics Inc., 16192 Coastal Highway, Lewes, Delaware 19958, USA
[3]UAVOS Inc., 541 Jefferson Ave, Ste 100, Redwood City, California 94063, USA

**Correspondence:** Sean Bailey (sean.bailey@uky.edu)

**Abstract.** This study investigates the use of a balloon-launched Uncrewed Aircraft System (UAS) for the measurement of turbulence in the troposphere and lower stratosphere. The UAS was a glider which could conduct an automated descent following a designated flight trajectory and was equipped with in-situ sensors for measuring thermodynamic and kinematic atmospheric properties. In addition, this aircraft was equipped with an infrasonic microphone to assess its suitability for the remote detection of clear-air turbulence. The capabilities of the UAS and sensing systems were tested during three flights conducted in 2021 in New Mexico, U.S.A.. It was found that the profiles of temperature, humidity and horizontal winds measured during descent were in broad agreement with those made by radiosonde data published by the U.S. National Weather Service, separated by up to 380 km spatially and by 3 to 5 hours temporally. Winds measured during controlled flight descent were consistent with the winds measured by global-positioning-system-derived velocity during balloon ascent. During controlled descent with this particular payload, a nominal vertical resolution on the order of 1 m during was achieved for temperature, relative humidity and pressure with a nominal vertical resolution of wind velocity vector on the order of 0.1 m, with the aircraft having a glide slope angle from $1°$ to $4°$. Analysis approaches were developed that provided turbulent kinetic energy and dissipation rate, but it was found that the corresponding Richardson number was sensitive to the methodology used to determine the vertical gradients from a single flight. The low-frequency content of the infrasonic microphone signal was observed to qualitatively align with long-wavelength wind velocity fluctuations detected at high altitude. Moreover, the microphone measured more broadband frequency content when the aircraft approached turbulence produced by the boundary layer.

## 1 Introduction

By enhancing the exchange of mass, momentum, and energy, the formation and evolution of atmospheric turbulence provides a significant contribution to weather and climate. As atmospheric turbulence is expected to increase in frequency and intensity

in response to climate change (e.g. Williams and Joshi, 2013), increased understanding of its role in atmospheric dynamics is needed. The presence of atmospheric turbulence, particularly clear air turbulence, also poses an aviation hazard that is challenging to predict and detect. This latter point is particularly true for high-altitude autonomous flight, a regime which is being increasingly pursued in the form of High-Altitude Pseudo-Satellite aircraft intended to provide communication and remote observation capabilities at relatively low cost (Gonzalo et al., 2018; D'Oliveira et al., 2016; Hasan et al., 2022). By the nature of the low density conditions under which these aircraft are designed to operate, they tend to be lightweight with narrow performance envelopes for which controlled flight can be maintained, making them susceptible to atmospheric disturbances. Hence, there is motivation to improve our ability to measure and predict stratospheric and upper-tropospheric turbulence, which can occur due to shear instabilities or gravity wave breaking despite the higher static stability at these altitudes. Turbulence in the upper troposphere may also be introduced through mesoscale convective systems (Liu et al., 2014), mountain waves (Cunningham and Keyser, 2015) and wind shear (e.g. as introduced by the jet stream).

Although the turbulence intensity is commonly quantified through the turbulent kinetic energy per unit mass

$$k = \frac{1}{2}\overline{u_i' u_i'}, \tag{1}$$

atmospheric turbulence is often also characterized through the related turbulent kinetic energy dissipation rate, formally defined as

$$\varepsilon = \frac{1}{2}\nu \overline{\left(\frac{\partial u_i'}{\partial x_j} + \frac{\partial u_j'}{\partial x_i}\right)\left(\frac{\partial u_i'}{\partial x_j} + \frac{\partial u_j'}{\partial x_i}\right)}. \tag{2}$$

Here, summation notation has been used with the index indicating the wind velocity components, $u_i$, and direction components $x_i$. Furthermore, $\nu$ indicates the kinematic viscosity, and $u_i' = u_i - \overline{u_i}$ indicates the unsteady contribution to the velocity component, with the overline denoting an average value.

The presence of turbulence is often related to the local gradient Richardson number

$$Ri = \frac{\frac{g}{\theta_v}\frac{\partial\overline{\theta_v}}{\partial x_3}}{\left(\frac{\partial\overline{u_1}}{\partial x_3}\right)^2 + \left(\frac{\partial\overline{u_2}}{\partial x_3}\right)^2} \tag{3}$$

where $g$ is the gravitational acceleration,

$$\theta_v = T\left(\frac{100000}{P}\right)^{0.2861}(1 + 0.61q) \tag{4}$$

is the virtual potential temperature, $u_1$ and $u_2$ are the horizontal components of the wind velocity vector, $x_3$ is the spatial component aligned in the direction opposing $g$, $q$ is the water vapor mixing ratio, $T$ is the temperature in Kelvin and $P$ is the

pressure in Pascals. The static stability is quantified through the square Brunt-Väisälä frequency,

$$N^2 = \frac{g}{\overline{\theta_v}} \frac{\partial \overline{\theta_v}}{\partial x_3} \tag{5}$$

and the square shear frequency provides a measure of potential for turbulence production due to mean velocity shear

$$S^2 = \left(\frac{\partial \overline{u_1}}{\partial x_3}\right)^2 + \left(\frac{\partial \overline{u_2}}{\partial x_3}\right)^2. \tag{6}$$

Hence, $Ri = N^2/S^2$, can also be used to attribute the source of turbulence as being from convective or shear instabilities (Söder et al., 2021; Sharman et al., 2014; Kim et al., 2020), and introduces the potential to model the relationship between turbulence in the stratosphere and tropospheric activity (Chunchuzov et al., 2021). A critical Richardson number of $Ri = 0.25$ (below which turbulence is likely) has been empirically identified, although a range of critical values $0.25 < Ri < 1$ have also been proposed (Abarbanel et al., 1984; Galperin et al., 2007).

When considering stratospheric turbulence measurements, most experiments have been conducted using balloon-borne instruments (e.g. Wescott et al., 1964; Ehrenberger, 1992; Haack et al., 2014; Alisse et al., 2000; Gavrilov et al., 2005) with some of the earliest of these experiments conducted in the 1960s (Enlich and Mancuso, 1968). Among the most relevant conclusions from these studies is that stratospheric turbulence tends to form in relatively thin atmospheric layers due to intrinsic static stability at these altitudes. In many cases, meteorological balloon soundings rely on indirect measures of turbulence, for example

through application of the analysis proposed by Thorpe and Deacon (1977) which analyzes temperature profiles to infer $\varepsilon$ and the Thorpe length scale (e.g. Clayson and Kantha, 2008).

One series of balloon-borne experiments investigating stratospheric turbulence using direct measures was the Leibniz-Institute Turbulence Observations in the Stratosphere (LITOS) experiments. The experiments, conducted by LITOS (Haack et al., 2014), were conducted using balloons equipped with a thermal anemometer to measure velocity and temperature fluctua-

tions at high frequency. The resulting measurements were within sub-centimeter resolution, and therefore suitable for resolving the finer scales of turbulence. This experiment reached altitudes up to 30 km, and when $\varepsilon$ was compared to both $Ri$ and $N^2$, an increase in $\varepsilon$ with altitude was observed with clear correlation between turbulent events and $Ri < 0.25$ (Haack et al., 2014). However, in some instances turbulent events were also observed when $Ri > 0.25$ although other studies attribute such behavior to the specifics of the $Ri$ calculation (Galperin et al., 2007; Haack et al., 2014) and later studies suggest that the measurements

may have been contaminated by the balloon wake (Söder et al., 2019).

Remote sensing techniques have also been deployed for turbulence studies, for example through the use of radar to measure the turbulent kinetic energy dissipation rate (Bertin et al., 1997; Fukao et al., 1994; Sato and Woodman, 1982; Barat and Bertin, 1984). Early studies found radar-determined dissipation rates were often underestimated due to the low vertical resolution. However, very-high-frequency and ultra-high-frequency radar measurements have presented a better prospect to resolve thin

layers of clear air turbulence (Barat and Bertin, 1984). For example, an altitude resolution of 150 m was able to detect the thin layers of turbulence in the lower stratosphere (Sato and Woodman, 1982).

Crewed aircraft measurements have also been utilized to measure high altitude turbulence (e.g. Nastrom and Gage, 1985). Currently, routine measurements of atmospheric turbulence are conducted for operational forecasting through Aircraft Meteorological Data Relay (AMDAR) reports generated by in-situ measurement systems on commercial aircraft using a turbulence detection algorithm developed by Sharman et al. (2014). These systems generally report the turbulence intensity in the form of eddy dissipation rate ($EDR$), defined as

$$EDR = \varepsilon^{1/3}, \tag{7}$$

which is currently used as a standard for turbulence reporting by the International Civil Aviation Organization. In the AMDAR $EDR$ calculation, a fully-formed von Kármán inertial subrange is assumed, and the $EDR$ is determined from either vertical-wind measurements, or the aircraft's gust response measured through acceleration.

Recently, it has become increasingly common to use uncrewed aircraft, or uncrewed aerial systems (UAS), equipped with in-situ sensors (e.g. hot-wire anemometers, sonic anemometers, hot-film probes, pitot tubes, or multi-hole pressure probes) for studies of turbulence in the atmospheric boundary layer and troposphere (e.g. Egger et al., 2002; Hobbs et al., 2002; Balsley et al., 2013; Witte et al., 2017; Rautenberg et al., 2018; Bärfuss et al., 2018; Jacob et al., 2018; Bailey et al., 2019; Al-Ghussain and Bailey, 2022; Lawrence and Balsley, 2013; Balsley et al., 2018; Reuder et al., 2012; Calmer et al., 2018; Luce et al., 2020; Kantha et al., 2017). Many of the UAS used for turbulence studies employ multi-hole probes, which measure the dynamic pressure of the air, with multiple pressure ports combined with a directional calibration used to determine the wind vector components relative to the probe axis. Due to their fragility, hot-wire probes, which measure the convective heat transfer across a very thin heated filament, are usually reserved for short-term scientific studies although they are standard instruments on some UAS (e.g. Hamilton et al., 2022). The fast response of the hot-wire anemometer allows detailed characterization of the turbulence, for example by allowing the measurement of small-scale fluctuations.

One measurement approach with potential for remotely detecting atmospheric turbulence is through infrasound detection, which involves capturing acoustic frequencies below 20 Hz (Shams et al., 2013). This typically requires specialized microphones designed for the infrasonic frequency range. Turbulent pressure fluctuations can generate sound waves that, for long-wavelength atmospheric turbulence, propagate as aerodynamic noise at low frequencies. One advantage of using infrasound for remote turbulence detection is its ability to propagate over long distances due to increased acoustic propagation at low frequencies and low kinematic viscosity (Whitaker and Norris, 2008), allowing infrasonic aerodynamic noise to travel through the atmosphere over distances ranging from a few hundred to a few thousand kilometers (Posmentier, 1974). Consequently, infrasonic microphones are being investigated for severe storm detection as well (e.g., Bowman and Bedard, 1971; Elbing et al., 2019; Wilson et al., 2023). Although techniques for quantitatively measuring turbulent properties using infrasound is not yet devised, an array of ground-based microphones was able to detect clear air turbulence at distances up to 360 km (Shams et al., 2013). Within the atmospheric boundary layer, infrasound energy measured in the frequency range spanning from 0.01 Hz to 15 Hz was found to correspond to coincidentally measured boundary layer turbulent kinetic energy, particularly in cases well

mixed turbulence such occurs with buoyantly-forced convective turbulence or in the presence of elevated jets (Cuxart et al., 2015).

Infrasound detection therefore holds potential for detecting clear air turbulence on airborne platforms, potentially providing a means to avoid hazardous conditions. As far as we know, these sensors have not yet been tested on UAS for this purpose. However, early studies using balloon-borne infrasonic sensors were able to detect a signal consistent with boundary layer turbulence (Wescott, 1964). Balloon-borne infrasonic microphones are also being investigated for various Earth-sensing applications. For instance, sensitive infrasonic microphones have shown promise in detecting acoustic low-frequency microbaroms (Bowman and Lees, 2015).

Here, we present measurements from a balloon-launched stratospheric glider UAS intended for turbulence measurement. Alongside in-situ sensors aimed at quantifying turbulence encountered by the aircraft, a novel infrasonic microphone was also installed to assess its capability in detecting airborne acoustic signatures of turbulence. This configuration was tested in a series of three high-altitude flight tests with the aircraft carried by balloon from 25 km up to 30 km above sea level. After reaching these altitudes, it was released from the balloon and conducted autonomous flight to a designated measurement location, thereafter following a pre-programmed helical flight path to a designated landing location. During its descent, the aircraft's glide slope was 1° to 4°, which allowed it to measure over a distance of up to 50 m horizontally for a 1 m change of altitude, while maintaining a relatively constant location over the surface. One drawback of this approach is that the current regulatory environment means deploying this type of UAS within a national airspace system requires permission from the corresponding regulatory body, requiring additional measures to ensure deconfliction with crewed aircraft. Here, the flights were conducted in the restricted airspace above Spaceport America, New Mexico U.S.A., coordinating with the adjacent White Sands Missile Range and utilizing airspace closed to crewed aircraft. However, this approach to deconfliction restricts the geographical location at which such measurements can be taken.

The remainder of this manuscript is divided into three main sections: Section 2 describes the aircraft and measurement systems, along with information about the flight location and flight path; Section 3 describes the approaches which were used to extract turbulence statistics from the data acquired during the flights; with Section 4 summarizing the main findings from this study.

## 2 Experiment Description

### 2.1 Aircraft

Key to this research was the use of the host UAS platform, the HiDRON H2 (see Fig. 1), operated by Stratodynamics Inc. The HiDRON H2 is a balloon-launched carbon-fiber/fiberglass glider UAS that is capable of autonomous and soaring flight modes. It has a wingspan of 3.8 m and its nominal flight weight is approximately 5.7 kg with the payload. To achieve initial altitudes in excess of 30 km, the HiDRON H2 is launched using latex sounding balloons. Different sizes of balloons were used in this study with 2000 g, 1200 g and 3000 g balloons employed for each of three flights, referred to as Flights 1, 2 and 3, respectively. After release, the aircraft is controlled by a UAVOS Inc. autopilot to follow a pre-programmed descent pattern

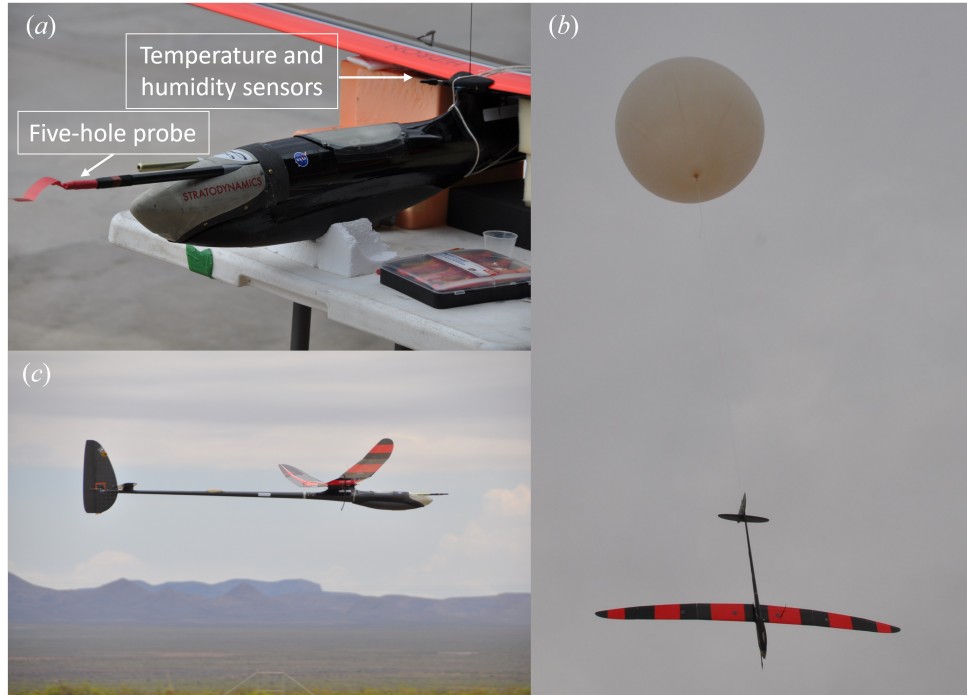

**Figure 1.** Images of HiDRON H2 showing: (*a*) close up of aircraft nose showing five-hole probe and temperature and humidity sensor location; (*b*) aircraft during launch; and (*c*) aircraft during landing.

towards a designated landing point. An operator can track the HiDRON H2 position from launch to landing and changes to the flight plan can be made in real time through radio telemetry, which also allows operational parameters to be transmitted to the ground at distances up to 100 km. Full telemetry information was produced at 10 Hz by the autopilot for all three flights,

including location, ground speed, 6 degree-of-freedom orientation information as well as pressure, temperature and humidity information from an integrated sensing system. This 10 Hz sample rate telemetry was intrinsic to the radio modem used for the command-and-control link and telemetry. The specific rate was selected for maximum efficiency in the transmission and recording of data packets and to maintain reliability of the radio communication and corresponding command-and-control link.

Other safety features include a parachute, dual-redundant balloon release system and geofencing safety protocols that prevent

the aircraft from leaving the designated airspace. During prior flights, including flights exceeding altitudes of 30 km, the HiDRON H2 has shown reliability in remaining controllable in high-wind (as high as 32 m s$^{-1}$), operating in low-temperature conditions ($< -60°$C), and in returning to a predefined landing site. For the flights reported here, the winds were less than 25 m s$^{-1}$ and the minimum temperature was -65°C.

For all flights, the autopilot was set to maintain the aircraft's kinetic energy (reflected through the indicated airspeed), with

the value selected near the optimal lift-to-drag ratio (the maximum distance that can be travelled per loss in altitude). To maintain the set airspeed the autopilot adjusts the pitch angle on the elevator to control the angle of attack of the main wing airfoil. The velocity of the aircraft relative to the air, $V_R$, may therefore fluctuate slightly due to the pitch angle adjustments.

Also, as the aircraft descends in altitude the air density increases and the HiDRON H2's aerodynamic performance improves; thus, $V_R$ gradually decreased as the aircraft descended. Figure 2 shows the magnitude of horizontal ground speed, $|V_G|$, descent speed, $V_D$, relative to the ground, as well as the magnitude of the aircraft's velocity relative to the air, $|V_R|$, for the helical descent portion of each of the three flights as a function of the altitude above sea level (a.s.l.). The horizontal velocities are related through $V_R = (V_{G_X} - u)e_1 + (V_{G_Y} - v)e_2 - (V_D e + w)e_3$ where $V_G = V_{G_X} e_1 + V_{G_Y} e_2$ is the horizontal velocity of the aircraft relative to the ground in the Earth-fixed, inertial frame of reference with $e_1$, $e_2$, $e_3$ the basis vectors aligned to the East, North, and up, respectively. The vector $U = u e_1 + v e_2 + w e_3$ is the wind vector and $-V_D e_3$ the aircraft's descent speed in the inertial frame. The method used to obtain the wind velocity vector is provided in Section 2.2.

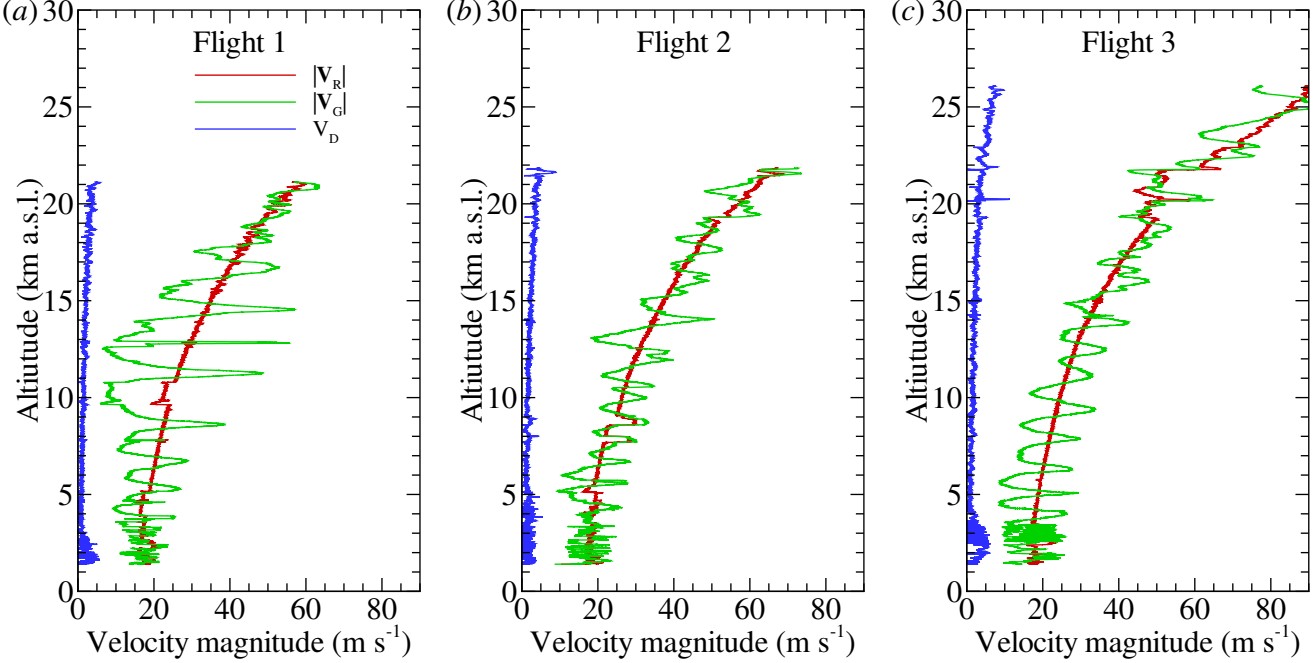

**Figure 2.** Horizontal ground speed in inertial Earth-fixed coordinates, $|V_G|$, magnitude of the descent speed in inertial Earth-fixed coordinates, $V_D$, and relative air velocity magnitude in aircraft-fixed coordinates, $|V_R|$, for (*a*) Flight 1, (*b*) Flight 2 and (*c*) Flight 3.

As noted above, to measure atmospheric conditions, the aircraft was equipped with an integrated InterMet Systems iMet-XF atmospheric sensing system having fast-response bead thermistor to measure air temperature, $T$, and a capacitive relative humidity, $RH$, sensor. The manufacturer-provided specifications (International Met Systems) state the pressure sensor provided a $\pm1.5$ hPa accuracy for pressure, $P$, with humidity sensor supporting a full 0 - 100 %$RH$ range at $\pm 5$ %$RH$ accuracy with a resolution of 0.7 %$RH$. The temperature sensor provided a $\pm 0.3°$C accuracy with a resolution of 0.01$°$C up to a maximum of 50$°$C. The stated response times of these sensors are on the order of 10 ms for pressure, 5 s for humidity and 2 s for temperature in still air, with the autopilot sampling these sensors at 10 Hz. The pressure and temperature sensors were mounted with the

sensing elements exposed to the airflow upstream of the wing support (see Fig. 1a) to ensure aspiration of the sensors during forward flight.

## 2.2 Payload

The turbulence-measuring payload was a combination of four components: (1) a five-hole probe; (2) an infrasonic microphone; (3) a data acquisition board; and (4) an embedded computer. These components were installed in the nose of the HiDRON H2, which could be accessed via removal of the nose cone, as shown in Fig. 3c.

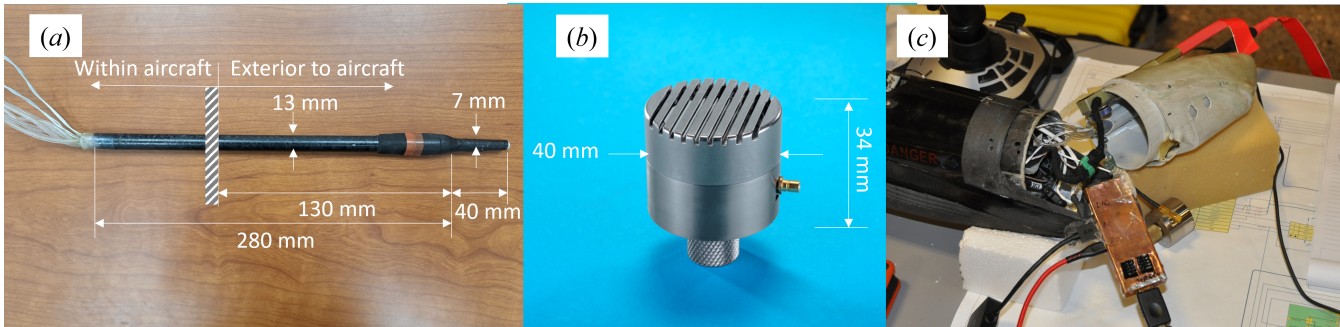

**Figure 3.** ($a$) Five-hole probe, ($b$) infrasonic microphone, and ($c$) nose payload bay open between flights, with embedded computer shown removed for data retrieval. Infrasonic microphone is below embedded computer but was installed in aircraft nose facing forward during flight.

### 2.2.1 Five-Hole Probe:

Wind speed and direction relative to the aircraft were measured using a bespoke five-hole probe mounted such that the probe projected upstream of the nose of the aircraft, as shown in Fig. 1. The probe, detailed in Fig. 3a, is similar to that used in Bailey et al. (2019, 2020); Al-Ghussain and Bailey (2022) and was manufactured from a carbon-fiber tube equipped with a beveled aluminum tip. The tip of the probe was arranged with one center hole normal to the probe axis surrounded by four other holes arranged symmetrically around it having their plane normal vector aligned $20°$ to the probe axis. For this measurement, the pressure difference between the central hole (measuring total stagnation pressure) and a series of additional holes arranged on the carbon fiber tube (measuring static pressure) were used to determine the approximate dynamic pressure at the probe tip. The pressure measured by this hole combination differs from the true dynamic pressure, $Q$, depending on the alignment of the wind vector with the probe axis. The two horizontally-opposed circumferential holes are arranged to produce a pressure difference which changes with the sideslip angle, $\beta$, of the air flow relative to the probe axis. Similarly, the two vertically-opposed circumferential holes were arranged to produce a pressure difference which changes with the angle of attack, $\alpha$, of the air flow relative to the probe axis. Thus, $Q$, $\alpha$ and $\beta$ can be determined by measuring the pressure differences across the different holes arranged on the probe.

Prior to installation on the HiDRON H2, the probe was calibrated in a wind tunnel to determine its directional response using an apparatus designed to pitch and yaw the probe at $\alpha$ and $\beta$ angles up to $25°$ relative to the mean wind vector. Based off

of established procedures, e.g. as outlined by Bohn and Simon (1975); Treaster and Yocum (1978); van den Kroonenberg et al. (2008) and Wildmann et al. (2014), and utilizing an implementation approach very similar to that used by Witte et al. (2017), the pressure differences at each $\alpha$ and $\beta$ angle combination experienced during directional response calibration were used to build pressure coefficients

$$C_\beta(\alpha, \beta) = \frac{\Delta P_{32}}{\Delta P_1 + 0.5(\Delta P_{32} + \Delta P_{54})} \tag{8a}$$

$$C_\alpha(\alpha, \beta) = \frac{\Delta P_{54}}{\Delta P_1 + 0.5(\Delta P_{32} + \Delta P_{54})} \tag{8b}$$

$$C_q(\alpha, \beta) = \frac{\Delta P_1 - Q}{\Delta P_1 + 0.5(\Delta P_{32} + \Delta P_{54})} \tag{8c}$$

where $\Delta P_1$ is the pressure difference between the central hole and the static pressure, $\Delta P_{32}$ is the pressure difference across the horizontal probe holes, $\Delta P_{54}$ is the pressure difference across the vertical probe holes and $Q$ is the dynamic pressure. The probe design resulted in unique combinations of $C_\beta$, $C_\alpha$ and $C_q$ for each $\alpha$ and $\beta$ angle of the probe to the relative air velocity vector. These relationships are stored in tables following the wind tunnel calibration.

To calculate the relative wind velocity vector measured during flight, $C_\alpha$ and $C_\beta$ were calculated for every sample point from the measured $\Delta P_1$, $\Delta P_{32}$ and $\Delta P_{54}$. Using the calibration tables, the values of $\alpha$ and $\beta$ which produces that combination of $C_\alpha$ and $C_\beta$ could be identified. A similar process was used to identify the corresponding value of $C_q$ measured at that $C_\alpha$ and $C_\beta$ combination, which allowed $Q$ to be determined. The result is knowledge of the magnitude and direction of the dynamic pressure vector relative to the probe axis for every sampled combination of $\Delta P_1$, $\Delta P_{32}$ and $\Delta P_{54}$. The dynamic pressure was then converted to velocity using the density determined from iMet-XF measurements of the ambient pressure, temperature and humidity.

The probe used on these flights was also heated to prevent ice formation within the probe during flight. This was accomplished by wrapping the probe body in nickel-chromium resistance wire. A feedback circuit, using a thermistor attached to the probe tip, passed current through the wire at a rate sufficient to maintain the probe tip temperature at $50°$C. By repeating the wind tunnel directional response calibrations with and without heating active, it was determined that there was no influence of probe heating on $\Delta P_1$, $\Delta P_{32}$ and $\Delta P_{54}$, or their dependence on $\alpha$ and $\beta$. These calibration checks were also conducted both at 20 m s$^{-1}$, which replicated the expected flight dynamic pressure of $Q \approx 200$ Pa, and at 5 m s$^{-1}$ to check for any Reynolds-number dependence of $C_\beta$, $C_\alpha$ and $C_q$. No evidence of Reynolds number was dependence found.

To measure $\Delta P_1$, $\Delta P_{32}$ and $\Delta P_{54}$, the probe was connected to differential pressure transducers through 50 cm of 1.75 mm diameter flexible polymer tubing, specifically Tygon polyvinyl chloride tubing. To ensure that the low-density conditions at flight altitude did not result in pressure differences below the sensitivity of an individual transducer, the measured pressure

difference was converted to analog voltage using two different sets of transducers by teeing the tubing to each transducer set, i.e. a low and high sensitivity transducer was used for each pressure difference $\Delta P_1$, $\Delta P_{32}$ and $\Delta P_{54}$. The low-sensitivity transducer set was comprised of TE Connected Measurements 4515-DS5A002DP differential pressure transducers with a 500 Pa range. The second transducer set was comprised of Allsensors DS-0368 differential pressure transducers with a 65 Pa range. Both sets of analog output voltages were linearly scaled relative to the maximum transducer range with a nominal span

of 4.5V and 4.0V respectively. During flight, the autopilot maintained flight speeds sufficient to produce pressure differences well within the range of the low-sensitivity transducers (i.e. the dynamic pressure was maintained between 100 Pa and 200 Pa) which exceeded the range of the high sensitivity transducer connected to $\Delta P_1$. Hence, only the readings from the low-sensitivity sensors were used for data analysis. However, the high sensitivity $\Delta P_{32}$ and $\Delta P_{54}$ transducers provided a means to estimate the uncertainty of the pressure measurement by comparison with their low-sensitivity counterparts, as will be described later.

To convert the air velocity vector components relative to the probe axis into a frame of reference relative to the ground, an additional coordinate transformation was conducted using the aircraft's pitch, yaw, and roll angles as measured by the autopilot. Details of this process are provided in Witte et al. (2017) and are based off of procedures described in Lenschow (1972) for measurements using similar probes mounted on crewed aircraft. This process assumes that the coordinate system of the probe during wind tunnel calibration is perfectly aligned with the body-frame coordinate system of the autopilot, which is difficult

to achieve in practice. Therefore, an additional procedure was used to correct for any misalignment of the probe axis and aircraft's body-frame axis. This approach, described in Al-Ghussain and Bailey (2021), also corrects for airframe influence on the measured $Q$, and time misalignment between the probe's pressure sensors and the aircraft's kinematic sensors.

As the autopilot and payload data acquisition systems were acquired asynchronously, alignment of autopilot kinematic data and five-hole-probe pressure data in time was conducted during post-processing by cross-correlating the dynamic pressure

measured by the aircraft's intrinsic pitot probe (recorded by the autopilot along with the aircraft position and orientation information) and $\Delta P_1$ of the five-hole probe recorded by the payload data acquisition system. The time lag between the two systems was then removed before performing the transformation of the velocity vector in the aircraft body frame to the Earth-fixed frame of reference. To do so, the aircraft position and orientation information was up-sampled from the autopilot's 10 Hz sample rate to the 1 kHz sample rate used by the on-board data acquisition system, with the up-sampling conducted using

simple linear interpolation.

The result of these procedures is a time-dependent wind velocity vector described using components $u(t)$, $v(t)$, and $w(t)$ which are aligned to the East, to the North, and up, respectively. The time-dependent horizontal wind velocity magnitude and direction could then be found from

$$U(t) = \left(u(t)^2 + v(t)^2\right)^{-0.5} \tag{9}$$

and

$$\gamma(t) = \text{atan2}\left(-u(t), -v(t)\right) \tag{10}$$

where atan2 indicates a numerical implementation of the $\tan^{-1}$ function used to disambiguate the polar direction using the quadrant formed by the sign of the velocity components.

Note also that, although data was acquired at a 1 kHz sample rate to ensure that the infrasonic microphone response was fully resolved, the actual probe maximum frequency response a was much lower due to viscous attenuation of the pressure fluctuations within the tubing, coupled with inaccuracies introduced by high-frequency resonance within the transducer cavity (Grimshaw and Taylor, 2016). Here, the term maximum frequency response is used to indicate the frequency at which there is a -3 dB attenuation in the measured signal relative to the corresponding input being measured. At ambient conditions, the maximum frequency response of the five-hole-probe was estimated to be on the order of 20 Hz using the same measurement approach utilized in Grimshaw and Taylor (2016) and Witte et al. (2017), which measures the response of the system following a step change in pressure. Similar response tests conducted in an environment chamber at -70°C and 8000 Pa indicated that, consistent with the model presented in Grimshaw and Taylor (2016), both the resonance in the transducer cavity and viscous attenuation can be expected to increase with altitude, resulting in a slightly lower maximum frequency response of 10 Hz.

Figure 4a shows an example of the intrinsic electrical noise observed in the pressure transducer signals. Although this particular excerpt was measured from the low-sensitivity $\Delta P_1$ transducer during the environment test at -70°C and 8000 Pa, similar signals were measured by the other pressure transducers at ambient conditions, during environment testing, and during Flights 1, 2 and 3. The electrical noise was found to be composed of a $\sim \pm 2$ Pa stochastic component combined with a -8 Pa to 0 Pa quasi-periodic component. As similar characteristics were observed for both low- and high-sensitivity transducers, scaling with the range of the transducers, it is hypothesized that this noise is introduced through the common five volt reference signal used to power all pressure transducers. As shown in Fig. 4b, which compares the one-sided autospectral density, $F_{PP}(f)$, calculated from five minutes of the low-sensitivity $\Delta P_1$ transducer time series measured during the environment chamber test at ambient conditions to an equivalent measurement made at -70°C and 8000 Pa, the frequency content of this noise signal was independent of ambient pressure and temperature conditions. Furthermore, Fig. 4c shows the equivalent autospectral density calculated from a quiescent period of the Flight 1 low-sensitivity $\Delta P_1$ time series in red, specifically from a five minute portion of the time series while the aircraft was on the ground prior to launch. This autospectral density shows similar energy content to that measured in the environment chamber was also present during the flight tests. Also shown in Fig. 4c is the frequency content measured during every five minute segment of the $\Delta P_1$ time series during the controlled descent portion of Flight 1. Although the low frequency content of the signal produced $F_{PP}(f)$ values up to four orders of magnitude higher than the intrinsic noise, at frequencies higher than the maximum frequency response of the five-hole-probe (specifically when $f > 100$ Hz), the frequency content of the signal remained independent of ambient conditions.

To minimize any possible impact of this electrical noise on the statistical analysis of the five-hole-probe measurements, during post-processing a background noise subtraction procedure was conducted. This procedure filtered the voltage signals by subtracting the background noise energy content in the frequency domain prior to scaling the voltage to pressure. For this purpose, a 5-minute segment of each pressure transducer signal measured before balloon launch was chosen, specifically when the probe's cover was installed (as depicted in Fig. 1a). This portion of the time series was assumed to be representative of the background electrical noise. The amplitude of its Fourier transform was subtracted from the corresponding Fourier-transformed

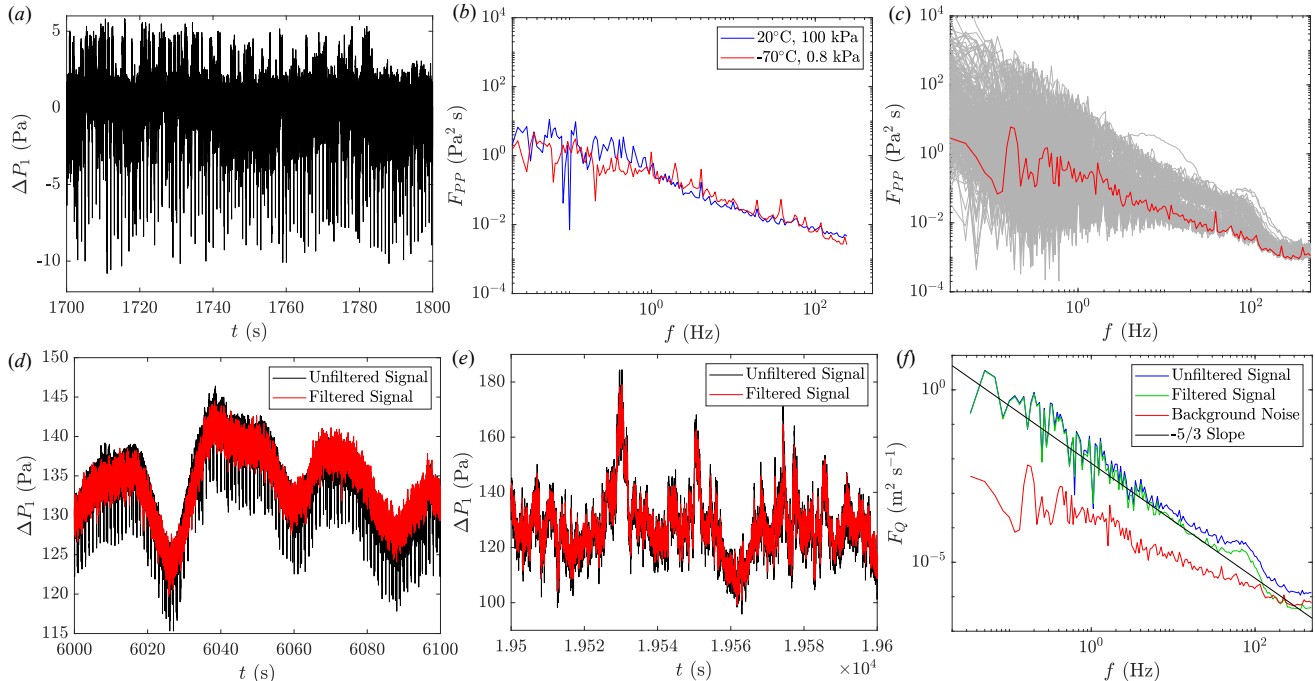

**Figure 4.** (*a*) Time-series showing intrinsic electrical noise in pressure transducer signal measured during environment test at -70°C and 8000 Pa, and (*b*) one-sided autospectral density from same test comparing frequency content in $\Delta P_1$ transducer from five minute periods at ambient conditions and -70°C and 8000 Pa. (*c*) One-sided autospectral density from $\Delta P_1$ transducer measured during quiescent five minute period before launch, shown in red, compared to one-sided autospectral density calculated from same transducer's time signal for every five minute period during controlled descent of HiDRON H2. Comparison of time series from the $\Delta P_1$ transducer measured before (unfiltered signal) and after (filtered signal) noise removal process (*d*) during Flight 1 while the HiDRON H2 was in the stratosphere and (*e*) while the HiDRON H2 was in the atmospheric boundary layer. (*f*) Comparison of one-sided autospectral density of $(\Delta P_1\ \rho^{-1})^{-1/2}$ measured while aircraft is in atmospheric boundary layer with and without noise removal applied along with corresponding background noise for comparison. Also shown in (*f*) is a solid line showing -5/3 slope.

voltage signal in five-minute intervals, while preserving the phase information of the original segment. The filtered signal was then returned to the time domain by conducting an inverse Fourier transform.

A 100 s long segment of the original, unfiltered, low-sensitivity $\Delta P_1$ time series is shown in Fig. 4d from a portion of Flight
1 when the HiDRON H2 was within the stratosphere, when the frequency content of the signal was small. Also shown in this figure is the same segment of the time series following the frequency domain filtering described above. The comparison indicates that the filter was successful in removing the quasi-periodic contribution of the noise but, due to its stochastic nature, the low-amplitude ±2 Pa contribution to the electrical noise remains. A similar comparison is provided in Fig. 4e from a portion of the $\Delta P_1$ time series measured when the HiDRON H2 was within the turbulent boundary layer during Flight 1. This figure
illustrates that when in conditions where there is significant frequency content in the signal, this content is preserved during the filtering process. This last point is perhaps better illustrated in Fig. 4f, which shows the one-sided autospectral density, $F_Q(f)$, of the frequency content of $(\Delta P_1\ \rho^{-1})^{1/2}$ (which approximately corresponds to the magnitude of the air velocity relative to

the probe), measured within the boundary layer during Flight 1 with and without the filter applied. Figure 4f illustrates how the low frequency ($f < 1$ Hz) content of the measured signal is largely unaffected by the filtering process, with the filter increasing in impact as the energy within the signal decreases at higher frequencies. Also illustrated in Fig. 4f is how application of the filter increases the agreement of the frequency dependent roll-off of $F_Q(f)$ with the $f^{-5/3}$ dependence expected for inertial subrange turbulence. The broadband peak in $F_Q(f)$ evident at $f \approx 100$ Hz is attributed to resonance within the transducer cavity, as it is not present in the background noise $F_Q(f)$ spectrum.

It should be noted that by ignoring the phase information of the background noise during the removal process, artifacts could be introduced into the time domain of the filtered signal, particularly when the amplitudes of the signal and subtracted background noise are similar (e.g., at high frequencies and when the air is quiescent). By comparing the statistics presented in Section 3 with and without the filtering applied, we find that the primary benefit of implementing this background noise subtraction process is that it reduces some of the scatter in the data, allowing trends to become more evident. However, caution has been exercised when interpreting results where distortion of the signal is likely.

Finally, the $\Delta P_{54}$ pressure line was inadvertently disconnected during maintenance prior to Flights 2 and 3 and not discovered until after Flight 3. Therefore, for these flights, conversion of five-hole probe voltages was conducted with $\alpha$ provided by the autopilot's telemetry data. This $\alpha$ was estimated from the true airspeed determined from the aircraft's pitot probe, which measures the relative air speed along the aircraft's longitudinal axis, and the speed of the aircraft through the air along the aircraft's vertical axis. This latter speed was determined using the aircraft's descent rate, which itself was estimated through a Kalman filter fusion of the static pressure rate of change, vertical acceleration, and global positioning system velocity. By assuming that the relative vertical component of velocity of the air to the aircraft was only due to the descent rate of the aircraft through the air, the speed in the aircraft's vertical axis could then be determined by combining the aircraft's descent rate and true airspeed information with the aircraft orientation measured by the autopilot gyroscopes. The latter information allowed transformation from inertial to body-fixed coordinates. The definition of $C_\alpha$ and $C_\beta$ used for directional calibration and recovery of $Q$ and $\beta$ were also modified to not include $\Delta P_{54}$. While not ideal, we were able to verify that this approach was justified by processing the Flight 1 data with and without inclusion of $\Delta P_{54}$. The average difference between the resulting $u(t)$, $v(t)$ and $w(t)$ values was found to be less than 0.25 m s$^{-1}$ with the average difference in the Reynolds normal stresses found to be less than 0.09 m$^2$ s$^{-2}$. Importantly, the altitude dependence of these quantities was preserved. Figures comparing the differences resulting from the processing methods are presented in Appendix A.

### 2.2.2 Infrasonic Microphone:

Infrasonic measurements of acoustic frequencies were conducted using an infrasonic microphone. For these tests a microphone and acoustic measurement system developed at NASA Langley Research Center was used. The microphone, shown in Fig. 1b, when coupled with a PCB Pezoetronics 485M49 amplifier, was capable of infrasound (acoustic frequencies lower than 20 Hz) detection with a sensitivity of 115 mV Pa$^{-1}$ (amplified to $\pm$10 V) having a power consumption of 35 mW and weight of 186 g. The geometry of the 38.1 mm diameter microphone diaphragm was designed with a high compliance (low diaphragm tension)

such that membrane motion was substantially critically damped and optimally dimensioned for the 0.01 Hz to 20 Hz frequency range, with the full bandwidth of the sensor being 0.01 Hz to 1 kHz.

The microphone was mounted rigidly within the nose of the aircraft with the diaphragm plane normal to the forward flight direction. By being located within the fuselage, the microphone was protected by dynamic pressure fluctuations. As the wavelengths of infrasonic sound exceed 10 m, sound attenuation by the fuselage was not expected for frequencies lower than 20 Hz.

### 2.2.3 Data Acquisition:

A Measurement Computing Systems MCC USB-1608FS-Plus data acquisition system (DAQ) was used to digitize the voltage output from the pressure transducers and microphone signal conditioning unit. The DAQ also provided the 5 V signal used to power the pressure transducers. This particular DAQ was capable of recording eight single-ended analog inputs simultaneously at 16-bit resolution at rates of up to 400 kHz. During these experiments the DAQ sampled at 1 kHz the seven channels containing the analog voltage signals from all six pressure transducers and the infrasonic microphone. Digitized voltage values were then transferred through universal serial bus to the embedded computer for logging.

### 2.2.4 Embedded Computer:

The DAQ was connected via universal serial bus (USB) to a mini stick computer with an Intel Atom Z8350 processor, 128 GB eMMC non-volatile memory, and 4 GB RAM. To minimize radio frequency interference and shield the computer from high altitude radiation, the computer was encased in a copper shield (Fig. 3c). A custom script was used to control data acquisition and storage. The computer stored all recorded data on its eMMC memory which was then downloaded post-flight via USB connection for archiving and further analysis. To allow payload operational verification during flight, an RS232 connection was established between the computer and the autopilot. Through this channel, sensor voltage variance and preliminary turbulence detection parameters were passed to the autopilot to be included in the telemetry stream. This information was also later available for preliminary temporal alignment of sensor and autopilot data.

### 2.3 Uncertainty Estimation of Wind Measurement

As described above, measurement of the wind vector was a multi-step process involving information from different sensors, calibrations and corrections. Therefore, to assess the uncertainty of the wind vector measurement, a Monte-Carlo method was used to estimate the error propagation from the sensors to the final wind estimate. To do so, the post-processing calculations were repeated for 100 iterations with each iteration having the sensor values perturbed from their measured value by an amount determined using normally-distributed random number generators. The standard deviation of the normal distributions were selected to correspond to each sensor's uncertainties.

Specifically, for a measured sensor value, here represented as $\Phi(t)$, a perturbed value $\Phi_i(t)$, was found for each iteration, $i$, following

$$\Phi_i(t) = \Phi(t) + \frac{E_B}{2}\mathcal{N} + \frac{E_P}{2}\mathcal{N}(t) \tag{11}$$

**Table 1.** Bias and precision errors for each sensor value used in wind estimate

| Measured value | $E_B$ | $E_P$ |
|---|---|---|
| $\Delta P_1, \Delta P_{32}, \Delta P_{54}$ | 5 Pa | 2 Pa |
| $P$ | 150 Pa | 1 Pa |
| $T$ | 0.3 °C | 0.01 °C |
| $RH$ | 5% | 0.7% |
| yaw | 0.2° | 0.01° |
| pitch | 0.2° | 0.01° |
| roll | 0.2° | 0.01° |
| $V_{G_X}, V_{G_Y}$ | 0.1 m s$^{-1}$ | 0.02 m s$^{-1}$ |

where $\mathcal{N}$ represents a normally distributed random number, drawn from a standard normal distribution having mean of zero and standard deviation of unity. The quantities $E_B$ or $E_P$ represent the estimated bias (accuracy) and precision (resolution) errors for the sensor being perturbed, with the bias error perturbation, $\mathcal{N}$, added to $\Phi(t)$ only once per iteration, and the precision perturbation, $\mathcal{N}(t)$, added for every sample in time. Here $E_B$ and $E_P$ are taken to represent the 95% uncertainty range, twice the standard deviation of their distribution.

The analysis produced an ensemble of 100 time-dependent (and hence altitude-dependent) measurements of wind for each flight, randomly distributed via the propagation of the perturbed sensor readings, each perturbed following Equation 11. The uncertainty estimate was then found for the magnitude and direction from twice the ensemble standard deviation at each measurement time, $t$, (corresponding to a 95% probability) of the 100 iterations. Note that this approach is similar to that employed by Van den Kroonenberg et al. (2008), but by employing the Monte-Carlo analysis this approach allows the uncertainty to vary in time (and hence altitude) as well as incorporating any additional uncertainty which may occur due to coupling of sensor errors.

The values for $E_B$ and $E_P$ used in the uncertainty propagation analysis are provided in Table 1 and required different methods for their determination. For the pressure differences measured by the transducers connected to the five-hole-probe ($\Delta P_1$, $\Delta P_{32}$, $\Delta P_{54}$), $E_B$ was estimated from the average difference between the low-sensitivity and high sensitivity $\Delta P_{32}$ and $\Delta P_{54}$ transducer values recorded during each flight, whereas $E_P$ was estimated using twice the standard deviation of the difference. For $P$, $T$ and $RH$, the manufacturer-provided accuracy and resolution values were used for $E_B$ and $E_P$, respectively. Finally, the uncertainty of the attitude and heading reference system (AHRS) used by the autopilot was not available, so the values for accuracy and resolution values from a different manufacturer's equivalent AHRS system were used as proxy values.

Note that this analysis does not take into account additional uncertainty introduced by errors in the directional calibration, misalignment in time between the sensors, and any altitude-dependent variability in the values provided in Table 1. To accommodate these undetected uncertainties in the analysis, the $E_B$ and $E_P$ values provided in Table 1 were doubled prior to implementation in Equation 11.

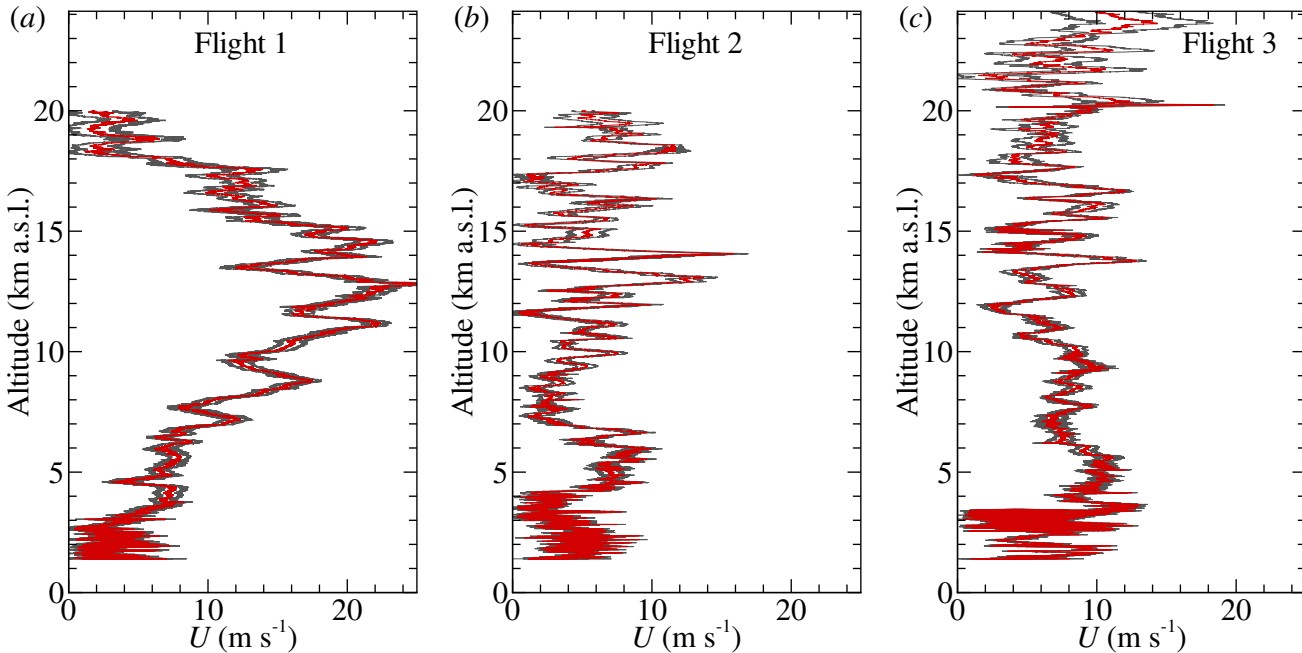

**Figure 5.** Result of uncertainty analysis for (*a*) Flight 1; (*b*) Flight 2; and (*c*) Flight 3. Red lines show velocity magnitude calculated from the mean of 100 ensembles, with corresponding 95% uncertainty bounds shown as black lines.

The mean value of $U$ across the 100 iterations at each time step is shown in Fig. 5a-c for Flights 1, 2 and 3, along with the corresponding 95% (two standard deviation) uncertainty bounds. The resulting uncertainty in the wind estimate was found to be altitude dependent, with the highest uncertainty of approximately 3.5 m s$^{-1}$ observed at the highest altitudes, decreasing to approximately 0.5 m s$^{-1}$ at the lowest altitudes measured. This altitude dependence directly arises from the influence of $P$,

$T$ and $RH$ uncertainty on the air density estimate. The magnitude of the uncertainty presented in Fig. 5 is consistent with the results of an intercomparison study presented in Barbieri et al. (2019). In this intercomparison, wind estimates from probes and procedures very similar to that used here were found to be within $\pm$ 1 m s$^{-1}$ of a ground-based reference velocity sensor.

Furthermore, although not directly apparent in Fig. 5, the uncertainty in wind magnitude was found to be most dependent on the yaw angle, which is consistent with the observations of Van den Kroonenberg et al. (2008). This is due to the dependence of

wind magnitude on aircraft yaw, pitch, and roll introduced during the coordinate transformation between body-fixed and inertial coordinate systems. During flight, the yaw angle can vary from $0°$ to $360°$ degrees, whereas the $\alpha$, $\beta$, pitch and roll angles are near zero. The result is that during the coordinate transformation the $u$ and $v$ wind components are primarily determined from the true airspeed multiplied by the sine of yaw angle and cosine of pitch angle for $u$ and cosine of yaw angle and sine of pitch angle for $v$ (see, for example, Van den Kroonenberg et al., 2008). The result is that during an orbit, the horizontal velocity

components will have highest sensitivity to error the aircraft's yaw angle both at yaw angles of $0°$ degrees (for $u$) and $90°$ (for $v$) which, for a helical flight path such as flown in this study, can introduce periodic artifacts into the wind estimates. The

vertical component of wind velocity, $w$, is most sensitive to error in pitch angle due to this component of wind being primarily determined by the true airspeed multiplied by the sine of the pitch angle (which is near $0°$ degrees for most of the flight).

## 2.4 Experiment Overview

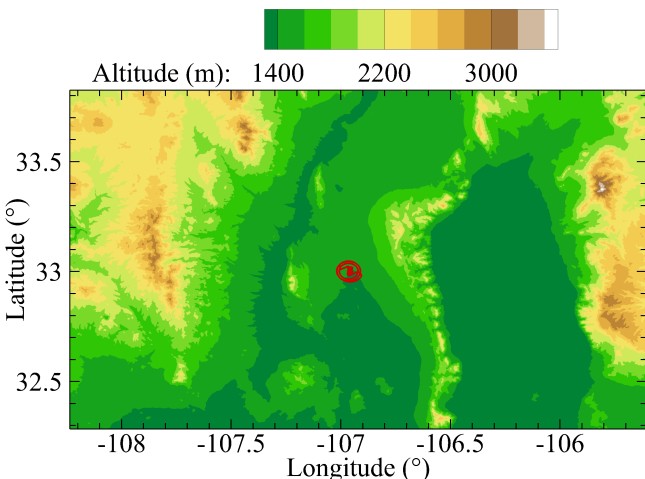

**Figure 6.** The topography of the flight area with Spaceport America with the trajectory of Flight 3 indicated by a red line to illustrate region of measurement.

The aircraft and payload combination was deployed in a measurement campaign conducted in the restricted airspace above Spaceport America, located near Truth or Consequences, New Mexico, U.S.A. between the Black Range and San Andres mountain ranges (Fig. 6), from June 1, 2021 through June 6, 2021. The height above sea level of launch and recovery of the aircraft was 1406 m above sea level (a.s.l.). Three flights were conducted by the HiDRON H2 on different dates, with Flight 1 being conducted on June 1; Flight 2 conducted on June 4; and Flight 3 being conducted on June 6. Each flight consisted
of a weather-balloon carrying the glider aloft at an ascent rate of approximately 7 m s$^{-1}$ to a release altitude of 25 km a.s.l. for Flights 1 and 2, and 30 km a.s.l. for Flight 3 (corresponding to $z = 23.6$ km and $z = 28.6$ km above ground level, a.g.l., respectively). After release from the balloon, the aircraft conducted an automated descent along a pre-determined flight path towards the Spaceport America runway. Autopilot-controlled approach, landing and recovery occurred on the Spaceport America main runway, at which point the aircraft, payload and all logged data were recovered.

## 2.5 Flight Profiles:

The flight trajectories for all three flights are presented in Fig. 7 showing balloon launch, ascent to 25 km or 30 km altitude a.s.l., release of the HiDRON H2 aircraft, controlled return to the airspace above the launch and control point, and helical descent to the landing point. The helical descent was conducted at an initial radius of approximately 5 km, selected as a compromise between optimizing aerodynamic efficiency of the aircraft in a turn (by minimizing the bank angle) and to stay
in close proximity to the landing runway. As the aircraft descended, this radius was reduced to approximately 4 km and,

eventually, 1 km to keep the aircraft within gliding distance of the designated landing point. During the descent phase of the flight, as shown in Fig. 2, the rate of descent decreased from 5 m s$^{-1}$ to 1 m s$^{-1}$ (producing a nominal descent rate of 2 m s$^{-1}$). Overall flight time was approximately 6 hours with 4 hours of that being the descent phase. Note that, during descent, several tests were conducted of the HiDRON H2's flight control system (including a test of autonomous soaring during Flight 3 which resulted in the aircraft gaining altitude for approximately 10 minutes). The result of these tests were occasional deviations of the flight path from the nominal helical trajectory.

All three flights started in the morning hours, with Flight 1 launched at 13:47 UTC (07:47 LT) on June 1 2021, released from the balloon at 14:35 UTC (08:35 LT), and landing at 18:42 UTC (12:42 LT). Flight 2 launched at 14:04 UTC (08:04 LT) on June 4 2021, released at 15:15 UTC (11:15 LT) and landed at 18:39 UTC (12:39 LT). Finally, Flight 3 launched at 14:07 UTC (08:07 LT) on June 6 2021, released at 15:17 UTC (09:17 LT), and landed at 19:43 UTC (12:43 LT). Local time, LT, at Spaceport America was Mountain Daylight Time (UTC -6:00).

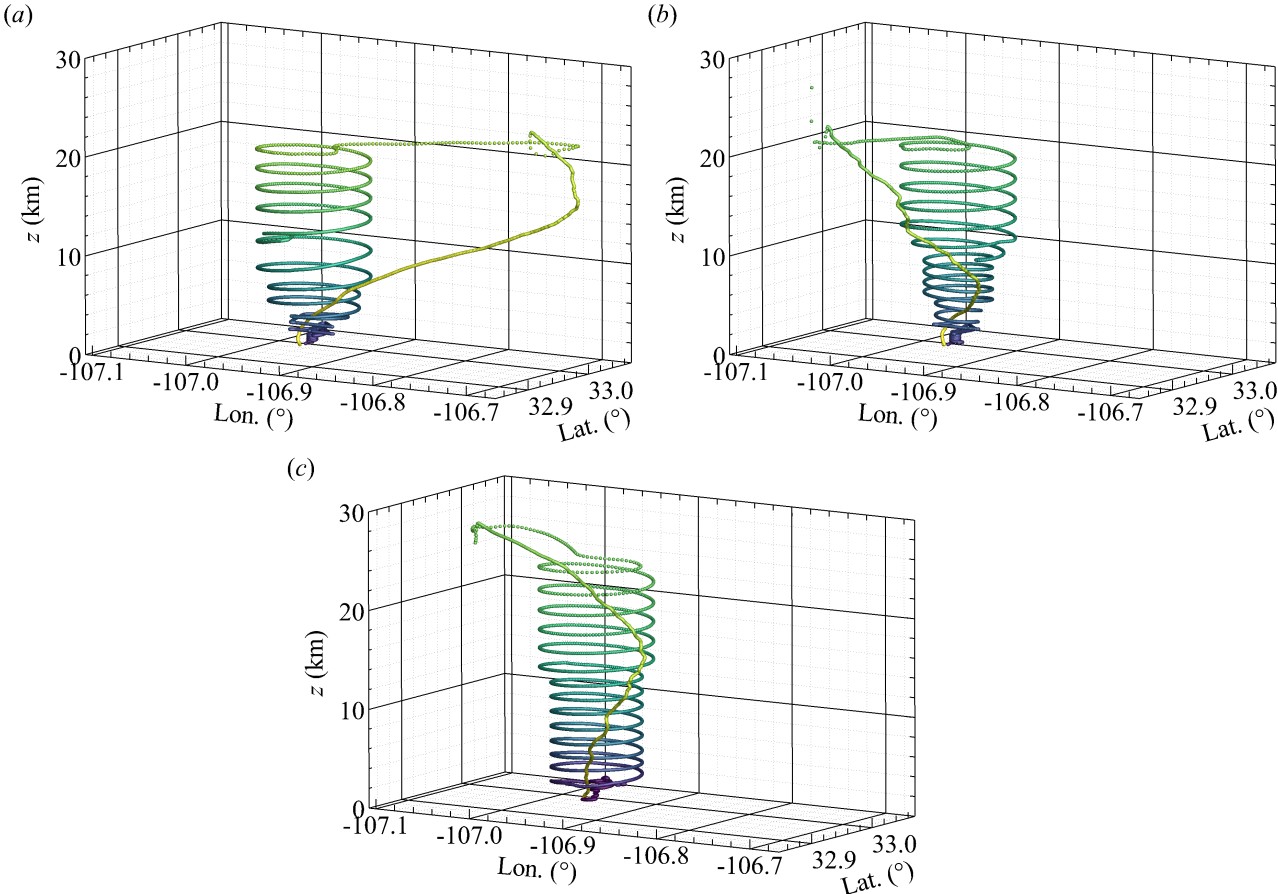

**Figure 7.** HiDRON H2 flight trajectory for (*a*) Flight 1, (*b*) Flight 2 and (*c*) Flight 3. Trajectory is colored by time, with lighter color indicating earliest phase of balloon ascent. $z$ indicates height above ground level (a.g.l.).

## 3 Results

In this section, we present the measured values of temperature, relative humidity, wind vector, and infrasonic microphone signal amplitude, as well as presenting calculation approaches and results for selected statistics. The temperature and humidity sensors were mounted on the front side of the wing support pillar (as shown in Figure 1a) and during ascent they were in a stagnant region within the wing pillar wake and not sufficiently aspirated to prevent self-heating and delayed air exchange with the environment. Hence, only data from the controlled descent phase of the flight are presented for temperature and humidity in this section. Here, $z$ is used to indicate the vertical distance referenced to ground level, i.e. above ground level (a.g.l.) with $z = 0$ corresponding to the launch and recovery altitude of 1406 m a.s.l.. In addition, when presenting results measured during descent we limit the data to $z < 20$ km for Flights 1 and 2 and $z < 25$ km for Flight 3, i.e. the portion of the flight when the aircraft was following a the helical trajectory (see Fig. 7) and within $\pm 5$ km of a fixed geographic location. This latter constraint is introduced since even the low-sensitivity $\Delta P_1$ exceeded the transducer's range shortly after the aircraft was released and started its flight towards the helical orbit location. By the time the aircraft reached the orbit location, $\Delta P_1$ had returned to a range measurable by the low-sensitivity transducer, although it never reduced to a value measurable by the high-sensitivity $\Delta P_1$ transducer.

The time series of $U(t)$ measured during descent is presented in Fig. 8a. For this figure, the time series has been low-pass-filtered at 1 Hz to better visualize some of the features of the time series. By inspection, it was observed that while the aircraft was in the stratosphere ($t < 7000$ s), the velocity fluctuations were largely composed of low frequency motion having periods on the order of 400 s, as illustrated in the segment of the time series shown in Fig. 8b. For the nominal $|V_R|$ of approximately 60 m s$^{-1}$ in this altitude range (see Fig. 2) this corresponds to wavelengths on the order of 24 km. Interspersed with these low frequency motions were irregular occurrences of high frequency fluctuations more characteristic of turbulence, such as shown in Fig. 8b around $t = 6000$ s. These latter events occurred in bursts lasting on the order of 100 s (or 6 km). As the aircraft descended through the troposphere, similar behavior was observed, although the duration of high-frequency events were typically longer, on the order of 300 s (as illustrated in Fig. 8c around $t = 12000$ s and $t = 12500$ s). However, $|V_R| \approx 20$ m s$^{-1}$ at these altitudes so the corresponding lengths of these events remained near 6 km. Towards the end of the flight, the intensity and frequency of the high frequency occurrences increased (Fig. 8d), consistent with the UAS approaching and entering the boundary layer ($t > 16500$ s). Similar behavior was observed for Flights 2 and 3.

Following these observations, to capture the statistical properties of the high frequency portions of the time-series, the descent portion of the flight was divided into 3 km statistical segments, determined using $|V_R| \Delta t$, where $\Delta t$ is the amount of time included in each segment. Due to the spiralling flight path (as illustrated in Fig. 7) each 3 km segment represents approximately 150 m of vertical descent. Quantities averaged over these segments use $\langle \, \rangle$ brackets. In order to decrease the vertical spacing between statistical values, each segment is overlapped with its neighbor by 50%, thereby decreasing the spacing of statistical quantities to 75 m vertically. Finally, to minimize the effects of low frequency bias on the statistics calculated for each segment, when statistics require the subtraction of a local mean value (e.g. as required to determine $u'(t)$, $v'(t)$ or $w'(t)$),

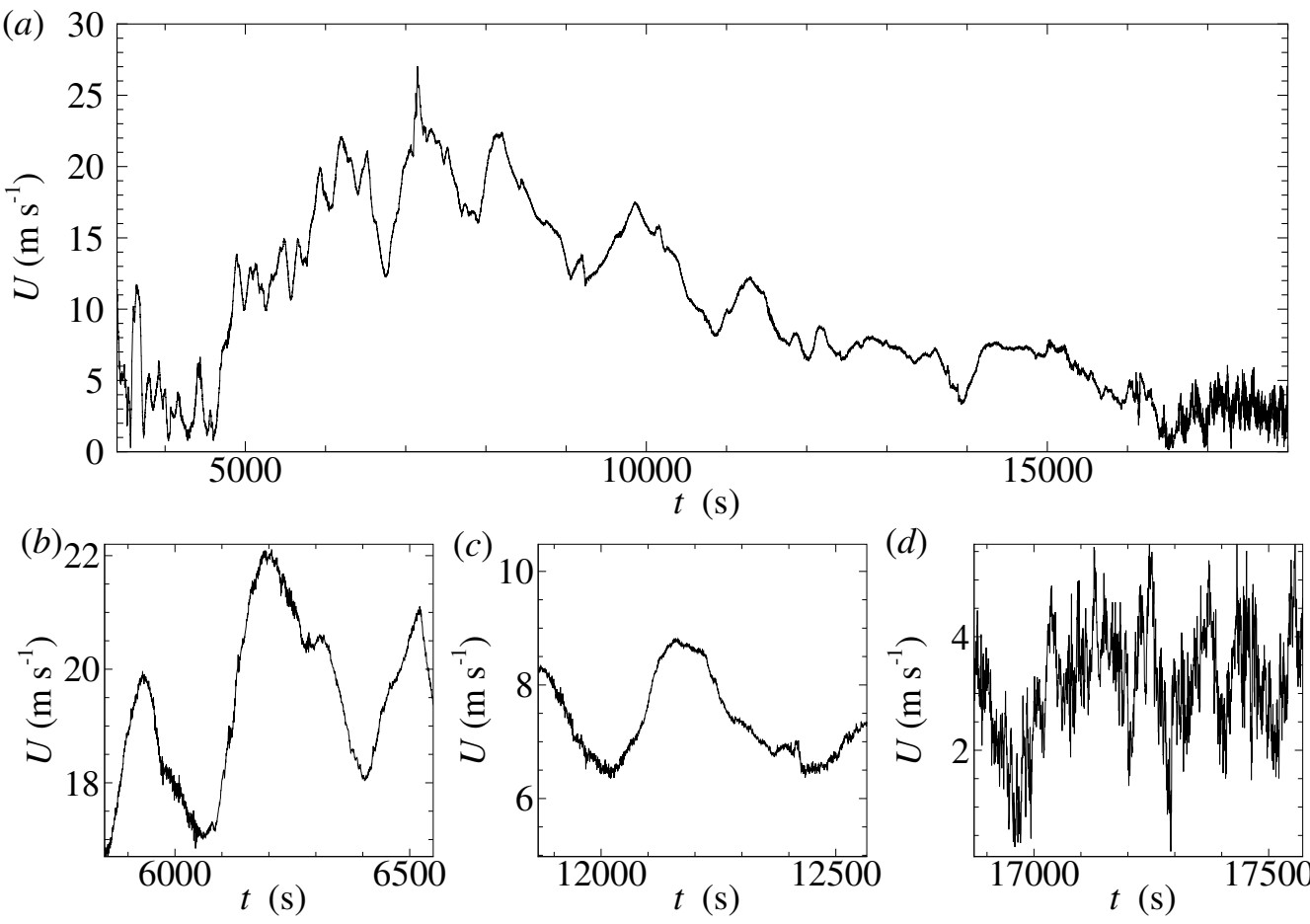

**Figure 8.** Time series of velocity magnitude low-pass-filtered at 1 Hz for (*a*) controlled descent portion of Flight 1 with $t = 0$ corresponding to startup of data acquisition system prior to balloon release. Subfigures (*b*), (*c*) and (*d*) show selected portions of time series in stratosphere, troposphere and boundary layer, respectively.

rather than subtracting the segment's mean value, we instead subtract the linear trend within that segment (i.e. by detrending the segment).

### 3.1 Mean quantities

To establish the ambient conditions during each flight we averaged $P$, $T$, $RH$, $U$ and $\gamma$ over each statistical segment and compare vertical profiles of these quantities to publicly-available 12:00 UTC National Weather Service (NWS) radiosonde
weather soundings launched from the El Paso (EPZ), Albuquerque (ABQ) and Tuscon (TUS) NWS forecast offices. These forecast offices and sounding times were selected due to their proximity to the launch site and flight times, with the launch site within the triangle formed by these three locations. To assist with comparison across sites in $z$, we have set $z = 0$ at 1406 m a.s.l. for all soundings. Furthermore, to assist with comparison, the National Oceanic and Atmospheric Administration

(NOAA) upper air maps at 250 mbar and National Aeronautics and Space Administration (NASA) satellite imagery for the approximate time of Flights 1 through 3 are provided in Appendix B with the NWS sounding sites and Spaceport America (SPA) indicated on them. The NWS soundings employed Graw DFM-17 radiosondes with manufacturer-provided accuracies of $\pm 0.2°$C in temperature, $\pm 4\%$ in $RH$, $\pm 0.1$ m s$^{-1}$ in wind speed and $\pm 1°$ in wind direction.

Vertical profiles of $\langle T \rangle$ and $\langle RH \rangle$, measured by the HiDRON H2, are compared to the radiosonde profiles for all three flights in Fig. 9. The HiDRON H2 temperature is consistent with the trends produced by the radiosonde values, but shows a noticeable warm bias compared to the radiosondes for $z > 16$ km that appears most evident in the results from Flight 3. Note that no density or other correction was applied to this sensor measurement to account for the reduced convective heat transfer at these altitudes. The lapse-rate in the troposphere also appears to deviate from the radiosonde lapse rates for $z < 5$ km, although it is not clear if this is due to a sensor-related issue or due to spatial variability in the atmospheric conditions.

Figures 9d-f compare the corresponding $\langle RH \rangle$ measurements from the HiDRON H2 and NWS radiosondes. Significant differences are clearly evident among the profiles. However, noting that the radiosonde data were obtained from disparate locations up to 380 km away from the flight location, differences can be attributed to spatial heterogeneity in the atmospheric moisture concentration. This is qualitatively illustrated by comparison of the cloud coverage in satellite observations (Appendix B). However, on all three days, the HiDRON H2 reported consistently lower $RH$ values for $z > 7$ km, so a dry bias in the humidity sensor under cold conditions cannot be discounted.

The magnitude and direction of the horizontal winds for all three flights are shown in Fig. 10. In addition to comparison to the radiosonde soundings from ABQ, EPZ and TUS, we also compare the profiles measured during descent to those estimated from the ascent phase of each flight using $(V_{G_X}, V_{G_Y})$ as measured by the autopilot's global positioning system (GPS). To mitigate any possible influence of pendulum motion of the aircraft on the balloon tether (estimated to have a natural period of approximately 5 s), the velocity values were filtered by applying a 25 s moving average. The resulting wind estimates are treated as approximately equivalent to those produced by radiosondes, although the significantly increased weight of the aircraft relative to a radiosonde can be expected to produce a corresponding increase in the time constant compared to that of a radiosonde.

In general, the wind magnitude and direction measured by the HiDRON H2 are within the bounds provided by the radiosonde soundings, with the wind direction measured during descent producing good agreement with that reported by the radiosondes and by the GPS on ascent. To assess the agreement in wind magnitude, we computed the average of the three NWS radiosonde profiles at each altitude recorded by the HiDRON H2. Subsequently, we determined the difference in $U$ at each altitude. The resulting median of the absolute values of these differences was 1.9 m s$^{-1}$, 3.4 m s$^{-1}$, and 2.3 m s$^{-1}$ for Flights 1, 2 and 3, respectively. Similarly, comparing $U$ measured during ascent and descent by HiDRON H2 yielded median absolute difference values of 2.4 m s$^{-1}$, 3.0 m s$^{-1}$, and 2.7 m s$^{-1}$ for Flights 1, 2 and 3, respectively.

With respect to the wind magnitude profile itself, the HiDRON measurements do contain short wavelength (e.g. <1 km) fluctuations that were evident in both ascent (GPS ground velocity) and descent (five-hole-probe) wind velocity measurements made using the HiDRON H2. One significant difference between ascent and descent measurements are stronger long-wavelength fluctuations (on the order of 2.5 km) measured during descent than were measured during ascent for Flight 1 in

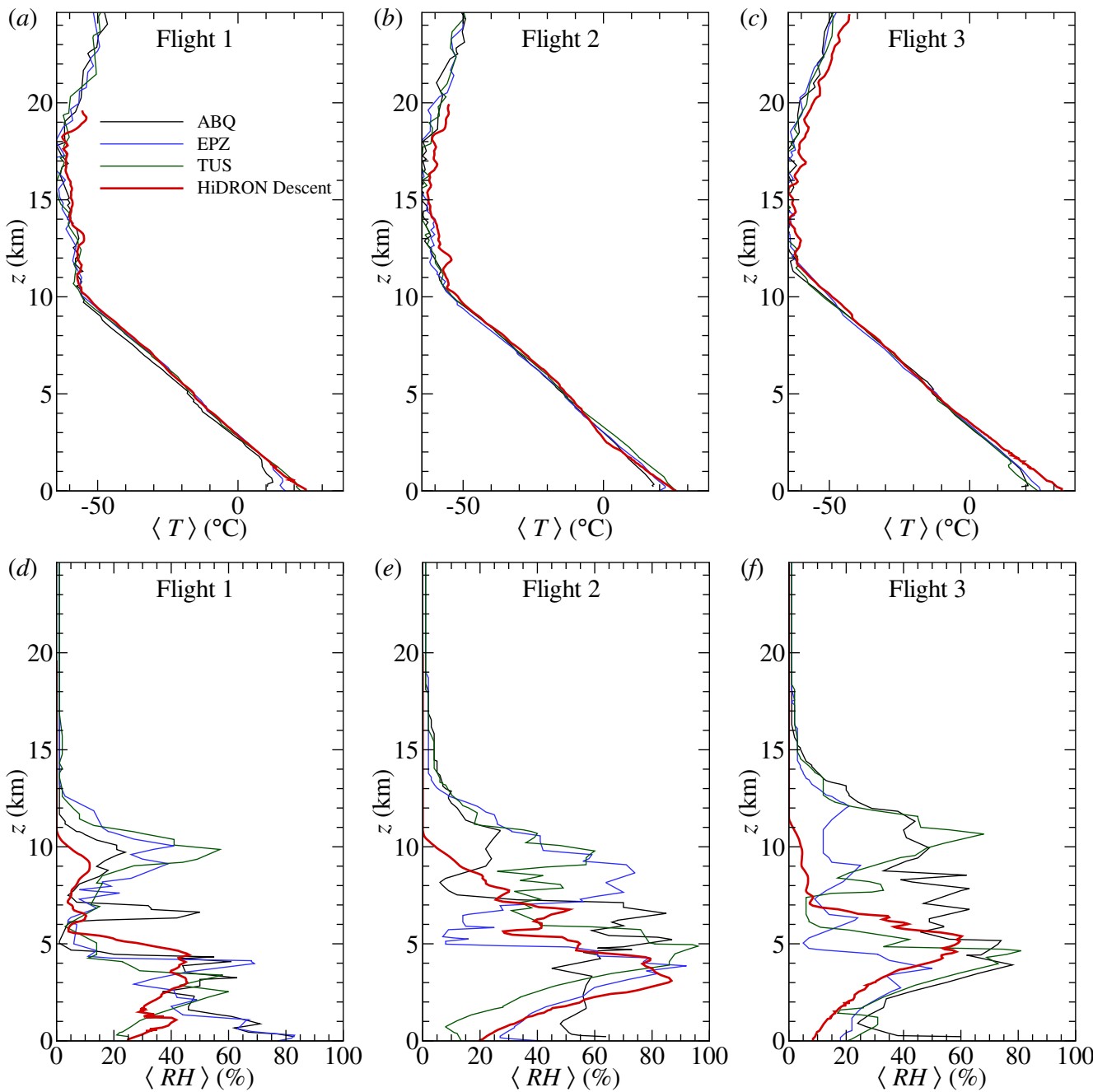

**Figure 9.** Temperature profiles measured during (*a*) Flight 1; (*b*) Flight 2; and (*c*) Flight 3 compared to NWS radiosonde soundings from the Albuquerque (ABQ), El Paso (EPZ), and Tuscon (TUS) forecast offices. Corresponding relative humidity profiles shown for (*d*) Flight 1; (*e*) Flight 2; and (*f*) Flight 3.

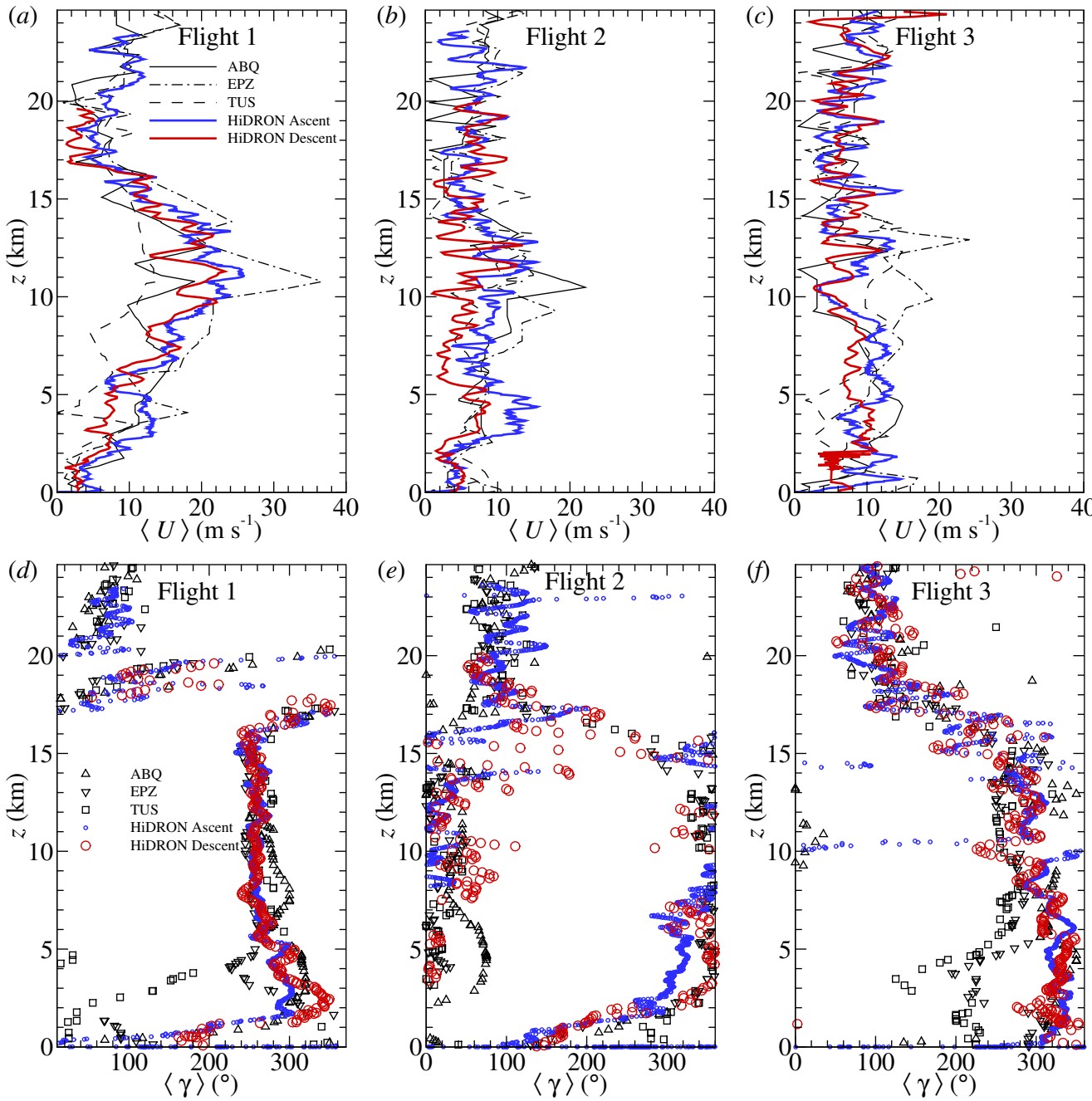

**Figure 10.** Horizontal wind magnitude a measured during (*a*) Flight 1; (*b*) Flight 2; and (*c*) Flight 3 compared to NWS radiosonde soundings from the Albuquerque (ABQ), El Paso (EPZ), and Tuscon (TUS) forecast offices. Corresponding wind direction shown for (*d*) Flight 1; (*e*) Flight 2; and (*f*) Flight 3.

the range $7.5 < z < 12.5$ km, and for Flight 2 in the range $11 < z < 13$ km. These low frequency waves are close to that of the orbit pitch at these altitudes (varying from 2 km to 3 km) and therefore may reflect bias in the wind estimate introduced by the orbital path. However, the magnitude of the differences exceeds the anticipated uncertainty bounds presented in Fig. 5. Furthermore, as discussed in Section 2.3, such periodic bias tends to be produced by error in yaw angle, which propagates through the transformation from body-fixed to inertial coordinate system. However, this yaw error influences $u$ and $v$ with phase and therefore also tends to appear in the wind direction as well (as shown in Al-Ghussain and Bailey, 2021, ,for example). As shown in Fig. 10d, for Flight 1 in the range $7.5 < z < 12.5$ km the $\langle \gamma \rangle$ estimates have close correspondence to the $\langle \gamma \rangle$ measured during ascent. It is also possible that the long-wavelength fluctuations in $\langle U \rangle$ may be evidence of horizontal heterogeneity in the wind magnitude at these altitudes, although this cannot be determined from the current measurements. Some periodicity in $\langle \gamma \rangle$ is evident during Flights 2 and 3 that is not measured during ascent, for example between 5 km and 10 km in the Flight 2 $\langle \gamma \rangle$ profiles. However, no corresponding periodicity is evident in $\langle U \rangle$ at these altitudes, and the wavelength of the periodicity is shorter than the pitch of the helical flight path at these altitudes, and therefore these oscillations are not believed to be due to bias introduced by the flight path. More detailed quantification of long-wavelength motions is provided through wavelet analysis presented in Section 3.4.

Finally, differences in wind magnitude on the order of 5 m s$^{-1}$ were observed between measurements made during ascent and descent in the troposphere during Flights 2 and 3. It is not clear what the source of this difference is, although there was a time difference of 2 to 4 hours between when the aircraft ascended through this region on its balloon and when it descended through the same region in a glide.

To summarize the observations of mean quantities provided in Figs. 9-10, atmospheric conditions were similar for Flights 2 and 3, which differed from Flight 1. The strongest winds occurred during Flight 1, which had an environmental lapse rate of 8.4 °C km$^{-1}$ (Fig. 9a) and winds coming from 270°, increasing with altitude to a peak value over 20 m s$^{-1}$ just above the tropopause ($z = 12.5$ km), after which the magnitude decreased with increasing $z$ to the stratospheric inversion near $z = 17$ km. This pattern of constant wind direction and high wind magnitude is consistent with the presence of a jet stream, and the NOAA upper air wind meteorological maps (e.g. as provided in Appendix B) indicate that during Flight 1 a tropical jet stream was centered to the southeast of the flight location, over central Texas, such that the flight path was on the outer edge of the jet. The relative position of the jet stream for Flight 1 also explains the higher wind magnitudes measured at EPZ, which was closer to the center of the jet. Above the jet stream for Flight 1, the winds increase with altitude again, with significant directionality shift indicating horizontal shear was present above the temperature inversion.

The jet stream had moved to the east by the time of Flights 2 and 3 (as shown in Appendix B), which is reflected in the reduced wind magnitude measured during these flights (Fig. 10b,e and Fig. 10c,f respectively), typically below $\langle U \rangle = 10$ m s$^{-1}$. Wind direction was consistently from the north for $z < 15$ km for Flight 2, with directional shear observed between $z = 15$ km and $z = 20$ km. An environmental lapse rate of 7.9 °C km$^{-1}$ was measured for Flight 2 with a corresponding value of 8.2 °C km$^{-1}$ measured for Flight 3. Wind observations during Flight 3 can be summarized as being nearly constant values of $\langle U \rangle \approx 10$ m s$^{-1}$ up to $z = 30$ km, with winds coming from 300° in the troposphere changing with altitude to be from 100° at $z = 20$ km.

## 3.2 Turbulent Quantities

The HiDRON H2 measurements can be used to quantify the intensity of turbulence at different altitudes. For example through the turbulent kinetic energy, $\langle k \rangle$ which was calculated for each statistical segment using

$$\langle k \rangle = \frac{1}{2} \left( \langle u'^2 \rangle + \langle v'^2 \rangle + \langle w'^2 \rangle \right). \tag{12}$$

Note that since $u(t)$, $v(t)$ and $w(t)$ were oversampled, an additional filtering step was taken to minimize the influence of high-frequency noise on $\langle u'^2 \rangle$, $\langle v'^2 \rangle$, and $\langle w'^2 \rangle$. To do this, these quantities were calculated by integrating the corresponding velocity spectrum, $F_{uu}(f)$, $F_{vv}(f)$, and $F_{ww}(f)$ (equivalent to the one-sided autospectral density function of the detrended velocity component, following the terminology of Bendat and Piersol, 2000). The velocity spectra were found for each statistical segment using Welch's periodogram method implemented with a variance-preserving Hanning window, three subintervals, and a 50% overlap, implemented using the Matlab function 'pwelch'. The final low-pass filtered estimates of $\langle u'^2 \rangle$, $\langle v'^2 \rangle$ and $\langle w'^2 \rangle$ were then determined by integrating $F_{uu}(f)$, $F_{vv}(f)$ and $F_{ww}(f)$ over the frequency range where they were above the noise floor. The upper limit of this range was determined by identifying the frequency at which noise began to dominate the integration over velocity fluctuations, leading to an increase in $F_{uu}$, $F_{vv}$, or $F_{ww}$ with rising $f$. The procedure employed for this process was to visualize the spectra as in compensated form, whereby $fF_{uu}$ is plotted on semi-logarithmic axes as a function of $f$. Such compensated, or 'pre-multiplied,' spectra facilitate the visualization of energy spectra when viewed in this manner because they allow for a clearer representation of the frequency dependence of the relative contribution of each frequency to the overall energy content. This is because, for example,

$$d\langle u^2 \rangle = F_{uu}df = fF_{uu}d(\log f). \tag{13}$$

Hence when $fFd(\log f)$ begins to increase on a semi-logarithmic plot at high frequencies, this indicates a frequency range where the energy content increases with $f$. Given that universal equilibrium range turbulence will decrease in energy content with $f$, the minimum in $fFd(\log f)$ indicates a frequency at which the noise begins to have a greater contribution to the variance than the turbulence content.

The upper bound identified using this approach was consistent with the probe's maximum frequency response in the boundary layer and varied between 1 Hz and 20 Hz above the boundary layer, with the higher upper frequency bounds corresponding to instances where there was increased low frequency content in $F_{uu}$, $F_{vv}$ or $F_{ww}$. It's worth noting that the Reynolds stresses filtered in this manner were observed to be an average of 85% of the variance of the corresponding unfiltered signal, reflecting the influence of high frequency noise on statistics.

Due to the time averaging used, the value of $\langle k \rangle$ will only incorporate contributions from relatively short wavelengths, corresponding to the distance travelled by the aircraft during the averaging time. Thus, an additional metric that can be used to quantify turbulence is the turbulent kinetic energy dissipation rate, $\varepsilon$. Within equilibrium homogeneous turbulence, $\varepsilon$ can be expected to scale with the rate of production of $k$, and within the inertial subrange it should scale with $k$ and the wavenumber

range of the inertial subrange. The dissipation rate is particularly useful for turbulent quantification in atmospheric turbulence for which the scale of the energy containing eddies may be quite large or ill-defined since $\varepsilon$ can be determined from small-scale fluctuations, without requiring resolution of the low wavenumber turbulence.

As determination of $\varepsilon$ using Equation 2 requires measurement of spatial gradients of velocity over distances on the order of the Kolmogorov scale, it is challenging to directly measure $\varepsilon$ in the atmosphere without additional assumptions. Thus, indirect estimates of $\varepsilon$ are usually employed. Here, we assume the presence of sufficiently high Reynolds number for the formation of an inertial subrange in the energy spectrum. Under such conditions, the one-dimensional longitudinal wavenumber velocity spectrum in the inertial subrange is expected to follow a scaling such that

$$E_{\ell\ell}(\kappa_\ell) = 0.49\varepsilon^{2/3}\kappa_\ell^{-5/3} \tag{14}$$

as suggested by Kolmogorov (1941) using the one-dimensional Kolmogorov constant of 0.49 empirically determined by Saddoughi and Veeravalli (1994). $\kappa_\ell$ is the longitudinal wavenumber and the wavenumber velocity spectrum calculated using the velocity component parallel to $\kappa_\ell$.

To determine $\langle\varepsilon\rangle$ within each statistical segment, $E_{\ell\ell}(\kappa_\ell)$ was estimated using the component of the wind velocity aligned with $-\boldsymbol{V_R}$. This was calculated by first rotating the $(u,v,w)$ coordinate system from the east-north-up alignment to instead align $u$ with an axis parallel to the velocity of the aircraft within the air, i.e. we define $u_\ell(t)$ as the component of the wind velocity vector found by projection of the wind velocity vector, $\boldsymbol{U}$, in the direction formed by $\langle-\boldsymbol{V}_R\rangle$. The velocity spectrum of $u_\ell(t)$ in the frequency domain, $F_{\ell\ell}(f)$, was then calculated on the rotated wind velocity vector following the same procedure used to calculate the Reynolds stresses. Noting that since Equation 14 is defined in the wavenumber domain, $F_{\ell\ell}(f)$ was then transformed to $E_{\ell\ell}(\kappa_\ell)$. To do this, the longitudinal wavenumber, $\kappa_\ell$, was approximated using Taylor's frozen-flow hypothesis such that $\kappa_\ell \approx f2\pi|\langle\boldsymbol{V}_R\rangle|^{-1}$. We then found the longitudinal velocity spectrum in the wavenumber domain as $E_{\ell\ell} = F_{\ell\ell}|\langle\boldsymbol{V}_R\rangle|(2\pi)^{-1}$, where this last operation was conducted to ensure that

$$\int_0^\infty E_{\ell\ell}(\kappa_\ell)d\kappa_\ell = \langle u_\ell^2\rangle. \tag{15}$$

Finally, Equation 14 was used to estimate $\langle\varepsilon\rangle$ by least-squares fit to $E_{\ell\ell}$. This fit was conducted over the $\kappa_\ell$ range corresponding to the frequency range used in the calculation of $\langle k\rangle$. The result is an estimate of $\varepsilon$ for each statistical segment, i.e. $\langle\varepsilon\rangle$. Note that the approach used here provides only an approximation of $\langle\varepsilon\rangle$ as $E_{\ell\ell}(\kappa_\ell)$ will only follow Equation 14 in the presence of inertial turbulence, whereas the fit will always provide a non-zero value of $\langle\varepsilon\rangle$ even if no turbulence is present. Therefore, some caution is required when interpreting these values.

The values of $\langle k\rangle$ are compared to the corresponding values of $\langle\varepsilon\rangle$ in Fig. 11. Although there is significant scatter, the general trend follows the $\langle k\rangle \propto \langle\varepsilon\rangle^{2/3}$ expected for this calculation approach, demonstrating some intrinsic consistency in the different calculations. Additionally, it's important to note that alongside the implicit assumptions inherent in calculating $\langle\varepsilon\rangle$,

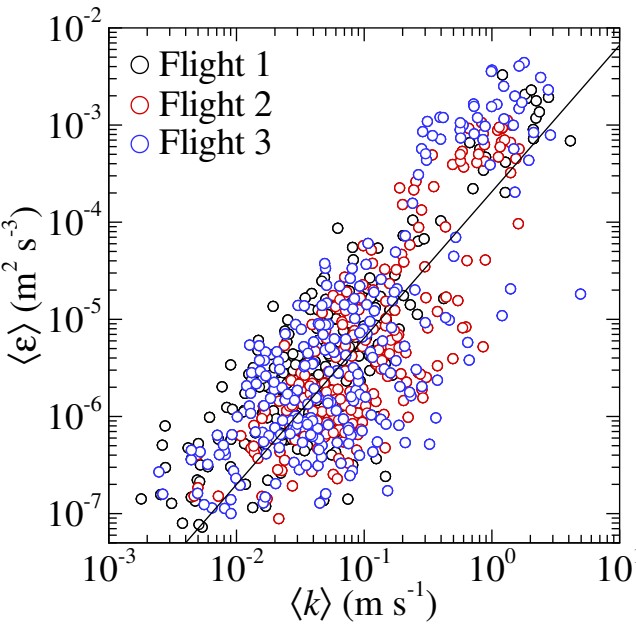

**Figure 11.** Comparison of turbulent kinetic energy and turbulent kinetic energy dissipation rate calculated within each statistical segment. Solid line indicates $k^{3/2}$ slope.

the method employed to compute $\langle k \rangle$ only incorporates the energy content corresponding to wavelengths smaller than the
statistical segment length (or frequencies higher than the inverse of the time taken to traverse that segment length).

The corresponding statistic of eddy dissipation rate, $EDR = \varepsilon^{1/3}$, is often used in the aviation industry to quantify turbulence. Following Huang et al. (2019), this metric allows the turbulence to be characterized as: steady for $EDR < 0.1$; weak for $EDR$ between 0.1 and 0.3; moderate for $EDR$ between 0.3 and 0.5; strong for $EDR$ between 0.5 and 0.8; and very strong for $EDR > 0.8$. The measured profiles of $\langle k \rangle$ and $\langle EDR \rangle^2$ are compared to each other for all three flights in Figs. 12a, c, and e.
For all flights, the $\langle EDR \rangle$ values indicate only weak turbulence was present. The most noticeable region of turbulence is just above and within the boundary layer, evident in Fig. 12a as an increase in $\langle k \rangle$ and $\langle EDR \rangle^2$ for $z < 2$ km, with slightly thicker regions of elevated $\langle k \rangle$ evident in Fig. 12b and c over the range $z < 4$ km and $z < 3$ km for Flights 2 and 3, respectively.

Above the boundary layer turbulence $\langle k \rangle$ and $\langle EDR \rangle^2$ are largely in agreement, although for stratospheric altitudes there are numerous localized regions where elevated $\langle k \rangle$ values were measured during Flights 2 and 3. To investigate these regions
further we employ the continuous wavelet transform, an approach often used in time-frequency analysis due to its ability to discriminate frequency content within a signal as a function of time. Here, we employ the wavelet transform to examine the energy content of the measured velocity signal due to the presence of both low-frequency and high-frequency motions, which, as observed in Fig. 8, vary with altitude and, consequently, with time. Wavelet transforms eliminate the need for segmenting the time series, as required for calculating $F_{\ell\ell}(f)$, thereby avoiding potential bias introduced by the selection of segment

length. Thus, we chose to leverage the wavelet transform's capability to discern the time (and hence altitude) dependence of the frequency content of $u_\ell(t)$.

The continuous wavelet transform is defined through convolution of the time-dependent signal, here $u_\ell(t)$, with a wavelet function, $\Psi$, such that

$$W(a,b) = \int\limits_{-\infty}^{\infty} u_\ell(t)\frac{1}{a}\Psi^*\left(\frac{t-b}{a}\right)dt \qquad (16)$$

where $\Psi^*$ is the complex conjugate of the selected wavelet function, $a$ is the scale parameter, and $b$ the position parameter. The result of the transform is the wavelet coefficient, $W$, as a function of $a$ and $b$ which can be related to $f$ and $t$, respectively (see, for example, Tavoularis, 2005). The wavelet coefficient is therefore a complex-valued result whose amplitude reflects the frequency-dependent amplitude of $\Psi$ providing the best agreement with the original signal at each point in its time series. In this implementation, we utilized the analytical Morlet wavelet in Matlab over a frequency range from 0.001 Hz to 5 Hz. These

frequencies correspond to approximately the rate at which orbits were completed and half the maximum frequency response of the probe at low pressure and temperature conditions, respectively. The Morlet wavelet is a complex harmonic function modulated by a Gaussian envelope and provides a measure of the frequency content of a signal over a short time interval $\sim 3/f$ s long.

Figures 12b, d, and f present the wavelet transform of $u_\ell(t)$ for Flights 1, 2 and 3 respectively as logarithmically-scaled

isocontours of the complex modulus of the wavelet coefficient squared, $\log_{10}|W|^2$, which is referred to as the wavelet power spectrum. To transform the results from the time-frequency domain to the spatial domain, the plot is presented as a function of $z$ (which is time-dependent) and $\kappa_\ell \approx f2\pi|\langle \boldsymbol{V}_R\rangle|^{-1}$. Note that at the start and end of the time series the scale of $\Psi$ which can be convoluted with $u_\ell(t)$ is limited by the length of the available signal in time, and hence the low frequency boundary of the wavelet power spectrum shown in Fig. 12b, d, and f is time-dependent.

Noticeable in Figs. 12b, d, and f is the significant long wavelength content for $\kappa_\ell < 0.003$ (wavelengths larger than 2 km), with the highest coefficient values measured during Flights 2 and 3 when $z > 10$ km. This long wavelength content is the signature of the long period fluctuations in Fig. 8 and fluctuations with altitude measured in Fig. 10. High frequency content is also evident, not only in the proximity of the boundary layer but also in thin regions corresponding to altitudes where elevated $\langle k\rangle$ and $\langle EDR\rangle$ are observed in Figs. 12a, c, and e. For example, for Flight 1, distinct regions of elevated $|W|$ appear for

$\kappa_\ell < 0.1$ at $z \approx 8$ km and $z \approx 12$ km and with higher wavenumber content around $z \approx 17$ km. It can be observed that the $\langle EDR\rangle^2$ and $\langle k\rangle$ have the greatest agreement when $u_\ell$ contains high wavenumber content, i.e., when an inertial subrange is present (as implied by the approach used to determine $\langle \varepsilon\rangle$). Conversely when $\langle k\rangle$ is much larger than $\langle EDR\rangle^2$, specifically the larger spikes in $\langle k\rangle$ measured during Flights 2 and 3 above $z = 10$ km, it can be observed that these events correspond to instances where $u_\ell$ has energy content at low wavenumbers. Hence, the elevated $\langle k\rangle$ for $z > 15$ km in Figs. 12c and e appears

to be due to low wavenumber energetic contributions to $\langle k\rangle$ and the $\langle EDR\rangle$ calculation approach used here is unable to capture these low wavenumber contributions.

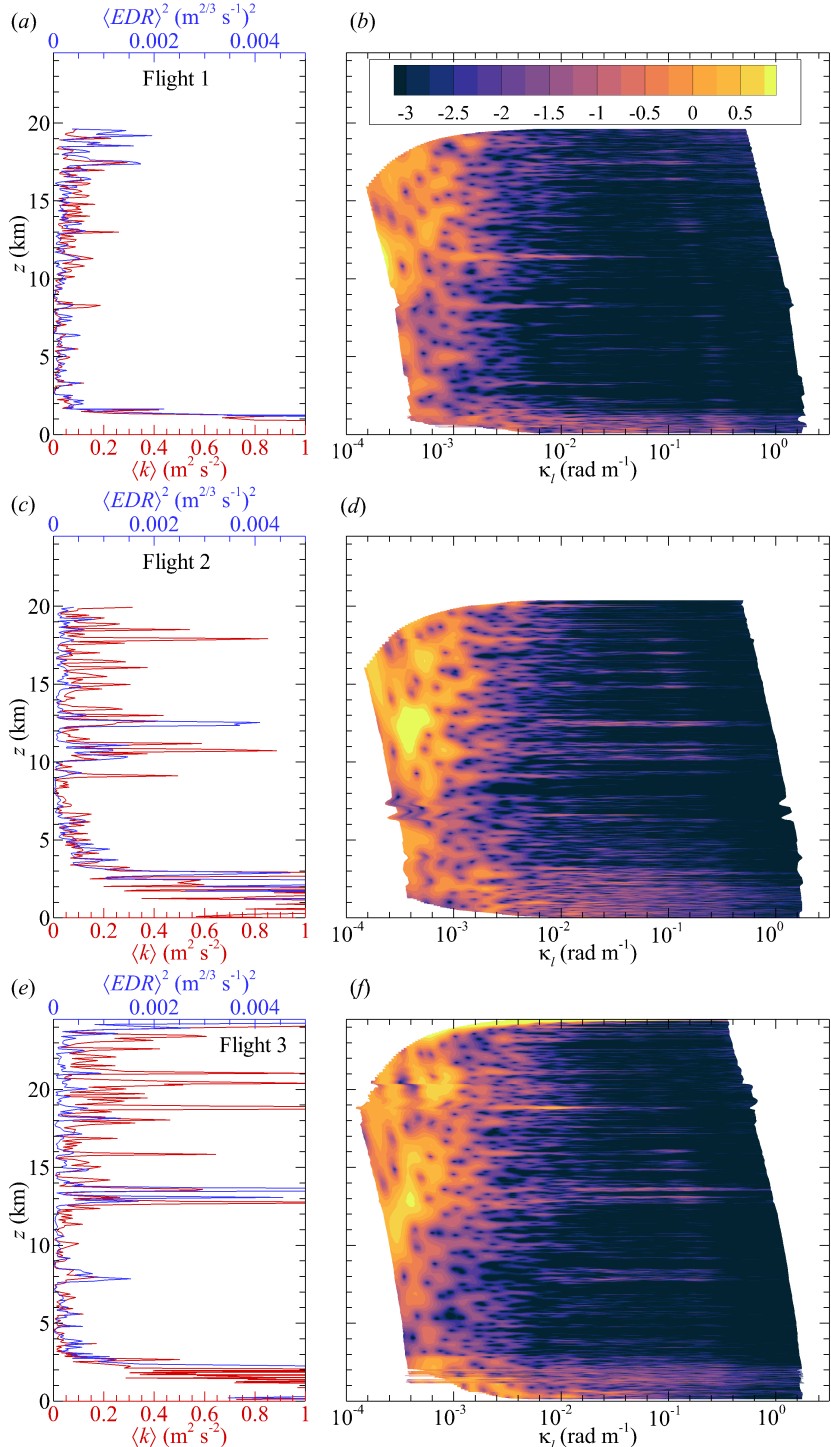

**Figure 12.** Profiles of $\langle k \rangle$ and $\langle EDR \rangle^2$ for (*a*) Flight 1, (*c*) Flight 2, and (*e*) Flight 3. Contours of $\log_{10} |W|^2$ are also shown for (*b*) Flight 1, (*d*) Flight 2, and (*f*) Flight 3 as functions of $\kappa_\ell$ and $z$. Contour levels are the same in (*b*), (*d*) and (*f*).

## 3.3 Infrasonic Detection of Turbulence

Before we can explore the connection between the infrasonic sound energy detected by the microphone and the turbulent kinetic energy measured by the five-hole probe, we need to quantify the measured infrasound energy. To do this we use the variance of the pressure fluctuations in the acoustic signal measured by the microphone, $\sigma^2$. We note that $\sigma^2$ will include pressure waves generated by the long wavelength fluctuations shown in Fig. 12b, d, and f (which indicate significant energy content in the velocity fluctuations at wavenumbers below 0.003) in addition to any inertial turbulence that may be present at higher frequencies. To capture this long wavelength motion and corresponding low frequency acoustic waves, we expand our segment length to 240 s (corresponding to the nominal time it took the HiDRON H2 to complete half an orbit) in order to ensure that the low frequency content is included in the variance calculation for both the infrasound and wind velocity fluctuations. Statistical averages calculated over this longer segment size are indicated by [ ] brackets. Profiles of $[\sigma^2]$ measured during all three flights are presented in Fig. 13a-c.

Noticeably, there was a decrease in signal amplitude measured with increasing altitude for all three flights. It was found that this decrease closely corresponds to the reduction in local atmospheric pressure, and therefore is attributed to increased atmospheric absorption due to the increase in molecular mean free path with altitude (Bass et al., 2007). Despite this altitude dependence, localized increases in $[\sigma^2]$ were observed, particularly near the boundary layer. When we isolate only the infrasonic part of the acoustic amplitude by integrating the power spectrum of the microphone signal over the frequency range below 20 Hz, which we refer to as $[\sigma^2]_{LF}$, we observe a similar but unequal attenuation, as also depicted in Fig. 13a-c.

Any altitude-dependent attenuation resulting from the decrease in molecular mean free path will be contingent on both local temperature and pressure and apply uniformly across the bandwidth of the microphone. Therefore, the ratio $[\sigma^2]_{LF}/[\sigma^2]$, which indicates the proportion of the total microphone energy measured within the infrasonic range, can serve to adjust for signal absorption with altitude, assuming consistent frequency response characteristics of the microphone across different altitudes. This ratio is compared to $[k]$ in Fig. 13d-f, with $[k]$ selected for this comparison over $\langle k \rangle$ due to its inclusion of the long wavelength turbulent kinetic energy content shown in Fig. 12d-f. When analyzed in this manner, the resulting profile of infrasonic amplitude shows a pronounced correspondence to the $[k]$ profile. Consequently, it appears that the infrasonic microphone is particularly sensitive to the long wavelengths detected in the stratosphere, and to a lesser extent, those in the vicinity of the boundary layer. However, the $[\sigma_{LF}^2]/[\sigma^2]$ profile appears as a filtered version of the $[k]$ profile.

Because the infrasonic microphone registers pressure waves that have traveled an unknown distance from their origin, whereas the five-hole probe can only measure the velocity fluctuations in-situ, we shouldn't anticipate a precise correlation between $[\sigma_{LF}^2]/[\sigma^2]$ and $[k]$. Moreover, the omnidirectional propagation of sound adds complexity, implying that the microphone may pick up sound from sources not measured by the five-hole-probe. Hence, these results can only provide qualitative support for the ability of the microphone to detect atmospheric turbulence, as a quantitative analysis would also require information about the distance and strength of the source of infrasonic energy.

It is worth noting that the normalization of $[\sigma_{LF}^2]/[\sigma^2]$ employed here does not capture the rise in $[k]$ observed near the boundary layer. This is due to the increase in $[\sigma]$ caused by the heightened broadband turbulent activity in the boundary layer.

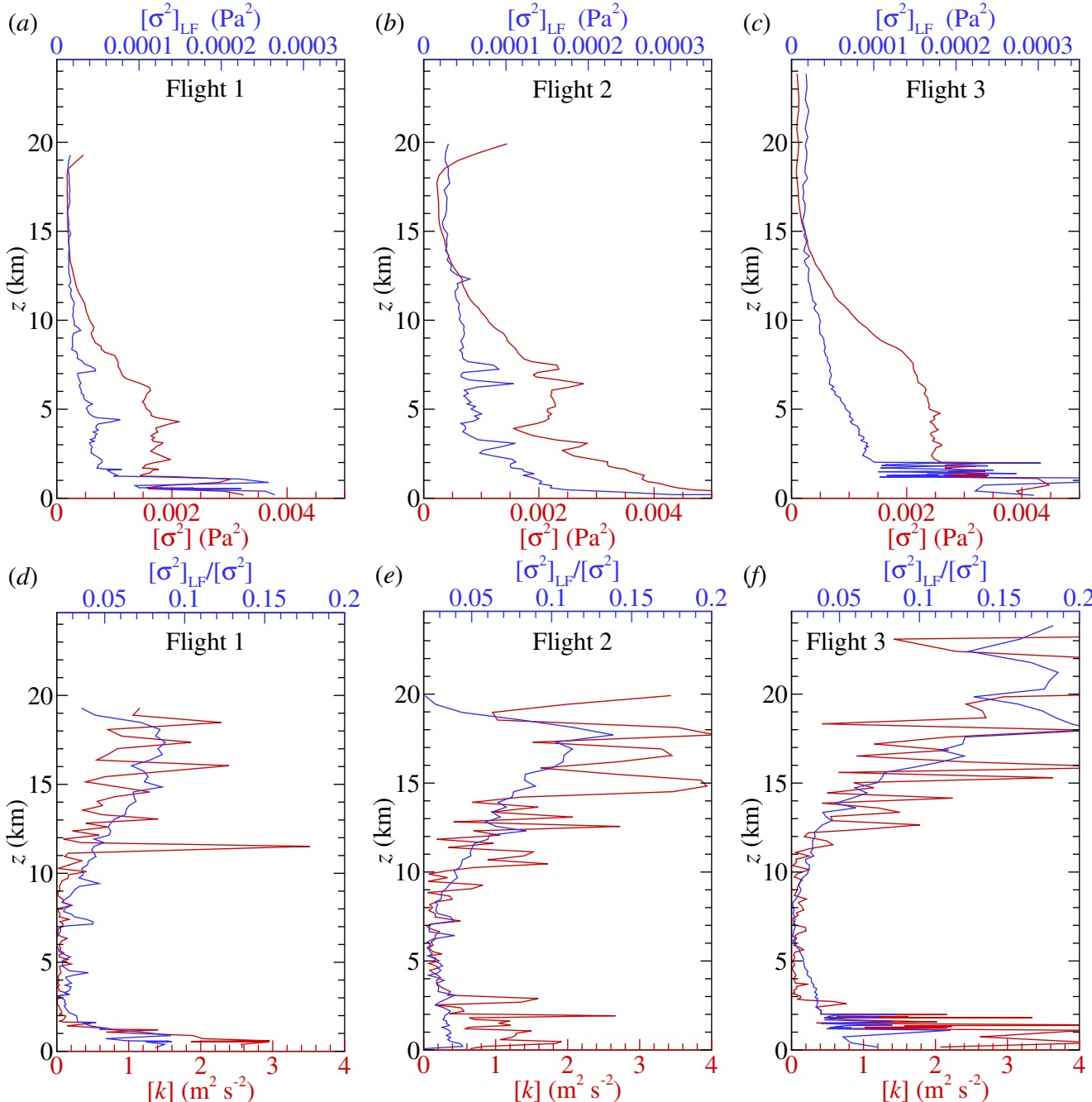

**Figure 13.** Infrasonic microphone signal as measured through its variance, $[\sigma^2]$, compared to variance of the signal low-pass-filtered at 20 Hz, $[\sigma^2]_{LF}$, for (*a*) Flight 1, (*b*) Flight 2, and (*c*) Flight 3. The ratio $[\sigma^2]_{LF}/[\sigma^2]$ compared to to turbulent kinetic energy, $[k]$, for (*d*) Flight 1, (*e*) Flight 2, and (*f*) Flight 3.

Consequently, the ratio $[\sigma^2_{LF}]/[\sigma^2]$ remains relatively constant across the boundary layer. However, Figs. 13a-c demonstrate that there is a rise in $[\sigma^2_{LF}]$ within the boundary layer, aligning with the increase in $[k]$. This observation is consistent with the ground-based findings of Cuxart et al. (2015).

## 3.4 Stability Conditions

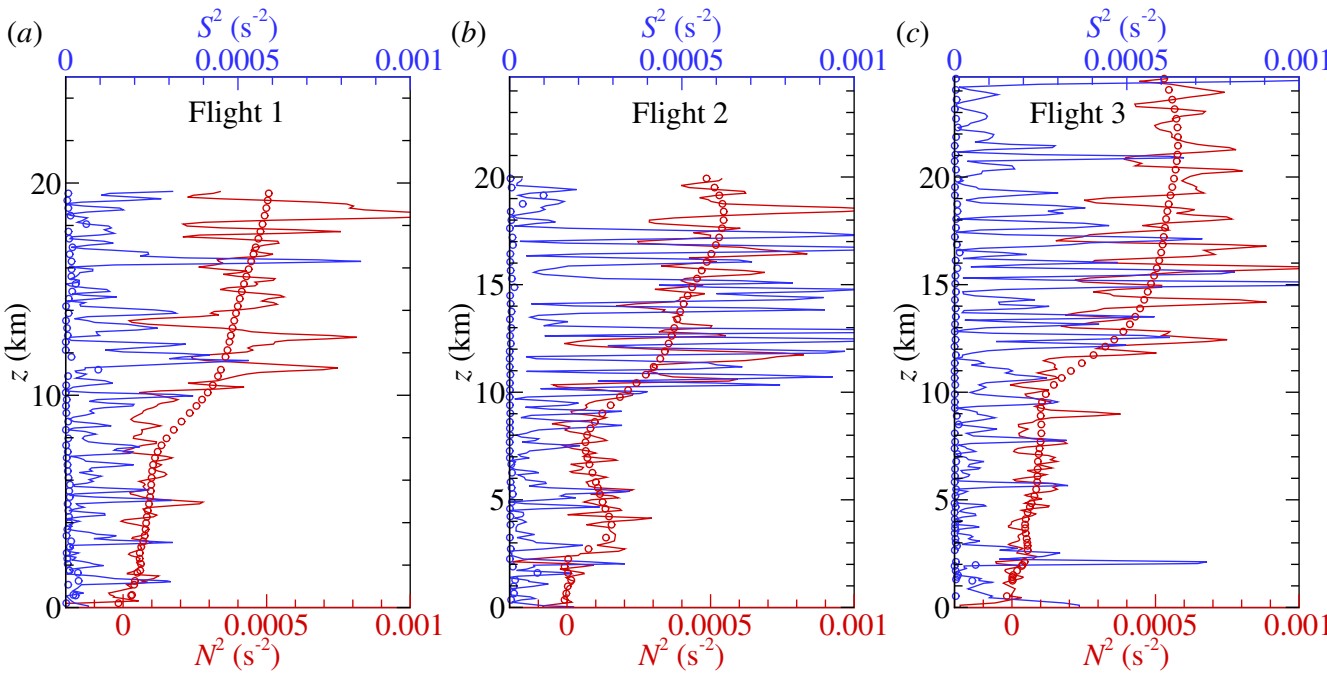

**Figure 14.** Square Brunt–Väisälä frequency, $N^2$, and square shear frequency, $S^2$, profiles for (a) Flight 1; (b) Flight 2; and (c) Flight 3. Solid lines show profiles calculated using vertical gradients calculated along flight trajectory, symbols show profiles calculated using vertical gradients calculated between neighboring orbits at same azimuthal position.

Profiles of the square Brunt–Väisälä frequency, $N^2$, and square shear frequency, $S^2$, provide some perspective about the altitude dependence of static stability and horizontal velocity shear, and hence potential for turbulence generation. However, approximating the mean vertical gradients of wind and virtual potential temperature required to calculate $N^2$ and $S^2$ using Equations 5 and 6 becomes somewhat ambiguous the HiDRON H2 measurements. This is due to the shallow glide slope of the particular helical flight path flown in these experiments, which introduces an increased sensitivity to horizontal gradients of $U$
and $\theta_v$ compared to that of the vertical gradients.

Therefore, different approaches were attempted to calculate $N^2$ and $S^2$. For the initial approach we present here, we assume horizontal homogeneity and attempt to replicate the expected response of a radiosonde to changes in the vertical wind structure, $\partial U/\partial z$. To do so, we assume a radiosonde time response for wind on the order of 40 s (Dirksen et al., 2014) which is introduced in commercial radiosondes by filtering of the pendulum effects and GPS noise, as well as the inertia of the balloon itself
(Scoggins, 1965). This time response, when combined with a typical rise rate of 5 m s$^{-1}$, equates to a vertical resolution of

approximately 200 m. Therefore, prior to calculating the vertical gradients, $u(t)$ and $v(t)$ were filtered using a moving average over a time span equivalent to a change in $z$ of 200 m. Then, to enable calculation of gradients in the vertical direction over $\Delta z = 100$ m intervals (the approximate turbulent layer thickness estimated by Ko et al., 2019), the wind components $u(t)$, $v(t)$ and $\theta_v(t)$ were then averaged over $\Delta z = 100$ m bins allowing calculation of the vertical gradients using the resulting

bin averaged values separated by $\Delta z = 100$ m. However, these gradient values were no longer at the same $z$ location as the location where $\langle \theta_v \rangle$ values were calculated. Therefore, prior to calculating $N^2$ and $S^2$, the gradients calculated at a resolution of $\Delta z = 100$ m had to be interpolated to the $z$ location corresponding to each statistical segment.

The square Brunt–Väisälä frequency and square shear frequency calculated this way are presented as solid lines in Fig. 14 for each flight. The results show large oscillations with altitude for all three flights, particularly in $S^2$, which can be expected

from Fig. 10. However, when $Ri = N^2/S^2$ is compared to both $\langle k \rangle$ and $\langle EDR \rangle$ as done in Fig. 15a and b respectively, the resulting trend is generally consistent with the expected behavior of increased turbulence intensity at low $Ri$. Specifically, the high turbulence intensity events (independent of whether quantified through $\langle k \rangle$ as done in Fig. 15a or $\langle EDR \rangle$ as done in Fig. 15b) largely occur when $Ri < 1$, although there are some instances of elevated turbulence when $Ri > 1$ and instances where no turbulence is evident when $Ri < 1$.

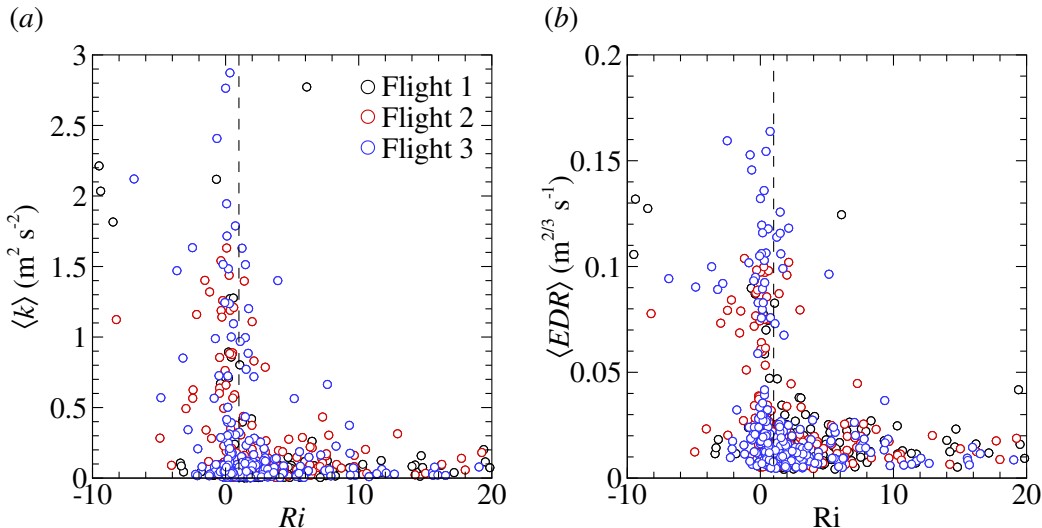

**Figure 15.** Comparison of (a) turbulent kinetic energy, $\langle k \rangle$, and (b) eddy dissipation rate, $\langle EDR \rangle$, to gradient Richardson number, $Ri$, calculated along flight trajectory. Dashed vertical line indicates $Ri = 1$.

As the helical descent of the aircraft results in approximately 1500 m of horizontal travel for every 100 m of decent, horizontal variability in the measured variables have an outsized impact on the calculation of $Ri$. In an attempt to eliminate the required assumption of horizontal homogeneity for the calculation of $Ri$, an alternative approach was attempted to calculate vertical gradients whereby the vertical profiles were expanded along the circumference of the aircraft's orbit. For each statistical segment the two nearest segments in the $z$ direction were used to calculate the vertical gradients using central differencing.

In this way, the gradients were calculated vertically across individual orbits, rather than along the helical path. However this results in a gradient calculation with a vertical scale of the same order as the pitch of the helix, which was approximately 2.5 km and therefore much larger than the thickness of the layers of turbulence observed in Fig. 12.

The values of $S^2$ and $N^2$ calculated this way are shown using symbols on Fig. 14a-c. This calculation approach effectively reproduces the trend of the $N^2$ vertical profiles calculated along the flight path, but contains fewer fluctuations, particularly for 730 $z > 10$ km. The corresponding $S^2$ profiles, however, were an order of magnitude smaller when calculated across the orbits than when calculated along the flight path (reflected in the values near zero in Fig. 14a-c) and therefore resulted in $Ri$ values on the order of $10^3$. The lower values of $S^2$ when calculated between the orbits likely reflects the larger vertical distances involved in the differencing across the orbits which will effectively filter out many of the vertical wind gradients shown in Fig. 10a-c. This approach is presented here as it may be more successful for flight trajectories with smaller orbit diameter, and smaller vertical 735 pitch between orbits.

## 4 Summary and Conclusions

This report describes a balloon-launched glider UAS and instrumentation intended for measuring the statistical structure of atmospheric turbulence. To conduct these quantitative measurements, the aircraft was equipped with a five-hole-probe for measuring the three-component wind vector, and pressure, temperature and humidity sensors integrated into the aircraft. 740 These instruments allowed the measurement of vertical profiles of $T$, $RH$, $U$ and $\gamma$ up to $z = 25$ km above ground level. These profiles compared favorably with the nearest publicly-available National Weather Service radiosonde data for $T$ and $RH$, although there were some noticeable differences in the $RH$ values, which were lower than the radiosonde values for $z > 7$ km. There were also indications that additional corrections may be needed to $T$ within the stratosphere. Wind profiles had a median difference on the order of 2.5 m s$^{-1}$ with the available National Weather Service radiosonde profiles and the 745 GPS-derived wind velocity measured during ascent.

The descent trajectory allowed for the calculation of $\langle k \rangle$ and $\langle \varepsilon \rangle$ over a large horizontal distance relative to the vertical distance traveled. The resulting vertical profiles suggest that isolated regions of weak turbulence were present in all three flights, and by examining the wavelet transform of the longitudinal velocity it was observed that these isolated regions could have different wavelength content depending on altitude. The wavelet transforms also indicated that long-wavelength fluctuations 750 provided a significant contribution to the energy content for $z > 10$ km. The on-board infrasonic microphone captured these long-wavelength fluctuations as low-frequency acoustic energy. As the UAS approached the boundary layer, the microphone also detected a rise in broadband acoustic energy, consistent with the broader frequency content of turbulence within this layer measured by the five-hole-probe. While these findings are qualitative, they imply that the infrasonic microphone is responsive to long-wavelength turbulent kinetic energy. Future research aims to establish a more quantitative relationship, necessitating 755 additional information about the distance of the sensor from the source of turbulent kinetic energy, as well as the turbulence strength at the source.

An attempt was also made to calculate the gradient Richardson number from the temperature and wind profiles, however the helical flight trajectory of the UAS introduces ambiguity in the time and length scales used for determining the mean vertical gradients. A method was proposed which smoothed the wind velocity prior to calculating the gradients across 100 m vertical increments. This resulted in gradient Richardson numbers that corresponded to enhanced $\langle k \rangle$ and $\langle \varepsilon \rangle$ when $Ri < 1$, however there were outliers which suggested improvements could be made to the calculation approach. An alternative approach was presented which calculated the vertical gradients across the pitch of the helical flight path. This produced similar results for $N^2$, however the $S^2$ results were an order of magnitude smaller due to the pitch being on the order of 2.5 km. The result was very high $Ri$ values that predicted stable conditions even when turbulence was present.

Despite this ambiguity, these initial flights suggest the potential exists for this measurement approach to be used for high altitude turbulence research, for example enabling detailed analysis of length scales and anisotropy of the turbulence. However, additional flights will be beneficial for assessing the capabilities of this measurement technique. For example, for the three flights considered here, the turbulence above the boundary layer was relatively weak and limited in occurrence. Measurements in more unstable conditions would lead to increased opportunities to conduct turbulent observations. Furthermore, flight patterns can be designed to allow examination of potential inhomogeneity of the statistics, for example by taking advantage of the helical trajectory taken by the glider during its descent to produce depictions of the horizontal heterogeneity of measured statistics. Additional flight patterns can also be designed with tighter helical descent to better calculate gradient Richardson number, with higher vertical resolution. Finally, improvements can be made to the data acquisition system that would lessen the signal-to-noise ratio of the acquisition system, simplifying post-processing of the measurements and providing higher resolution of instances of any turbulence encountered.

*Data availability.* Data from these flights are available from the corresponding author on request.

*Video supplement.* A video compilation of aircraft preparation, flight and recovery is publicly available at https://vimeo.com/568101900

*Author contributions.* S.B., G.P., N.C. planned the experiment which was realized by S.B., R.N., G.P. and A.S.. Data analysis was conducted by A.H. and S.B. who also prepared the initial draft of the manuscript with input from the remaining authors.

*Competing interests.* The authors declare no competing interests are present.

*Acknowledgements.* Financial support for this work was provided by NASA under the Flight Opportunities Program through award number 80NSSC20K0102 with Paul A. De León as the NASA Technical Officer. The authors would also like to dedicate this work to Dr. Qamar Shams who worked at NASA Langley Research Center and graciously assisted with the implementation and the loan of the infrasonic microphone from NASA.

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

## Appendix A: Comparison of wind and Reynolds normal stress profiles with and without use of all transducers

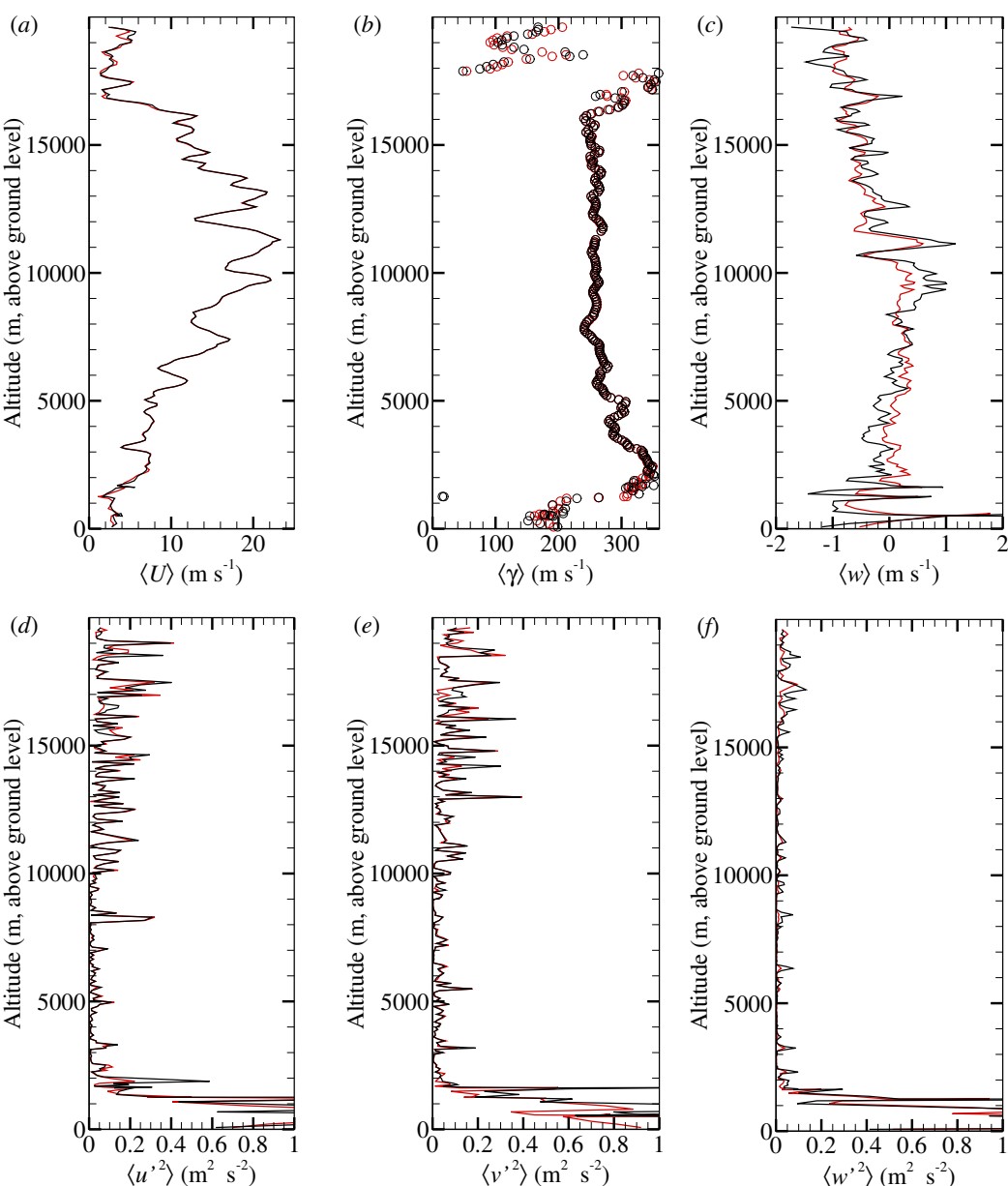

**Figure A1.** Figures showing comparison of (*a*) horizontal wind velocity magnitude, (*b*) horizontal wind direction, and (*c*) vertical component of wind velocity calculated using the $\Delta P_1$, $\Delta P_{32}$, and $\Delta P_{54}$ transducers to find $Q$, $\alpha$ and $\beta$ (shown in red) with the same properties calculated using only the $\Delta P_1$ and $\Delta P_{32}$ transducers to calculate $Q$ and $\beta$ with $\alpha$ determined from the aircraft angle of attack measurement (shown in black). Comparison of resulting (*d*) $\langle u'^2 \rangle$, (*e*) $\langle v'^2 \rangle$, and (*f*) $\langle w'^2 \rangle$ Reynolds stress tensor components is also shown.

## Appendix B:  Upper Air Maps and Satellite Imagery

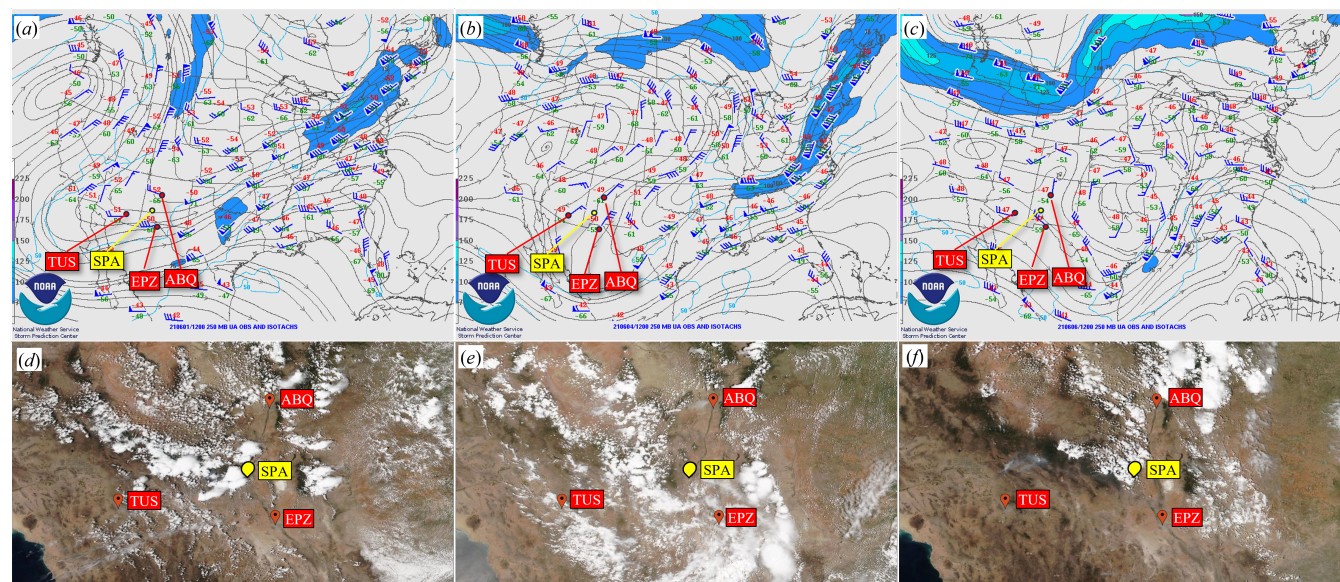

**Figure B1.** NOAA Jet stream maps at 250 mb (10.366 km) for (a) Flight 1, (b) Flight 2, (c) Flight 3 and satellite images of cloud cover (NASA) for (d) Flight 1, (e) Flight 2, (f) Flight 3. Red pins indicate NWS sounding sites and yellow pin indicates measurement location at SpacePort America (SPA).