# Peer review of "High-altitude balloon-launched uncrewed aircraft system measurements of atmospheric turbulence and qualitative comparison with infrasound microphone response"

_EGUsphere, 2023_

## Referee Comment (RC2)

**Review of the manuscript "High-altitude atmospheric turbulence and infrasound measurements using a balloon-launched small unscrewed aircraft sustem" by A. N. Haghighi et al.**

**General comments:**

The authors present an exciting UAV system that mainly includes a five-hole probe and an infrasonic microphone for probing turbulence in the troposphere and lower stratosphere. The technical developments are unquestionably to be welcomed. They may represent a new step towards a technology better suited to in situ turbulence and small-scale structure measurements at high altitude, particularly in the stratosphere, by providing decisive information on the coupling between the fine-scale stratification, mixing processes and gravity waves. However, the proposed article appears to have major shortcomings that should be remedied before a possible publication. If some of the issues are due to misinterpretation, the description of the methods should be clarified. There are also a number of inaccuracies or blunders which should also be corrected and some parts should be expanded to facilitate the interpretation of the results. Consequently, in view of the potential interest of this work, I propose that it be *publishable after a very thorough revision and perhaps re-evaluation of certain parameters*. The review does not comment the part relative to infrasonic measurements because I don't have enough knowledge to evaluate it.

**Major comments**

**(1)** The proposed comparisons between temperature, humidity and wind profiles from radiosondes and sUAS cannot be conclusive, as they are made with data that are separated by several hours (up to around 6-7 hours) and launch sites separated by ~ 160 km (this information only appears on line 270 when comparisons are discussed), and the fields are not stationary. Under these conditions, performance evaluation is difficult, if not impossible, and cannot "allow validation", as stated line 248, because disagreements can always be explained by the non-colocalization and non-simultaneity of the measurements. It would have been more useful to include a radiosonde under the balloon during its ascent, in order to make more appropriate wind comparisons. The PTU IMET-XF data during the ascent may provide better conditions for temperature and humidity comparisons, even

if they are corrupted at small scales. On the other hand, the representation of data points with circles, rather than continuous lines, systematically used in the figures, does not allow simple comparisons between profiles and identification of the peaks referred to in the text. From a qualitative point of view, the three wind profiles measured by sUAS seem to indicate fluctuations compatible with the presence of gravity waves at *all* altitudes with a vertical wavelength of the order of 1 km or less, whereas the balloon data do not seem to reveal such fluctuations (but, once again, the graphical representation makes analysis difficult). The technical data of the radiosondes used (such as the vertical resolution) should be indicated. Current standard sondes are now able to measure profiles at 1 Hz, which does not seem to be the case here. We can wonder whether the wind fluctuations observed by the sUAS, and apparently not by the balloon, are the result of an instrumental artifact or not. A spectral analysis of the whole (sUAS and balloon) wind (and T) profiles would provide useful information to check the consistency between the data.

**(2)** The calculation of $\varepsilon$ described on page 18 contains several important errors that need to be corrected. The conversion of the frequency spectra into wavenumber spectra must be made with the magnitude of the *relative air wind speed*, NOT the aircraft's ground speed. Description of the method can be found in Frehlich et al. (JAS, 60, 2487-2495, 2003) or Kantha et al. (PEPS, 4:19, 2017). As the difference can be important, especially at high wind speeds, the impact on $\varepsilon$ should be far from negligible. As a corollary, the longest wavelength of velocity fluctuation is not given by the sole horizontal velocity of the aircraft, but by the relative velocity (lines 348-350). Because the *relative air wind speed may significantly vary with altitude* (which is why it should be quantified and shown), TKE estimated for frequencies $f < 5\ Hz$ and shown in Figure 9 is likely not correct because it indicates the total energy for *variable* wavenumber bands. In addition, $k_1$ as defined in (9) is NOT the component of the wavenumber vector in the direction of the flight path (line 362), but along the direction of the relative wind vector formed by the direction of flight path and horizontal wind vector. There is no difference between the two, only when the UAV flies in the direction of the wind. Therefore, TKE dissipation rates (and TKE) must be recalculated.

**(3) Section 3.4**

The method used to reconstruct the distributions of parameters in the $\alpha - z$ plane is not clear. The authors should very clearly explain how the interpolation method works as this is not a standard method of visualizing the data. However, it does not seem to be feasible. Because only one value is obtained for a given altitude, it is not possible to interpolate the distribution of a parameter for any value of $\alpha$ at this altitude. The method will systematically produce artifacts (isolated structures with vertical bands) unless the layer probed by the sUAS has a thickness at least greater than the vertical distance covered by the instrument to make 360°. It turns out that all the plots and discussions are incorrect, and section 3.4 should be deleted in its entirety, unless the authors can demonstrate the merits of their approach.

**Specific comments**

Line 8: see comment (1) above.

Line 34-41: Some important references on in-situ measurements of turbulence are missing, e.g., Barat and Bertin (JAS, 41, 819-827, 1984, and references therein), Bertin et al. (Radio Science, 32, 791-804, 1997), Alisse and Sidi (JFM, 402, 137-162, 2000), Gavrilov et al. (Ann. Geophys., 23, 2401–2413 2005). In addition, the manuscript ignores references from the radar literature (e.g. Sato and Woodman, JAS, 39, 2546-2552, Fukao et al., JGR, 1994, and many others). UHF or VHF clear air radars have enlightened the layering structure of the stratosphere, mentioned line 37, to be attributed to thin and horizontally extended turbulent layers or/and stable layers. Incidentally, the authors recognize that the horizontal stratification is a key feature of the stratosphere. Is this feature consistent with the "vertical structures" supposed to have been detected with the sUAS measurements in section 3.4?

Line 65-67: The reference list about turbulence measurements from sUAS must be thoroughly revised. Some of them are not about turbulence (Bäfuss et al., 2018; Rautenberg et al., 2018, Jacob et al., 2018) and many others are missing. For example: Lawrence and Balsley (JTECH, 30, 2352-2366, 2013), Balsley et al., (BLM, 147, 165–178, 2013), Balsley et al. (JTECH, 35, 619-642, 2018), Reuder et al. (Acta Geophysica, 60, 1454-1473, 2012), Shelekhov et al., (Atmos. Ocean. Phys., 57, 533-545), Kanthe et

al., (PEPS, 4:19, 2017), Luce et al., (JAS, 77, 231-2326, 2020), Calmer et al., (AMT, 11, 2583-2399, 2018), among others.

Line 86-87: "However, due to the transient nature of their Lagrangian flight trajectory, balloon-based approaches are not necessarily amenable to obtaining detailed statistical descriptions of turbulence at high altitudes." The comment is unclear. What do the authors mean?

Line 91-93: The introduction of the "infrasonic microphone" has already been made in the previous paragraph. It is this redundant. In general, the various paragraphs of the manuscript should be better organized to avoid such redundancies (they occur several times). This gives the impression of a juxtaposition of paragraphs with no guiding line.

Line 121: Please convert km/h into m/s. The controllability of the sUAS is an important parameter and more information about limitations and performance should be given. We understand that the UAV can safely fly for wind conditions up to 31 m/s at least. How is the horizontal velocity of the glider controlled? It must be significantly high than 31 m/s for stability. A figure showing the ground speed of the sUAS with altitude (and the relative air speed, for the reason described in (2) Major comments) would be informative.

Line 129: please explain why the sampling is made at 10 Hz.

Line 192: The authors seem to indicate that the five-hole probe sensor had an effective time response of 0.1 s. The reason is unclear (but I do not have the background to understand). The corresponding spectra should show a gap from ~ 10 Hz. It is roughly observed in Figure 8 at z=10 km (but around 5 Hz) and there is no evidence of a transition at z=1 and 18 km. How do the authors interpret this feature?

Line 227: The reason(s) of the choice of large values in circle radius (1-5 km) is not explained.

Line 229: A figure showing the descent rates of the sUAS with altitude would be useful to figure out the conditions of sampling.

Line 243: "horizontal distances": with respect to the ground? If yes, it means the ground speed of the sUAS varied between 10 m/s and 80 m/s but this information is not provided in the manuscript. If these distances are expected to characterize the largest scales

sampled during 30 sec by the instrument, it is not correct, because the relative speed should be considered (see (2) of Major comments).

Line 252-271: These two paragraphs must be re-written. They are particularly confusing and not rigorous. For example:

- the first paragraph seems to describe general properties for the 3 flights but the second paragraph focuses on flights 2 and 3. So, we *deduce* that the description made in the first paragraph is for Flight 1. In addition, almost all the statements are disputable or not well-introduced. The agreement within 10% is unclear (please add information/figure that corroborates this result) and it is not true for  at all altitudes for example. But the paragraph focuses on temperature profiles. So, does this quantification only apply to temperature? But then why introduce the paragraph with "with the exception of RH"…?

- The altitude of the top of the boundary layer is estimated to be "roughly lower than 3 km" in the first paragraph, but up to ~5 km in Flight 3. First, these altitudes (especially 5 km) are not realistic even for convective boundary layer. Second, the criteria used to estimate this altitude are not explained. Third, there is *no* indication consistent with these estimates in the figures.

- The figures are not correctly labeled: "Fig. 5(a,c,e,)" should be "Fig 5 (a,b,c)". 'Fig. 5c' should be 'Fig 5b'. Fig.5 b, d, f should Fig 5 d.e.f. and at other places. Please check.

- What is the criterion used to define the tropopause altitude? It is found at 11 km (line 258) (presumably for Flight 1 in the first paragraph), quite consistent with figure 5a, but indicated to be at 12.5 km on line 278. It is found at 13 km for Flight 2 while the temperature inversion is actually observed at 11.5 km in figure 5b. It is stated that it is at 14 km for Flight 13, but an inversion can be found at 12 km. How are quantified the lapse rates in the troposphere and stratosphere and what is the interest to estimate such values (and tropopause altitudes) if they are not compared between the instruments? This comment also applies to humidity and velocity profiles, since the text does not describe the differences and similarities between the profiles, but rather their characteristics, which is another objective. There are too many caveats.

From lines 290: This part should be separated from the previous ones because it is not about comparisons between radiosonde- and sUAS-derived profiles anymore. It seems to me important to show $N^2$ (squared BV frequency) and shear profiles before describing Ri profiles, since one of the purposes of the manuscript is to assess the performance of the sUAS measurements. In addition, low Ri values can have different causes, i.e. a strong shear and/or low $N^2$. The knowledge of these two parameters can help the interpretation of the turbulent events.

Equation (7): In practice, the impact of the variation of g with altitude can be ignored. The error is much less than all the other uncertainties.

Line 301: "Here we assume the critical Richardson number takes on a value somewhere in the range 0.25<Ri<1". It is unclear. Ric=0.25 is a necessary condition below which air can become dynamically unstable and turbulent (if Ric>0.25, a shear instability cannot develop). Once turbulent, there is a critical Richardson number at which the flow begins to laminarize: it is generally accepted to be between 0.2 and 1 but turbulence can be found for Ri >>1 according to Galperin et al. (2007) (but the corresponding turbulent regime should strongly differ from the turbulent regime for small Ri values). In practice, these thresholds must be used with caution because the Richardson numbers estimated from in-situ data are scale-dependent, i.e. depend on the vertical resolution at which they are calculated. It is common to apply arbitrary thresholds (i.e. "Ri is minimum and small ~0.25-1")

Figure 8: the information is interesting. Why not showing the corresponding spectra for v and w? In Figure 8b and 8c, it should be F instead of Φ (y label).

Line 333: I am again skeptical about the interpretation of the increased TKE layer up to 4 km as corresponding to the CBL top

**Figure 9**: EDR and TKE should be presented in logarithm scale (and continuous lines), because the linear scale over-represents the maxima near the ground. Indicating that "TKE is close to 0", line 333, is symptomatic of the fact that the linear scale is unsuitable for the present purpose. Figures *showing the slopes, as calculated in Figure 8 for 3 cases, vs altitude and for the 3 wind components* should be included. It would enable us to identify the altitudes where the inertial slope is indeed observed, those where a different

regime is observed, and those where instrumental noise is dominant for all frequencies. This information is *essential* for the purpose of the manuscript. A characterization of the slopes vs other parameters (e.g. TKE, EDR, Ri, etc) could be very informative.

The dataset offers the possibility to show $\langle u'^2 \rangle$, $\langle v'^2 \rangle$ and $\langle w'^2 \rangle$ separately. Plotting, for example, $\langle u'^2 \rangle$ vs $\langle w'^2 \rangle$ would be interesting for quantifying anisotropy. A discussion of this anisotropy in light of $\varepsilon$, Ri, etc, would be very enlightening.

TKE and $\varepsilon$ are related by a master length scale (e.g. Mellor and Yamada, Rev. Geophys. Space Phys., 20, 851-8751982) which of great interest for the characterization of turbulence. The dataset shown in Figure 9 offers the potential to estimate this scale.

 Line 337: "…caused by *inertial* turbulence" and remove "elevated" in the same sentence because the comment is valid for all levels of TKE. The spectra with a -1 slope may either reveal another turbulent regime or be due to a white noise contamination even for f< 5 Hz when the atmospheric signal is weak. Figure 8c is apparently in favor of the first interpretation for the selected case but it is not necessarily always true, especially when the instrumental noise dominates.

Line 343: "…more active turbulence conditions during these flights". This statement should be nuanced because (1) the rejection was based on the u spectra only, (2) the non-detection of a -5/3 slope does not mean the absence of turbulence, (3) the corresponding levels of EDR of flight 1 (qualified as "weakly active") shows a significantly higher background than flight 2, indicating higher spectral levels but not consistent with an inertial subrange.  As we do not know about the interpretation of the observed non-inertial subranges, turbulence activity cannot be qualified.

Line 345: Please remove this sentence. The reference of Kelvin waves is not suited here because equatorial Kelvin waves (Fujiwara et al. 2003) are waves trapped around the Equator similarly to coastally-trapped Kelvin waves.

**Equation (9):** The notation $\Phi_{11}$ of the spectrum may not be appropriate ($F_{11}(k_1)$ would fit better the notation used in Figure 8). $\Phi_{ij}$ generally refers to the spectral density tensor (see e.g. Doviak and Zrnic', Doppler radar and weather observations, p. 326, 1984). The sUAS is "sensitive" to the 1-D *longitudinal* spectrum (see e.g. Hocking (EPS, 1999)).

Line 367 refers to $\Phi(k_1)$ ($\Phi_{11}$?) and F ($F_{11}$?). Line 368: $k_1^n$ should be $k_1^{-5/3}$. Strictly, the power-law fitting should be applied to a limited range of $k_1$ since the smallest wavenumbers are not well-resolved.

**Technical comments**

Line 4-5: The sentence is unclear, please rephrase.

Line 20-21: please add references.

Line 24: "Despite the higher stability of the stratosphere"

Line 25, see also line 318: "…due to mechanical and thermal disturbances" -> "due to shear instabilities and gravity wave breaking". The formulation is unsuitable because the mechanical sources of turbulence refer to those produced by obstacles close to the ground. The rest of the paragraph is awkward –and not rigorous- and references of the "classical" literature should be included instead.

Line 38: "high turbulent kinetic energy dissipation rate": please be more quantitative.

Line 45: "and for identifying the inner scale of turbulence."

Line 45: "This experiment": please be more specific with references.

Line 45-47: The Richardson number is not defined and it is not explained why Ri=0.25 is an important value and why turbulence when Ri >0.25 should be noted. The authors should indicate that some LITOS results were corrupted (Soder et al., AMT, 2019) due to balloon wake and that turbulence observed when Ri >>1 was suspect. In addition, useful information on the relationship between Ri and $\varepsilon$ can be found in earlier references mentioned in the specific comments (line 34-41).

Line 63-72. Pitot tubes are also used (e.g. Lawrence and Balsley (JTECH, 30, 2352-2366, 2013).

Line 90: "air masses" generally refer to "large bodies of air" at synoptic scales in meteorology. The term is not suitable here.

Line 90: "geostationary" usually refers to satellite orbits. Do the authors mean "relatively constant location above the ground?"

Line 92: ""traditional" -> "standard"

Paragraph 2.1: It seems more natural to present the instruments first, then the configuration of the experiment. 2.1 -> after 2.3.4 and before 2.4

Fig.1 : Please add the location of the balloon launch site (El Paso) and show the distance in km (in Fig. 4 also). The distances are crucial for the interpretation of the radiosonde and sUAS data and longitude/latitude coordinate system is of little use here.

Line 102: "Three *sUAVs* were flown" (?)

Line 106: The altitudes are given in km m.s.l but the profiles are shown from z=0 (i.e. above the ground (line 241). Please indicate the corresponding altitude AGL, even if it can be roughly deduced from Fig. 1.  The third sUAS was released from 30 km m.s.l (~28.5 km AGL ?), but the profiles are shown up to 25 km. Please clarify.

Line 191: "… the actual probe *frequency* response…"

Line 194: remove "disconnected"

Lines 232-235: Please indicate Local Time instead UTC (and avoid MDT). By doing so, the reader does not need to convert by himself when interpreting the PTU profiles measured by the sUAS and the radiosondes at different times (Figure 5).

Line 246-249: The first sentence is not necessary and the second has already been written. A more detailed description of the radiosonde data is necessary (see (1) of major comments).

Figure 5: please add LT times for all the radiosonde and sUAS flights. A figure showing the trajectories of (and horizontal distance between) both instruments is necessary.

Line 258: For flight 1, "the temperature continued to decrease with altitude at a rate of 1C/km" -> the temperature continued to decrease at a mean rate of 1C/km between 11 and 19 km" (otherwise it is confusing, see specific comments also)

Line 274: please add "(not shown)".

Line 281: please show the NOAA upper air wind maps. The absence of reference points make difficult to confirm the statements.

Line 282-285: what do the authors mean ?

Line 307: The term "potential instability" refers to "an atmospheric condition in which otherwise stable air would become unstable if forced to rise (e.g. over high ground) thereby reaching its saturation point." See e.g. www.encyclopedia.com. Please replace "potential" by "possible shear".

Line 309: "marginally unstable tropopause": what do the authors mean?

Line 316: Here again, the terminology is improperly used. "An atmosphere is said to be "conditionally unstable" if the environmental lapse rate is between the moist and dry adiabatic lapse rates. This means that the buoyancy (the ability of an air parcel to rise) of an air parcel depends on whether or not it is saturated." (see glossary of meteorology) . "suggesting the possibility of localized buoyant production". Do the authors refer to statically unstable conditions, ie. Ri <0? If yes, it must be clearly stated and defined earlier, e.g. around line 302.

Line 318: "mechanical turbulence": see above, comment for line 25

Line 328: Do the Hanning window preserve variance?

Line 329: Do the cut-off at 5 Hz related to the effective limited time response indicated line 192?

Line 334-335:The description is unclear because of the use of a linear scale and a dot representation (see "specific comments")

Line 336: Please remove "although the regions of elevated k appear at different altitudes". This comment is not useful.

Line 344: Please indicate the altitude of the tropopause in Figure 9.

Line 354: "As direct measurements of …" please explain more or add a reference

Line 355-356: The sentence indicates a condition that has no reason to exist at this stage, since inertial domains have been identified by spectral analysis.

Line 520: Please remove "flux". The gradient Richardson number and the flux Richardson number have two distinct definitions.

---

## Author Comment (AC1)

**Response to Referee 1**

We appreciate the time taken by all referees in providing insightful and detailed comments about our manuscript. Following the reviewers recommendations, numerous changes have been made in the analysis approach including: (1) Revisiting the conversion of the five-hole-probe wind estimate; (2) adjusting the statistical ensemble sized from being based on 30 s ensembles in time to 1500 m ensembles in space (with 750 m overlap between ensembles to retain spatial resolution); (3) revised the spectra calculation with the correct transformation from frequency to time domain; (4) updated the method used to calculate vertical gradients and adding Brunt-Väisälä and shear frequency measurements to the paper; and (5) addressing issues in the methodology used to generate contour plots. Of these changes, the largest impact on the results was the change made to the vertical gradient calculation, which impacted the gradient Richardson number, $Ri$, values and the revision to the contour plots. As a result of these changes, and other additional changes made to address specific comments made by the reviewers, we believe the revised version of the manuscript is more clear and complete than the version originally submitted. To help clarify where changes have been made in the revised version, we have highlighted all changes made using blue text.

Below, we respond to the individual comments made by the referee. To do so, we have reproduced the original review, with our comments provided in blue text.

**Reviewer(s)' Comments to Author:**

Haghighi et al. present an exciting new measurement system that allows to sample the atmosphere up to 25 km with high resolution. It is great to see that the authors could perform these measurements and that they could measure exciting features in the atmosphere. I have some concerns about data processing and analyses which need to be revised before I can recommend the manuscript to be published in AMT.

General comments:

- I do not trust the Ri-number calculations which are presented. The oscillations that can be seen in the vertical profile seem unrealistic and I have a strong suspicion that the error starts with the wind measurement, as described below in the specific comments. Please check and make sure that the wind measurements are not heading-dependent and estimate the uncertainties for your system.

As we were also concerned with apparent wind magnitude variations with altitude having a vertical wavelength similar to the altitude difference between successive aircraft orbits, we have thoroughly revisited the wind measurements and found several places where improvements could be implemented, including improving the time alignment between autopilot kinematic variables and payload sensors, identifying and correcting pitch and yaw probe misalignment (of less than 10°) between aircraft and sensor coordinates, and discovering an error in probe rotation. However, these improvements only modified the wind magnitude by 10% at most and did not affect the observed vertical profile in any meaningful manner.

We have also closely examined the dependence of the vertical profiles of wind magnitude with heading, as shown in Figure 1. The most notable similarity between vertical separation of orbits and vertical wavelengths in wind magnitude occurred during Flight 1. However examining successive orbits shows that the wavelengths are not identical, with the orbit vertical distance slightly longer. Therefore, if there is a bias in the measurement, it is not rigidly correlated to the heading.

Note that a formal error propagation analysis is quite challenging for this type of measurement, as it involves numerous sensors on the aircraft autopilot (GPS, gyroscope, accelerometers, and associated Kalman filters, timing clock), on the payload (pressure transducers, thermistor, capacitive hygrometer, data acquisition systems, and timing clock) and on the wind tunnel calibration (Pitot probe, directional gimbal accuracy, data acquisition system). Therefore providing a formal uncertainty estimate is non-trivial. We therefore utilize the less formal approach by perturbing different elements of the calculation, and found that the wind estimate appears to be robust to most of these changes. Thus we can only roughly estimate our uncertainty using previous intercomparison studies with a very similar probe at $\pm 1$ m s$^{-1}$.

Finally, we note that the $Ri$ periodicity is directly related to the calculation approach, whereby we simply

[Figure]

Figure 1: Vertical profiles of wind magnitude from all three flights shown with horizontal lines indicating location where aircraft heading passed through 180°. (left) Flight 1 (middle) Flight 2 (right) Flight 3

calculated the vertical gradients from the time dependence with of altitude. However, this approach assumes horizontal homogeneity in properties along the aircraft orbit, which may not be strictly correct over the distances travelled, particularly with the wind measurements. Correspondingly, these values were also sensitive to the ensemble size used. We therefore present an updated approach within the revised manuscript that attempts to calculate $Ri$ using a more localized vertical gradient.

- The title and the abstract suggest a deeper analysis of the infrasound measurements, the error sources and the uncertainties, but the manuscript leaves me with the impression that these measurements can not really be used because it is often unclear where the detected signals originate from and how to process / correct them. It would be good to at least show a path forward for this.

We have expended significant effort examining the infrasound measurements, and your assessment is correct that although we gained understanding of the system and signals, we have only been able to identify qualitative relationships between the infrasound amplitude and the presence of turbulence. Ultimately, we can attribute this to two reasons: (1) the turbulence above the boundary layer that occurred during these measurements was insufficient to generate a signal significant enough to be sensed; and (2) we were unable to find a scaling that successfully removed the effects of density on the microphone response for all three flights. However, the increase in amplitude within the boundary layer and at some localized regions within the troposphere do indicate that the sensor was detecting turbulent signals. We believe that additional measurements with some planned modifications to the system will allow greater insight into how best to deploy the sensor operationally.

- The gravity wave analyses are very nice. The polar plots give some insight, but it would be easy to analyse the origin of the waves better when some model data is included and 2D maps of the wave structure at different altitudes was shown. I encourage the authors to consider that to make their very vague and speculative statements more robust.

Unfortunately, the appearance of gravity waves appears to have been contaminated by the two-dimensional interpolation technique used to in the $\langle T \rangle'$ and $\langle w \rangle'$ analysis used to discern the presence of gravity waves. Although we believe there is still merit in the concept behind the approach, we could not come up with a suitable implementation in time to include in the revision. We have therefore removed the gravity wave discussion from the revised manuscript.

Specific comments:

p.8, l.192: was the attenuation of the probing also verified by experiment? what can you read from the spectra?

The attenuation was measured directly by introducing a step pressure change at the probe tip and measuring the voltage response. The settling time following the step change was 0.06 s which corresponds to a frequency response of 20 Hz. The velocity spectra only show the tubing influence for $f > 100$ Hz, however we use 20 Hz in the revised manuscript as a conservative estimate of the highest frequency accurately captured.

p.8, l.195f: if this data is shown, you need to explain the method in more detail.

We have expanded on this discussion further in the revised manuscript.

p.11, l.253: Where do these 10% come from? What do they even mean? It does not look like there is a relative error below 10% for wind speed and wind direction at all times. I would not expect it, given the temporal and spatial separation, but the value should be explained.

We had been referring to an average difference. However, in response to reviewer 2's concerns about the radiosonde comparison, we have revised this section through inclusion of additional radiosonde measurements at different locations and updated the text accordingly.

p.14, Fig.7: The periodicity in the Ri-number with height is very suspicious. Looking at Fig.4 I get the feeling that this could be caused by flight direction. It is known that wind estimatino from multi-hole probes on fixed-wind aircraft is very sensitive to heading estimation. Please show the dependency of your wind measurements to heading. I think the periodicity already shows in wind speed and wind direction measurements. I doubt that the Ri-number calculations are meaningful with this uncertainty in the wind estimation. It should be qualified with an uncertainty estimation.

This comment has been discussed and, we believe addressed, above.

p.18, l.341: how were the thresholds for $n$ chosen here?

This range of values were somewhat arbitrarily selected as being within $-5/3 \pm 10\%$. We have added this range to the revised manuscript.

p.18, l.362: so, if I understand this correctly, in a circular flight, the horizontal flight direction changes all the time and thus does the wind vector component you are using for EDR estimation. I think it would be more reasonable to align the rotated wind vector to the mean wind direction. $u$ and $v$ are not expected to show the same spectral characteristics. In a circular flight you are also distorting the measurement, even if the Taylor hypothesis is valid. Maybe the radius is so large that within 30 seconds, the curvature can be neglected, but you should reflect on this.

The inertial subrange scaling described by equation 9 (equation 11 in the revised manuscript) is only valid in the longitudinal direction, i.e. specifically for the component of the velocity vector aligned with the component of the wavenumber vector. As the only wavenumber vector component we are capable of measuring is along aircraft's flight trajectory, this is the component of the velocity vector that should be included in the calculation of the energy spectrum. Although the mean value of $u$ and $v$ velocity components may not be equal, we subtract the mean value during the calculation, and therefore are only calculating $EDR$ using the velocity fluctuations in those directions (which should be homogeneous and isotropic in the inertial subrange assumed by equation 11 and therefore not impacted by direction of flight relative to the mean wind). Note that in our original calculation, we assumed the advection of the eddies due to the mean wind was negilible, which the other reviewers pointed out may bias the results. Both our wavenumber and velocity component used to calculate the spectra have been revised to adjust for the advection velocity and the manuscript has been updated with revised example spectra, $\varepsilon$ and $EDR$ values.

p.20, l.414: This is a bit misleading. You did not add horizontal flight legs at these altitudes, it is still the same flight pattern, spiraling down, right?

This is correct. This statement was only intended as a generality based on the aircraft descending only 2 km for every 30 km of horizontal flight along the orbit. We have revised this statement to be more precise.

p.21, l.428ff: These are quite interesting observations. It would help to show the temperature and velocity fluctuations for distinct altitudes on a horizontal (map) plot.

As noted above, we found that these fluctuations were contaminated by the interpolation scheme used and we have removed the discussion of these fluctuations in the revised manuscript.

p.21, l.429: How do you determine the wavelength?

Wavelength was determined by $d\lambda = Rd\alpha$ where $R$ is the radius of the aircraft orbit and $d\alpha$ the distance between peaks in the periodic observations.

p.22, Fig.12: The shaded region is Ri¿1 or Ri¡1? It is not clear from the caption.

the shaded region was intended to indicate indicates more stable conditions ($Ri > 1$). The revised $Ri$ calculation approach however has resulted in $Ri$ values much higher than unity, so we do not include this region in the revised manuscript.

p.25, l.465: you mean figures 12 d and e, right?

Correct. We have fixed this typo.

Figs. 12, 13, 14: The variable names and units should be given in the plot itself, not only in the caption.

We have added variable names and units to the figures.

p.27, l.534: I would highly recommend to obtain some reanalysis data from NWP models (e.g. ERA5) to see if conditions were favourable for gravity waves and if they can be seen in the model. This could be nicely added in an appendix.

We did contact someone familiar with satallite observations and investigated reanalysis data as part of the revision process. However, as noted above, the gravity wave discussion has been removed from the revised manuscript due to a lack of confidence in the initial results.

Technical corrections:

p.1, l.3: "thermodaynamic"

Correction made.

p.3, l.61: "from with"

Correction made.

p.7, l143f: two times "changes with the horizontal axis" should probably be vertical axis the second time.

Correction made.

p.8, Eq.4: *dir* is not a proper variable symbol.

We have changed *dir* to $\gamma/$

p.8, l.194: disconnected disconnected

Correction made.

---

## Author Comment (AC2)

**Response to Referee 2**

We appreciate the time taken by all referees in providing insightful and detailed comments about our manuscript. Following the reviewers recommendations, numerous changes have been made in the analysis approach including: (1) Revisiting the conversion of the five-hole-probe wind estimate; (2) adjusting the statistical ensemble sized from being based on 30 s ensembles in time to 1500 m ensembles in space (with 750 m overlap between ensembles to retain spatial resolution); (3) revised the spectra calculation with the correct transformation from frequency to time domain; (4) updated the method used to calculate vertical gradients and adding Brunt-Väisälä and shear frequency measurements to the paper; and (5) addressing issues in the methodology used to generate contour plots. Of these changes, the largest impact on the results was the change made to the vertical gradient calculation, which impacted the gradient Richardson number, $Ri$, values and the revision to the contour plots. As a result of these changes, and other additional changes made to address specific comments made by the reviewers, we believe the revised version of the manuscript is more clear and complete than the version originally submitted. To help clarify where changes have been made in the revised version, we have highlighted all changes made using blue text.

Below, we respond to the individual comments made by the referee. To do so, we have reproduced the original review, with our comments provided in blue text.

**Reviewer(s)' Comments to Author:**

Review of the manuscript "High-altitude atmospheric turbulence and infrasound measurements using a balloon-launched small unscrewed aircraft system" by A. N. Haghighi et al.

General comments:

The authors present an exciting UAV system that mainly includes a five-hole probe and an infrasonic microphone for probing turbulence in the troposphere and lower stratosphere. The technical developments are unquestionably to be welcomed. They may represent a new step towards a technology better suited to in situ turbulence and small- scale structure measurements at high altitude, particularly in the stratosphere, by providing decisive information on the coupling between the fine-scale stratification, mixing processes and gravity waves. However, the proposed article appears to have major shortcomings that should be remedied before a possible publication.

If some of the issues are due to misinterpretation, the description of the methods should be clarified. There are also a number of inaccuracies or blunders which should also be corrected and some parts should be expanded to facilitate the interpretation of the results. Consequently, in view of the potential interest of this work, I propose that it be publishable after a very thorough revision and perhaps re-evaluation of certain parameters. The review does not comment the part relative to infrasonic measurements because I don't have enough knowledge to evaluate it.

Major comments:

(1) The proposed comparisons between temperature, humidity and wind profiles from radiosondes and sUAS cannot be conclusive, as they are made with data that are separated by several hours (up to around 6-7 hours) and launch sites separated by ~160 km (this information only appears on line 270 when comparisons are discussed), and the fields are not stationary. Under these conditions, performance evaluation is difficult, if not impossible, and cannot "allow validation", as stated line 248, because disagreements can always be explained by the non-colocalization and non-simultaneity of the measurements. It would have been more useful to include a radiosonde under the balloon during its ascent, in order to make more appropriate wind comparisons. The PTU IMET-XF data during the ascent may provide better conditions for temperature and humidity comparisons, even if they are corrupted at small scales. On the other hand, the representation of data points with circles, rather than continuous lines, systematically used in the figures, does not allow simple comparisons between profiles and identification of the peaks referred to in the text. From a qualitative point of view, the three wind profiles measured by sUAS seem to indicate fluctuations compatible with the presence of gravity waves at all altitudes with a vertical wavelength of the order of 1 km or less, whereas the balloon data do not seem to reveal such fluctuations (but, once again, the graphical representation makes analysis difficult). The technical data of the

radiosondes used (such as the vertical resolution) should be indicated. Current standard sondes are now able to measure profiles at 1 Hz, which does not seem to be the case here. We can wonder whether the wind fluctuations observed by the sUAS, and apparently not by the balloon, are the result of an instrumental artifact or not. A spectral analysis of the whole (sUAS and balloon) wind (and T) profiles would provide useful information to check the consistency between the data.

We agree that the radiosonde comparison is not conclusive and tried to reflect that only a broad general agreement was present in the text. Unfortunately, at the time of the test flights, we did not have the capability to include radiosonde measurements in the test campaign and instead, as noted in the paper, used publically-available National Weather Service (NWS) soundings for comparison. As such, the resolution of these measurements were likely downsampled prior to being made available and we do not have access to the higher resolution original data set.

In the revised manuscript, we have added more information about the NWS radiosondes, and have added additional profiles made from two other weather stations nearest to the experiment site. We believe these additional soundings add context to the differences observed between the aircraft and weather balloon measurements.

We have added technical info about the NWS radiosondes to the manuscript.

With regards to using the iMet-XF data on ascent, we intentionally discard it as we found there to be non-negligible differences between the ascent and descent which are caused by the sensors placement. During ascent, they are in a stagnant region within the fuselage wake and therefore not sufficiently aspirated to prevent self-heating and delayed air exchange with the environment. They are also close enough to the fuselage to be within the thermal wake of the aircraft. The result was that the $T$ and $RH$ measurements had a significant altitude-dependent warm bias of 5 to 15 degrees K on ascent relative to descent and the radiosonde measurements. On descent, these sensors are introduced into the oncoming airflow and properly aspirated, resulting in better comparison with the radiosonde measurements. Note that the Hidron H1 used a similar setup and found good agreement with WRF model results on the descent (they also found significant difference between ascent and descent: Schuyler TJ, Gohari SMI, Pundsack G, Berchoff D, Guzman MI. Using a Balloon-Launched Unmanned Glider to Validate Real-Time WRF Modeling. Sensors. 2019; 19(8):1914. https://doi.org/10.3390/s19081914

With regards to the formatting of the profiles using symbols instead of lines, this is simply an artifact of a preference to use symbols to present discretely acquired data points wherever possible to avoid biasing the viewer regarding the interpolation between these points that comes with connecting the discrete data points. However, in the revised manuscript we have replotted all data presented in profiles using lines instead of data points.

In summary, although we now wish that we had conducted co-located radiosonde launches during these flight tests, we neglected to do so and have to resort to publicly available data for comparison. We've updated the comparison with additional soundings in the revised manuscript, but providing co-located data is not feasible without repeating the experiments.

(2) The calculation of $\varepsilon$ described on page 18 contains several important errors that need to be corrected. The conversion of the frequency spectra into wavenumber spectra must be made with the magnitude of the relative air wind speed, NOT the aircraft's ground speed. Description of the method can be found in Frehlich et al. (JAS, 60, 2487-2495, 2003) or Kantha et al. (PEPS, 4:19, 2017). As the difference can be important, especially at high wind speeds, the impact on $\varepsilon$ should be far from negligible. As a corollary, the longest wavelength of velocity fluctuation is not given by the sole horizontal velocity of the aircraft, but by the relative velocity (lines 348-350).

Because the relative air wind speed may significantly vary with altitude (which is why it should be quantified and shown), TKE estimated for frequencies $f < 5Hz$ and shown in Figure 9 is likely not correct because it indicates the total energy for variable wavenumber bands. In addition, $k_1$ as defined in (9) is NOT the component of the wavenumber vector in the direction of the flight path (line 362), but along the direction of the relative wind vector formed by the direction of flight path and horizontal wind vector. There is no difference between the two, only when the UAV flies in the direction of the wind. Therefore, TKE dissipation rates (and TKE) must be recalculated.

Thank you for raising this concern. Our initial approach was selected in a misguided attempt to minimize the effect of bias introduced by the dependence of Taylor's hypothesis on the wavenumber dependence of the wind velocity used for its application (discussed by Moin, JFM, 2009). Our experience in the ABL has indicated that neglecting the advection due to mean wind does not impact the resulting spectra, however these measurements are typically made in winds an order of magnitude smaller than the UAS ground speed. As pointed out by the reviewer, this is not necessarily the case for the measurements reported here. We therefore have revised the calculation of the longitudinal wavenumber spectrum in the current version of the manuscript to account for the advection due to mean wind. This includes adjusting the estimate of wavenumber from aircraft relative air speed and using the relative velocity component of the wind when calculating the spectra.

As for the frequency cutoff used for integration and its dependence on the relative velocity of the aircraft to the air, we note that our application of this cutoff in the frequency domain is directly analogous to any measurements made with any sensor having finite frequency response (i.e. any sensor measuring in time). However, to address the reviewer's concerns we have compared the TKE and Reynolds stresses calculated with a finite frequency cutoff and a finite wavenumber cutoff and found no difference. This is because the contribution to overall variance at the higher frequency/wavenumber ranges measured by the sensor is minimal so small differences in integration ranges at the high end have no difference.

What did make a difference is that, as noted above, we have modified our statistical approach to use 1500 m long ensembles rather than 30 s long ensembles. This change impacted the low wavenumber bound of the variance calculation and this change did modify many of the statistics (most notably TKE, as could be expected).

(3) Section 3.4

The method used to reconstruct the distributions of parameters in the $\alpha - z$ plane is not clear. The authors should very clearly explain how the interpolation method works as this is not a standard method of visualizing the data. However, it does not seem to be feasible. Because only one value is obtained for a given altitude, it is not possible to interpolate the distribution of a parameter for any value of $\alpha$ at this altitude. The method will systematically produce artifacts (isolated structures with vertical bands) unless the layer probed by the sUAS has a thickness at least greater than the vertical distance covered by the instrument to make 360°. It turns out that all the plots and discussions are incorrect, and section 3.4 should be deleted in its entirety, unless the authors can demonstrate the merits of their approach.

When preparing Section 3.4 we were also concerned with the potential for bias due to interpolation and had ensured that there was at least one measurement point per interpolation grid cell. However, when revisiting these figures during the revision we found that the numerical interpolation scheme (which was an implementation of Delaunay triangulation) was creating cell values very different from the measurement point within the cell. Therefore the reviewer's concerns were well-founded.

As Section 3.4 interpreted these figures in the context of $\langle \phi \rangle'$, and this approach used the interpolated data for background subtraction, we have removed this approach from the manuscript completely rewritten this section (which is now Section 3.5 in the revised manuscript). We still believe that there is merit in this visualization approach, however, since it does show interrelationship between statistical values determined at measurement points at the same azimuthal locations along the orbit. In the updated section, the contours are determined by direct triangulation between points and we have included the actual measurement locations and values in the revised figures to allow the reader to directly assess the impact of the triangulation on the contours.

Specific comments:

Line 8: see comment (1) above.

We have updated the abstract to be more precise

Line 34-41: Some important references on in-situ measurements of turbulence are missing, e.g., Barat and Bertin (JAS, 41, 819-827, 1984, and references therein), Bertin et al. (Radio Science, 32, 791-804, 1997), Alisse

and Sidi (JFM, 402, 137-162, 2000), Gavrilov et al. (Ann. Geophys., 23, 2401–2413 2005). In addition, the manuscript ignores references from the radar literature (e.g. Sato and Woodman, JAS, 39, 2546-2552, Fukao et al., JGR, 1994, and many others). UHF or VHF clear air radars have enlightened the layering structure of the stratosphere, mentioned line 37, to be attributed to thin and horizontally extended turbulent layers or/and stable layers. Incidentally, the authors recognize that the horizontal stratification is a key feature of the stratosphere. Is this feature consistent with the "vertical structures" supposed to have been detected with the sUAS measurements in section 3.4?

We have updated the literature review with these references

Line 65-67: The reference list about turbulence measurements from sUAS must be thoroughly revised. Some of them are not about turbulence (Bäfuss et al., 2018; Rautenberg et al., 2018, Jacob et al., 2018) and many others are missing. For example: Lawrence and Balsley (JTECH, 30, 2352-2366, 2013), Balsley et al., (BLM, 147, 165–178, 2013), Balsley et al. (JTECH, 35, 619-642, 2018), Reuder et al. (Acta Geophysica, 60, 1454-1473, 2012), Shelekhov et al., (Atmos. Ocean. Phys., 57, 533-545), Kanthe et al., (PEPS, 4:19, 2017), Luce et al., (JAS, 77, 231-2326, 2020), Calmer et al., (AMT, 11, 2583-2399, 2018), among others.

We have updated the text but note that Barfuss 2018, Rautenberg 2018 and Jacob 2018 do discuss turbulence measurements (although their focus may not have been on detailed analysis of the measured statistics) and Balsley 2013 was already cited. We have added most of the additional suggested citations.

Line 86-87: "However, due to the transient nature of their Lagrangian flight trajectory, balloon-based approaches are not necessarily amenable to obtaining detailed statistical descriptions of turbulence at high altitudes." The comment is unclear. What do the authors mean?

Our intent was to point out that by advecting with the wind, statistics like horizontal spectra and structure functions become more difficult to calculate. We have revised the text accordingly.

Line 91-93: The introduction of the "infrasonic microphone" has already been made in the previous paragraph. It is this redundant. In general, the various paragraphs of the manuscript should be better organized to avoid such redundancies (they occur several times). This gives the impression of a juxtaposition of paragraphs with no guiding line.

This paragraph is introducing our specific experiments, and therefore we are specifically indicating that such a microphone was used in the present experiments. The previous paragraph was a general review of previous studies using infrasonic microphones. It is not clear to us what redundancy is being referred to here. We have made some adjustments to the text to avoid further confusion with the readers.

Line 121: Please convert km/h into m/s. The controllability of the sUAS is an important parameter and more information about limitations and performance should be given. We understand that the UAV can safely fly for wind conditions up to 31 m/s at least. How is the horizontal velocity of the glider controlled? It must be significantly high than 31 m/s for stability. A figure showing the ground speed of the sUAS with altitude (and the relative air speed, for the reason described in (2) Major comments) would be informative.

We have added airspeed/groundspeed figures to the revised manuscript and changed the km/h to m/s.

We have also added text describing how, the autopilot was set to maintain kinetic energy, and typically set near the optimal lift over drag ratio (the maximum distance that can be travelled per loss in altitude). To maintain the set airspeed the autopilot adjusts the pitch angle (the angle of attack of the mainwing airfoil). The horizontal velocity is a resultant of setting the airspeed and may fluctuate slightly from the pitch angle adjustments. Also, as the aircraft descends in altitude the air density increases and the HiDRON's aerodynamic performance improves; thus, the horizontal velocity gradually decreased as the aircraft descended.

Line 129: please explain why the sampling is made at 10 Hz.

The HiDRON utilizes a radio modem for the command-and-control link and telemetry. The radio communication is also employed to send flight parameters to the ground station where the flight data is logged. Based

on the modem model (Microhard P400), 10 Hz telemetry is selected for maximum efficiency to transmit and record data packets and to maintain reliability of the radio communication and command-and-control link.

The text was updated with this additional description.

Line 192: The authors seem to indicate that the five-hole probe sensor had an effective time response of 0.1 s. The reason is unclear (but I do not have the background to understand). The corresponding spectra should show a gap from ~ 10 Hz. It is roughly observed in Figure 8 at z=10 km (but around 5 Hz) and there is no evidence of a transition at z=1 and 18 km. How do the authors interpret this feature?

These sensors experience viscous damping in the tubing, as well as resonance in the transducer cavity which impact their frequency response characteristics. Therefore each probe has slightly different response characteristics and this particular sensor response was measured directly by introducing a step change in pressure and measuring the settling time of the tubing/transducer system by sampling the data at high rates (30kHz) during the step change.

We do not expect a gap in the spectra as the effect of the tubing is to damp out the response of the probe to fluctuations and this effect is countered by a resonance in the transducer cavity at higher frequencies (100 Hz in this case). The net combination tends to counteract each other and make it difficult to identify where in the spectrum inaccuracies are introduced by probe's response characteristics without measuring the response.

Line 227: The reason(s) of the choice of large values in circle radius (1-5 km) is not explained.

The 5 km radius was selected as a comprise between optimizing aerodynamic efficiency of the aircraft in a turn - by minimizing the bank angle and for safety to stay in proximity to the landing runway. For the 5 km radius turn the bank angle was approximately 5 to 7 degrees, and increasing the turn radius further would provide only a slight change in the bank angle. We have revised the manuscript to include this description.

Line 229: A figure showing the descent rates of the sUAS with altitude would be useful to figure out the conditions of sampling.

We have added this figure.

Line 243: "horizontal distances": with respect to the ground? If yes, it means the ground speed of the sUAS varied between 10 m/s and 80 m/s but this information is not provided in the manuscript. If these distances are expected to characterize the largest scales sampled during 30 sec by the instrument, it is not correct, because the relative speed should be considered (see (2) of Major comments).

As noted, we have updated the segmentation of the time series to be spatially equivalent to 1500 m using the relative air speed at the start of each segment. We have also added figures showing relative velocity and ground velocity as a function of altitude.

Line 252-271: These two paragraphs must be re-written. They are particularly confusing and not rigorous. For example:

- the first paragraph seems to describe general properties for the 3 flights but the second paragraph focuses on flights 2 and 3. So, we deduce that the description made in the first paragraph is for Flight 1. In addition, almost all the statements are disputable or not well- introduced. The agreement within 10% is unclear (please add information/figure that corroborates this result) and it is not true for ¡U¿ at all altitudes for example. But the paragraph focuses on temperature profiles. So, does this quantification only apply to temperature? But then why introduce the paragraph with "with the exception of RH"...?

We have completely revised these paragraphs as we have added additional nearby radiosonde profiles to better illustrate how the aircraft measurements compare to the range of values measured by the NWS radiosondes.

- The altitude of the top of the boundary layer is estimated to be "roughly lower than 3 km" in the first paragraph, but up to ~5 km in Flight 3. First, these altitudes (especially 5 km) are not realistic even for convective boundary layer. Second, the criteria used to estimate this altitude are not explained. Third, there is no indication consistent with these estimates in the figures.

These values are likely an oversight, being remnants from early drafts where we were using altitude in MSL rather than $z$, which we define as AGL which would result in a 1400 m difference. We had also only used the elevated TKE as a rough indicator of the PBL height. In the revised manuscript we have used values determined by more rigorous methods to estimate the PBL height and updated the text accordingly.

- The figures are not correctly labeled: "Fig. 5(a,c,e,)" should be "Fig 5 (a,b,c)". 'Fig. 5c' should be 'Fig 5b'. Fig.5 b, d, f should Fig 5 d.e.f. and at other places. Please check.

Thank you for catching this. Once again, this is due to a change made in the figure organization while drafting the manuscript but failed to update the corresponding text. We have corrected the figure referencing accordingly.

- What is the criterion used to define the tropopause altitude? It is found at 11 km (line 258) (presumably for Flight 1 in the first paragraph), quite consistent with figure 5a, but indicated to be at 12.5 km on line 278. It is found at 13 km for Flight 2 while the temperature inversion is actually observed at 11.5 km in figure 5b. It is stated that it is at 14 km for Flight 13, but an inversion can be found at 12 km. How are quantified the lapse rates in the troposphere and stratosphere and what is the interest to estimate such values (and tropopause altitudes) if they are not compared between the instruments? This comment also applies to humidity and velocity profiles, since the text does not describe the differences and similarities between the profiles, but rather their characteristics, which is another objective. There are too many caveats.

Again, these values are likely remnants from early drafts where we were using altitude in MSL rather than $z$, which we define as AGL which would result in the observed 1400 m difference (rounded to 1500 m). We have updated the text and, as noted above, including additional radiosonde measurements and try to utilize these figures more carefully.

From lines 290: This part should be separated from the previous ones because it is not about comparisons between radiosonde- and sUAS-derived profiles anymore. It seems to me important to show $N^2$ (squared BV frequency) and shear profiles before describing Ri profiles, since one of the purposes of the manuscript is to assess the performance of the sUAS measurements. In addition, low Ri values can have different causes, i.e. a strong shear and/or low $N^2$. The knowledge of these two parameters can help the interpretation of the turbulent events.

We have moved the *Ri* discussion to a new section and added corresponding profiles of $N^2$ and $S^2$ which allow better interpretation of the static conditions and wind shear present during each flight.

Equation (7): In practice, the impact of the variation of g with altitude can be ignored. The error is much less than all the other uncertainties.

We have reverted to $g = const$, in accordance with common practice.

Line 301: "Here we assume the critical Richardson number takes on a value somewhere in the range 0.25¡Ri¡1". It is unclear. Ric=0.25 is a necessary condition below which air can become dynamically unstable and turbulent (if Ric¿0.25, a shear instability cannot develop). Once turbulent, there is a critical Richardson number at which the flow begins to laminarize: it is generally accepted to be between 0.2 and 1 but turbulence can be found for *Ri* >> 1 according to Galperin et al. (2007) (but the corresponding turbulent regime should strongly differ from the turbulent regime for small Ri values). In practice, these thresholds must be used with caution because the Richardson numbers estimated from in- situ data are scale-dependent, i.e. depend on the vertical resolution at which they are calculated. It is common to apply arbitrary thresholds (i.e. "Ri is minimum and small 0.25-1")

This was our intent with specifying critical Richardson number as lying within a range and broadly defining

this as a range of possible values below which turbulence could develop. Note that the revised $Ri$ calculation approach has significantly increased the measured values above this critical range and therefore much of the discussion of critical $Ri$ was edited out as it is no longer relevant.

Figure 8: the information is interesting. Why not showing the corresponding spectra for v and w? In Figure 8b and 8c, it should be $F$ instead of $\Phi$ (y label).

The only reason $v$ and $w$ weren't included was simply to maintain clarity of the figure. However, in the revised manuscript we have decided to present the spectra in wavenumber domain, rather than the frequency domain, as it better fit with the discussion of inertial subrange slope at which they are introduced.

Line 333: I am again skeptical about the interpretation of the increased TKE layer up to 4 km as corresponding to the CBL top

As noted above we have applied a more rigorous definition of the boundary layer height in the revised manuscript, we have also made sure to bette distinguish being near the boundary layer to being within the boundary layer.

Figure 9: EDR and TKE should be presented in logarithm scale (and continuous lines), because the linear scale over-represents the maxima near the ground. Indicating that "TKE is close to 0", line 333, is symptomatic of the fact that the linear scale is unsuitable for the present purpose. Figures showing the slopes, as calculated in Figure 8 for 3 cases, vs altitude and for the 3 wind components should be included. It would enable us to identify the altitudes where the inertial slope is indeed observed, those where a different regime is observed, and those where instrumental noise is dominant for all frequencies. This information is essential for the purpose of the manuscript. A characterization of the slopes vs other parameters (e.g. TKE, EDR, Ri, etc) could be very informative.

As noted above, we have changed our presentation of all figures to use lines instead of data points. We have also changed the profiles of $TKE$, $EDR$, and $Ri$ to use logarithmic axes. Finally, we present the distribution of measured slope in Section 3.5.

The dataset offers the possibility to show $\langle u'^2 \rangle$, $\langle v'^2 \rangle$ and $\langle w'^2 \rangle$ separately. Plotting, for example, $\langle u'^2 \rangle$ vs $\langle w'^2 \rangle$ would be interesting for quantifying anisotropy. A discussion of this anisotropy in light of $\varepsilon$, Ri, etc, would be very enlightening.

TKE and $\varepsilon$ are related by a master length scale (e.g. Mellor and Yamada, Rev. Geophys. Space Phys., 20, 851-8751982) which of great interest for the characterization of turbulence. The dataset shown in Figure 9 offers the potential to estimate this scale.

We agree with both these sentiments, and following submission of this manuscript have continued examining these and other aspects of the data. However as the current draft of the paper is already over 30 pages, we feel that an in depth statistical analysis of the data set is beyond the scope of this initial paper (intended to describe the measurement system, measurement approach and capabilities.)

Line 337: "...caused by inertial turbulence" and remove "elevated" in the same sentence because the comment is valid for all levels of TKE. The spectra with a -1 slope may either reveal another turbulent regime or be due to a white noise contamination even for $f < 5Hz$ when the atmospheric signal is weak. Figure 8c is apparently in favor of the first interpretation for the selected case but it is not necessarily always true, especially when the instrumental noise dominates.

Our expectation is that the instrumentation noise would be indicated by a noise floor on the spectra which does not appear until much higher frequencies than those presented. As the measured spectra are consistently above this noise floor (with content over an order of magnitude higher), we do not expect this deviation to be due to white noise contamination (at least from instrumentation noise) and instead expect it to be due to a different turbulent regime.

Line 343: "... more active turbulence conditions during these flights". This statement should be nuanced because (1) the rejection was based on the u spectra only, (2) the non- detection of a -5/3 slope does not mean the absence of turbulence, (3) the corresponding levels of EDR of flight 1 (qualified as "weakly active") shows a significantly higher background than flight 2, indicating higher spectral levels but not consistent with an inertial subrange. As we do not know about the interpretation of the observed non-inertial subranges, turbulence activity cannot be qualified.

*This was only meant to broadly refer to the increased scatter in the TKE profiles. We have revised the text to be more clear.*

Line 345: Please remove this sentence. The reference of Kelvin waves is not suited here because equatorial Kelvin waves (Fujiwara et al. 2003) are waves trapped around the Equator similarly to coastally-trapped Kelvin waves.

*We have revised as suggested*

Equation (9): The notation $\Phi_{11}$ of the spectrum may not be appropriate ($F_{11}(k_1)$ would fit better the notation used in Figure 8). $\Phi_{ij}$ generally refers to the spectral density tensor (see e.g. Doviak and Zrnic', Doppler radar and weather observations, p. 326, 1984). The sUAS is "sensitive" to the 1-D longitudinal spectrum (see e.g. Hocking (EPS, 1999)).

*We intentionally defined $F$ to indicate velocity spectra defined in the frequency domain and $\Phi$ to indicate velocity spectra in the wavenumber domain (different notation is required as $\Phi \neq F$ due to the conversion required to ensure variance is preserved when both integrating $F$ in $f$ and when integrating $\Phi$ in $\kappa$). Note that the turbulent velocity spectrum tensor is also commonly referred to as $\Phi_{ij}$ in the turbulence literature (see textbooks by Pope or Tennekes and Lumley, for example) which is why we used it here. However, to avoid conflicting with established nomenclature, we have replaced usage of $\Phi(\kappa)$ with $E(\kappa)$.*

Line 367 refers to $\Phi_{11}(k_1)$ ($\Phi_{11}$?) and $F$ ($F_{11}$?). Line 368: $k_1^n$ should be $k_1^{-5/3}$. Strictly, the power-law fitting should be applied to a limited range of $k_1$ since the smallest wavenumbers are not well-resolved.

*Line 367 was describing the conversion described above and therefore required both $\Phi_{11}(\kappa_1)$ and $F_{11}(f)$. Line 368 wass specifically describing the power law fit (which does not pre-assume $n = -5/3$, as we are using it to evaluate how closely the inertial subrange slope, $n$ is to -5/3, and we also specifically note that the wavenumber range is limited to that where the probe response is reliable. For the 30 s average we found that a lower frequency bound was not necessary for the fit, however the larger statistical windows used in the revised calculations required implementation of a low wavenumber bound of $\kappa_\ell > 0.1 \text{ m}^{-1}$.*

Technical comments:

Line 4-5: The sentence is unclear, please rephrase.

*This sentence has been revised.*

Line 20-21: please add references.

*Citations added.*

Line 24: "Despite the higher stability of the stratosphere"

*This sentence has been revised.*

Line 25, see also line 318: "... due to mechanical and thermal disturbances" -¿ "due to shear instabilities and gravity wave breaking". The formulation is unsuitable because the mechanical sources of turbulence refer to those produced by obstacles close to the ground. The rest of the paragraph is awkward –and not rigorous- and references of the "classical" literature should be included instead.

This sentence has been revised.

Line 38: "high turbulent kinetic energy dissipation rate": please be more quantitative.

Revised with the addition of a quantity.

Line 45: "and for identifying the inner scale of turbulence."

Our preference here is to retain finer scales (as referencing the fine scale structure of turbulence, as used by Townsend for example) as opposed to 'inner scale' since inner scales have very precise definitions in boundary layer turbulence which are not appropriate here). We also considered using the term microscale structure, consistent with how Kolmogorov scales are typically described, but were concerned that this may conflict with the 'microscale' spatial and temporal scales used in meteorology.

Line 45: "This experiment": please be more specific with references.

This sentence has been revised.

Line 45-47: The Richardson number is not defined and it is not explained why Ri=0.25 is an important value and why turbulence when Ri ¿ 0.25 should be noted. The authors should indicate that some LITOS results were corrupted (Soder et al., AMT, 2019) due to balloon wake and that turbulence observed when Ri ¿¿ 1 was suspect. In addition, useful information on the relationship between Ri and $\varepsilon$ can be found in earlier references mentioned in the specific comments (line 34-41).

We have moved the mathematical definitions of $Ri$, $\varepsilon$, and $N^2$ into the introduction and added some text introducing the critical Richardson number.

We have als added the Soder et al reference to the discussion of the LITOS experiment.

Line 63-72. Pitot tubes are also used (e.g. Lawrence and Balsley (JTECH, 30, 2352-2366, 2013).

This sentence has been revised to include Pitot tubes.

Line 90: "air masses" generally refer to "large bodies of air" at synoptic scales in meteorology. The term is not suitable here.

Replaced with 'turbulent eddies' which more aptly describe our intended meaning.

Line 90: "geostationary" usually refers to satellite orbits. Do the authors mean "relatively constant location above the ground?"

That is indeed what we meant, we have revised the text as suggested

Line 92: ""traditional" → "standard"

Revised accordingly

Paragraph 2.1: It seems more natural to present the instruments first, then the configuration of the experiment. 2.1 → after 2.3.4 and before 2.4.

Although we felt that 2.1 served as an introduction to the overall experiment, we have moved the section as requested.

Fig.1 : Please add the location of the balloon launch site (El Paso) and show the distance in km (in Fig. 4 also). The distances are crucial for the interpretation of the radiosonde and sUAS data and longitude/latitude

coordinate system is of little use here.

The balloon launch site was well outside both these figures and would not be suitable to include. Instead, we have added an Appendix with upper air wind maps and satellite imagery on which the radiosonde launch sites were indicated

Line 102: "Three sUAVs were flown" (?)

Three flights were conducted with the same sUAS. We have updated this sentence to be more clear.

Line 106: The altitudes are given in km m.s.l but the profiles are shown from z=0 (i.e. above the ground (line 241). Please indicate the corresponding altitude AGL, even if it can be roughly deduced from Fig. 1.

We have added the launch/recovery altitude to the experiment description and added the a.g.l. release altitudes to Line 106.

The third sUAS was released from 30 km m.s.l ( 28.5 km AGL ?), but the profiles are shown up to 25 km. Please clarify.

Although the release was at 30 km, it took about 3 km of altitude before the aircraft returned to its controlled orbit (See figure 4c). For consistency, we only report data only from the controlled orbital descent phase of the flight.

Line 191: "... the actual probe frequency response..."

Revised

Line 194: remove "disconnected"

Revised

Lines 232-235: Please indicate Local Time instead UTC (and avoid MDT). By doing so, the reader does not need to convert by himself when interpreting the PTU profiles measured by the sUAS and the radiosondes at different times (Figure 5).

We have provided launch/release/recovery times in UTC and LT in the revised manuscript.

Line 246-249: The first sentence is not necessary and the second has already been written. A more detailed description of the radiosonde data is necessary (see (1) of major comments).

We have revised the text accordingly.

Figure 5: please add LT times for all the radiosonde and sUAS flights. A figure showing the trajectories of (and horizontal distance between) both instruments is necessary.

We have added the LT as requested, however a figure with trajectories is not feasible due to the distances involved (glider orbits were only 10km diameter). Instead when introducing the radiosonde data, we clearly indicate the separation distances and involved

Line 258: For flight 1, "the temperature continued to decrease with altitude at a rate of 1C/km" → "the temperature continued to decrease at a mean rate of 1C/km between 11 and 19 km" (otherwise it is confusing, see specific comments also)

Revised.

Line 274: please add "(not shown)".

Revised.

Line 281: please show the NOAA upper air wind maps. The absence of reference points make difficult to confirm the statements.

We have added the NOAA upper air wind maps to Appendix A.

Line 282-285: what do the authors mean ?

We simply were explaining why the jet stream was no longer evident during Flights 2 and 3. We have updated the text to improve clarity.

Line 307: The term "potential instability" refers to "an atmospheric condition in which otherwise stable air would become unstable if forced to rise (e.g. over high ground) thereby reaching its saturation point." See e.g. www.encyclopedia.com. Please replace "potential" by "possible shear".

This sentence has been revised. We had only meant to indicate that there was potential for an instability to develop.

Line 309: "marginally unstable tropopause": what do the authors mean?

We were referring to the lapse rate classifications for the troposphere. We inadvertently replaced troposphere with tropopause. This typo has been corrected and we have added a reference for the lapse rate classifications.

Line 316: Here again, the terminology is improperly used. "An atmosphere is said to be "conditionally unstable" if the environmental lapse rate is between the moist and dry adiabatic lapse rates. This means that the buoyancy (the ability of an air parcel to rise) of an air parcel depends on whether or not it is saturated." (see glossary of meteorology) . "suggesting the possibility of localized buoyant production". Do the authors refer to statically unstable conditions, ie. Ri ¡0? If yes, it must be clearly stated and defined earlier, e.g. around line 302.

This sentence was originally written in a slightly confusing manner. The conditions for buoyant production in the boundary layer were referring to the observations $Ri < 0$ in the boundary layer. The reference to the 'conditionally unstable troposphere' was referring specifically to the measured lapse rate, although it was not clear in the sentence as written. The revised $Ri$ and $N^2$ calculation approach reflects statically stable conditions were present throughout the altitudes measured (except the boundary layer) and therefore this sentence was no longer relevant.

Line 318: "mechanical turbulence": see above, comment for line 25

This sentence has been revised.

Line 328: Do the Hanning window preserve variance?

Yes. This was confirmed prior to use.

Line 329: Do the cut-off at 5 Hz related to the effective limited time response indicated line 192?

Yes. We cannot have confidence in the frequency content measured above 20 Hz and the initial 5 Hz threshold was selected to provide additional confidence in the range used. However, to increase the data points included to the calculation we have revised the integration range to include frequency content up to 20 Hz.

Line 334-335:The description is unclear because of the use of a linear scale and a dot representation (see "specific comments")

We have updated all the figures to use lines and logarithmic axes where appropriate.

Line 336: Please remove "although the regions of elevated k appear at different altitudes". This comment is not useful.

Revised.

Line 344: Please indicate the altitude of the tropopause in Figure 9.

The logarithmic presentation of the updated figures obscures the previous observation of an increase in $k$ and $EDR$ near the tropopause, therefore we have removed this statement and feel adding an indicator of the tropopause location to the figure unnecessary.

Line 354: "As direct measurements of . . . " please explain more or add a reference

We have revised the text to expand our explanation

Line 355-356: The sentence indicates a condition that has no reason to exist at this stage, since inertial domains have been identified by spectral analysis.

Please note that we are applying this calculation to all 1500 m long segments within the time series to determine $\varepsilon$, hence we need to formally assume equation 9 is at least approximately valid in order to obtain an estimate of $\varepsilon$ for each segment, even if no inertial subrange is evident.

Line 520: Please remove "flux". The gradient Richardson number and the flux. Richardson number have two distinct definitions.

We have corrected this typo and are aware of the distinction.

---

## Author Comment (AC3)

**Response to Referee 3**

We appreciate the time taken by all referees in providing insightful and detailed comments about our manuscript. Following the reviewers recommendations, numerous changes have been made in the analysis approach including: (1) Revisiting the conversion of the five-hole-probe wind estimate; (2) adjusting the statistical ensemble sized from being based on 30 s ensembles in time to 1500 m ensembles in space (with 750 m overlap between ensembles to retain spatial resolution); (3) revised the spectra calculation with the correct transformation from frequency to time domain; (4) updated the method used to calculate vertical gradients and adding Brunt-Väisälä and shear frequency measurements to the paper; and (5) addressing issues in the methodology used to generate contour plots. Of these changes, the largest impact on the results was the change made to the vertical gradient calculation, which impacted the gradient Richardson number, $Ri$, values and the revision to the contour plots. As a result of these changes, and other additional changes made to address specific comments made by the reviewers, we believe the revised version of the manuscript is more clear and complete than the version originally submitted. To help clarify where changes have been made in the revised version, we have highlighted all changes made using blue text.

Below, we respond to the individual comments made by the referee. To do so, we have reproduced the original review, with our comments provided in blue text.

**Reviewer(s)' Comments to Author:**
This paper presents some intriguing results using a new measurement platform for profiling the atmospheric column descending from about 20km. However, the quality of the observed data is not clear, given the periodic variations that may be a result of the periodic orbit of the gliding aircraft platform. As a result, the conclusions drawn about the viability of the sensing method and the relation to potential atmospheric structures and sources is tenuous, without furhter examination of the correlations between signal variations and platform motions.

As we were also concerned with apparent wind magnitude variations with altitude having a vertical wavelength similar to the altitude difference between successive aircraft orbits, we have thoroughly revisited the wind measurements and found several places where improvements could be implemented, including improving the time alignment between autopilot kinematic variables and payload sensors, identifying and correcting pitch and yaw probe misalignment (of less than $10°$) between aircraft and sensor coordinates, and discovering an error in probe rotation. However, these improvements only marginally modified the wind magnitude and did not affect the observed vertical profile in any meaningful manner.

We have also closely examined the dependence of the vertical profiles of wind magnitude with heading, as shown in Figure 1. The most notable similarity between vertical separation of orbits and vertical wavelengths in wind magnitude occurred during Flight 1. However examining successive orbits shows that the wavelengths are not identical, with the orbit vertical distance slightly longer. Therefore, if there is a bias in the measurement, it is not rigidly correlated to the heading.

We also perturbed different inputs into the wind estimation, and found that the vertical wavelengths in the wind estimate appears to be robust to these changes. In summary, we could find no conclusive evidence of bias in the measurements introduced by the aircraft heading and the periodic orbit.

Detailed comments:

Introduction: "The trajectory of the glider allowed for improved statistical convergence and higher spatial resolution of derived statistics measured by the in-situ sensors." Refers to balloon-borne measurements, but such improvements and higher spatial resolution were not demonstrated in the paper.

Here we were referring to the airspeed of the glider able to transect the flow at flight speeds over 20 m/s which, when compared to balloon-borne measurements, means that more wavelengths of turbulence can be measured over the same duration of sampling time. However, spatial resolution has different connotations and is not the correct term for what we are trying to describe. We have revised the text to better reflect our intended meaning.

Similarly: "which allowed the connection to be made between the locations of increased turbulence intensity and the source of its generation" was tenuous, only to the level of "consistent with".

[Figure]

Figure 1: Vertical profiles of wind magnitude from all three flights shown with horizontal lines indicating location where aircraft heading passed through 180°. (left) Flight 1 (middle) Flight 2 (right) Flight 3

This is a fair assessment, we have revised the text accordingly.

40: "with these results used to model the relationship between turbulence in the stratosphere as well as tropospheric activity" not clear: "as well as" vs. "and"?

Revised. 'and' should have been used.

85 "However, due to the transient nature of their Lagrangian flight trajectory, balloon-based approaches are not necessarily amenable to obtaining detailed statistical descriptions of turbulence at high altitudes." Why? Aircraft are also transient, and if GPS guided, only see what is advected past. Balloons with altitude profiling are not Lagrangian vertically, so also sample more than one parcel of air. The statement is vague: it depends what statistics are being evaluated.

True, and addresses the same point as the first detailed comment. We have revised the statement to be less vague.

90: "A glider offers advantages over traditional balloon launches by being able to maximize time at altitude during its descent phase" Vertical rates for the glider vary from 5m/s to 1 m/s, very similar to descending balloons. "These qualities facilitate the statistical analysis necessary for quantification of non-stationary properties" is not supported by evidence in the paper.

We respectfully, but strongly, disagree on this point. Note that in 1000 m of altitude change, for the current experiments the orbit of the glider means that it samples approximately 15,000 m along its flight path, whereas a balloon will sample only the 1000 m. This is a significant difference in the amount of atmosphere and range of eddy sizes that are sampled over the same vertical distance. Note also that the current configuration of the aircraft means that the turbulent eddies are acquired at an order of magnitude higher temporal resolution as well. It therefore would be very difficult to reproduce the spectra of Figure 11 (and corresponding $k$ and $\varepsilon$ estimates), and azimuthal distributions of these statistical properties (as shown in Fig. 14) with a balloon, particularly over the wavenumber range and at the vertical resolution that the glider can measure.

We have added the above discussion to the conclusions to ensure that these points are not overlooked by other readers.

Difficulty of conducting UAS measurements of this type in the NAS was not discussed, nor the conditions under which the reported flights were allowed. Was this in restricted airspace? Under who's auspices? Or was this in the NAS under a COA?

The reviewer raises a good point as the current regulatory environment prevents these types of measurements from being routinely conducted. In the current experiment, flights were conducted in restricted airspace managed by the SpacePort America facility and coordinated with the nearby White Sands Missile range. We have added revisions to the manuscript to include these points.

125: The iMET sensor specifications were not referenced. These accuracies and time constants tend to degrade at lower pressures and temperatures, and this was not indicated.

We have added a citation to the manufacturer's webpage where this information was obtained. Note that these sensors were previously flown on a similar platform and found to be consistent with model results (Schuyler TJ, Gohari SMI, Pundsack G, Berchoff D, Guzman MI. Using a Balloon-Launched Unmanned Glider to Validate Real-Time WRF Modeling. Sensors. 2019; 19(8):1914. https://doi.org/10.3390/s19081914).

Figure 3 would benefit from the addition of dimensions to the components pictured.

Dimensions have been added to the figures

150: "Comparison of calibrations with and without heating active indicated that there was no influence of probe heating on the five-hole-probe response characteristics." Not clear what response characteristics means: time constant? Calibration coefficients? Noise level?

We are referring to the calibration surfaces and have revised the text to be more precise.

153: "Each hole on the probe was connected to differential pressure transducers through 1.75 mm diameter flexible polymer tubing." What was the other port of the differential pressure sensor connected to? Presumably this was the "static pressure port", but this was not shown or described in the paper. How long was the tubing (this can have a detrimental effect frequency response of the air speed measurement, as noted later in the paper).

We have added these details to the text.

160: "Note that the during flight, the autopilot maintained flight speeds sufficient to produce pressure differences well within the range of the low-sensitivity transducers and hence only the readings from these sensors were used for this analysis." Please quantify the airspeeds obtained, and the corresponding average differential pressures.

We have added figures showing relative air velocities and noted the value of dynamic pressure in the text.

195: how was aircraft sideslip angle determined? How did the use of this affect the quality of the horizontal wind measurements?

While revisiting the wind measurement procedures, it was found that the probe was actually rotated 90 degrees relative to what the authors initially thought. This meant that it was actually the pitch holes that were disconnected for flights 2 and 3 and not that yaw holes. We also compared flight 1 data with and without the revisions required to calculate winds for flight 2 and found that the differences were negligible. The text has been revised the text accordingly.

Generally, the details of this particular mutli-hole probe and its calibration and resulting accuracy were not provided. Can these be referenced from an earlier publication?

The probe used here is derived directly from the probes used in :

Barbieri, L. and Kral, S. T. and Bailey, S.C.C. and Frazier, A.E. and Jacob, J.D. and Reuder, J. and Brus, D. and Chilson, P.B. and Crick, C. and Detweiler, C. and others (2019) "Intercomparison of small unmanned aircraft system (sUAS) measurements for atmospheric science during the LAPSE-RATE campaign," *Sensors* 19(9), 2179.

and utilize calibration systems and approaches described in:

Witte, B.M., Singler, R.F. and Bailey, S.C.C. (2017) "Development of an Unmanned Aerial Vehicle for the Measurement of Turbulence in the Atmospheric Boundary Layer," *Atmosphere*, 8(10), 195.

Al-Ghussain, L. and Bailey, S. C. C. (2022) "Uncrewed Aircraft System Measurements of Atmospheric Surface-Layer Structure During Morning Transition," *Boundary Layer Meteorology*, v185, 229-258.

We have added these references to the revised manuscript.

Note, that these probes have also been successfully deployed in previous studies, including:

Bailey, S.C.C., Smith, S. W., Sama, M.P., Al-Ghussain, L. and de Boer, G. (2023) "Shallow katabatic flow in a complex valley: An observational case study leveraging uncrewed aircraft systems," *Boundary Layer Meteorology*, v186, 399–422.

Bailey, S.C.C., Sama, M.P., Canter, C.A., Pampolini, L.F, Lippay, Z.S., Schuyler, T.J., Hamilton, J.D., MacPhee, S.B., Rowe, I.S., Sanders, C.D., Smith, V.G., Vezzi, C.N., Wight, H.M., Hoagg, J.B., Guzman, M.I. and Suzanne Weaver Smith (2020) "University of Kentucky measurements of wind, temperature, pressure and humidity in support of LAPSE-RATE using multisite fixed-wing and rotorcraft unmanned aerial systems," *Earth System Science Data*, 12(3), 1759-1773.

Bailey S.C.C., Canter C.A., Sama M.P., Houston A.L. and Smith S.W. (2019) "Unmanned aerial vehicles reveal the impact of a total solar eclipse on the atmospheric surface layer" *Proceedings of the Royal Society A*, 47520190212.

so they, and their use, are not untested.

205: How was the microphone mounted on the vehicle? Was it protected from dynamic pressure fluctuations? If so, how did this filter the infrasound pressure waves? Could aircraft motions (that are also dependent on ambient turbulence) influence these measurements?

The microphone was mounted rigidly within the nose of the aircraft, with the diaphragm facing forward. Being within the fuselage, the microphone was protected by dynamic pressure fluctuations. Note that infrasonic sound waves will be of the order of 30 m and larger, so attenuation of the sound waves by the fuselage is not expected in this configuration. Due to the rigid mounting of the probe in the aircraft, it is not anticipated that aircraft motion could influence the microphone, however we were not able to verify this assumption from the current set of measurements.

We have added this information to the revised manuscript.

How are winds calculated?

Winds were calculated based on the procedures described in lines 153 to 187 (lines 199 to 231 in the revised manuscript).

How is airspeed calculated? No plots of airspeed were provided. What was the airspeed as a function of altitude?

Airspeed was measured by both the aircraft's Pitot probe and the five-hole probe using the standard procedure of measuring dynamic pressure across total pressure (central hole) and static pressure and found to be in agreement between the two instruments. Plots of relative air velocity have been added to the revised manuscript.

220: temporal alignment can be intricate. How was this accomplished with this data? Was there a common time reference?

Additional clarification added to text. Initially we intended to use some of the statistics calculated by the payload and sent to autopilot via RS232 communication and recorded in the the aircraft telemetry stream, but found correlating the dynamic pressure measured from the aircraft pitot probe and measured from five hole probe to be a more reliable alignment indicator due to suspected buffering delays in the RS232 connection. Note that, although not mentioned in the manuscript for brevity, we were able to identify and remove a 0.005% difference in clocks between the two systems by windowing the correlation of the airspeed and five-hole probe. This corresponds to a 1 second difference in timing over the six hours of measurement.

230: what does "controlled landing" mean here? Manual landing (RC), or automatic landing (autopilot)?

Landings were conducted by the autopilot. This has been noted in the revised manuscript.

229: A portion of the descent from the 30km release seems to be very steep. There were also some very tight circles at isolated points in the first two descents. Why?

These flights were also test flights for the aircraft. During the flight, the operators conducted several tests of their systems which included adjusting the flight profile mid-flight and improving the response of the aircraft following release. Note that the steep release at 30 km is due to the requirement to achieve sufficient dynamic pressure to produce enough lift for controlled flight. The lower density means that the aircraft must fall a certain distance before the aircraft can travel to it's measurement location.

235: I think you mean UTC -6:00 here.

Correct. Revision made.

239: "Due to the configuration of the sensors on the aircraft". Vague. Please describe what about the configuration makes the sensor readings unreliable on ascent.

During ascent, they are in a stagnant region within the wing pillar wake and therefore not sufficiently aspirated to prevent self-heating and delayed air exchange with the environment. The result was that the $T$ and RH measurements had a significant altitude-dependent warm bias of 5 to 15 degrees K relative to descent and the radiosonde measurements. On descent, these sensors are introduced into the oncoming airflow and properly aspirated. We added more details to the revised manuscript.

289: "with backing"?

Revised.

295: central differencing between adjacent 30 sec averaged values?

That is correct, although we have updated the revised manuscript to use spatially regular, rather than temporally regular, segments.

230: regression fit to a constant function over 150 sec? Central 30 sec interval with 2 intervals before and 2 after?

Prior to differencing we employ a 5 point moving average where the smoothed value at the central segment is found by averaging the values averaged within the central segment with average values from the 2 segments before and 2 segments after. We have added more details in the text.

271: "this is likely due to spatial heterogeneity in the atmospheric moisture concentration". Could also be due to an instrumentation anomaly.

It is possible. We have added more radiosonde data to these figures to better represent the spatial heterogeneity. The results suggest there might be some dry bias in the $RH$ sensor in cold temperatures.

The following statement "cloud conditions near Truth or Consequences, NM (near Spaceport America) were different" does not help. Different how? At what altitudes?

The ASOS reports only qualitative conditions (i.e. scattered, clear, overcast, etc.). However, the additional radiosonde profiles made the ASOS reported cloud unnecessary and this text was removed. We also now include satellite imagery in Appendix A, which provides a more nuanced illustration of the different humidity conditions which could be expected.

276: compare well given the spatial offset and the local weather conditions, and if the local periodic variations are ignored. These variations are suspiciously periodic with altitude, raising questions about artifacts from the platform airspeed/attitude/descent rate that may be varying with the same period. (See the related comments about Ri later). Some evidence should be provided that these results are not correlated with aircraft motions.

This was addressed when responding to the initial comment. The additional radiosonde profiles in the revised manuscript also provide increased context for the wind fluctuations observed.

300: Although it probably does not make much difference, z in this formula should be altitude MSL, not AGL. Recommend that MSL be used throughout for consistency and for interpretation of the results. Also, I can't seem to find the altitude of the ground at the launch location.

True, this was a typo as we made our calculation using MSL. However, based off of Reviewer 2's comments, we have removed the altitude dependent $g$ calculation from the revised manuscript. We have also added the m.s.l. altitudes of the launch and recovery location manuscript.

303: The Ri profiles seem to have highly periodic excursions with altitude. Could these be at the same period as the orbits the plane executes on the descent? That is, how do we know this is not an artifact of the sensors or the periodic motion of the platform? It would also be good to see how this correlates with the bank angle of the plane, since this will not be constant in wind. It will be difficult to take the results at face value without careful checking for motion/attitude/descent rate artifacts from the platform. Similarly, the Ri values seem suspiciously low, with < 0.25 values for much of the flight. Are these low values periodic anomalies in the measurements?

We have extensively updated the text addressing this variability and present an improved method to calculate Richardson number. See additional discussion above regarding the dependence of wind velocity on the orbits.

323: how does $< u >$ differ from $< U >$ used earlier?

We use $u$, $v$, and $w$ to indicate the components of the wind vector having a projected horizontal magnitude of $U$. These distinctions were defined on line 183 in the original submission with $U$ specifically defined in equation 3 (now equation 7 in the revised manuscript).

327: what were the subintervals and overlap used in the Welch method? The "spectra" in Figure 8 seem to have a 40s period fundamental frequency, so this is confusing.

We applied the Welch method without using only three subintervals and a 50% overlap. We have also verified that the application of Welch's method does not impact the shape of the resulting velocity spectrum. The 40s fundamental period is due to mislabeling of the figure, as discussed later.

$F_{uu}$ and $Phi_{uu}$ are both used for the power spectral density (please be consistent), and these are incorrectly called the "frequency spectrum".

Please note that we are using colloquial language used in contemporary fundamental turbulence literature whereby the power spectral density is referred to as the velocity spectrum, or more broadly the energy spectrum, or shortened to simply the spectrum depending on context. Furthermore, these "spectra" are often distinguished as to whether their dependence is in the frequency domain or wavenumber domain by referring to them as the "frequency spectrum" or "wavenumber spectrum" respectively (see, for example, Pope (2000) "Turbulent Flows", Cambridge University Press.) We made the distinction in the original manuscript between $F_{uu}(f)$ (i.e. the frequency spectrum) and $\Phi_{uu}(\kappa)$ (i.e. the wavenumber spectrum) as these are separate functions since both $\int_0^\infty F df$ and $\int_0^\infty \Phi d\kappa$ must both return the variance of the velocity component that they are calculated for (e.g. $u$ in the above case). As $\kappa \approx 2\pi f / V_{rel}$ this means that $\Phi \approx F(V_{rel}/2\pi)$. We realize that this distinction is confounded by the fact that the original figure 5 was mislabeled as $\Phi_{uu}$ instead of $F_{uu}$ for two of the subfigures, but this was a typographical error and not intentionally inconsistent. In the revised manuscript we strive to be more clear that we are referring to the velocity spectrum, and try to be more clear with the distinction between $F(f)$ and $\Phi(\kappa)$. Reviewer 2 also did not like the use of $\Phi(\kappa)$, so we have now replaced that with $E(\kappa)$. We also now present the wavenumber spectra instead of the frequency spectra in the revised manuscript as it was more relevant for the discussion of determining the inertial subrange slope.

328: Was the Hanning window variance-preserving?

Yes. We confirmed it is variance preserving prior to implementing it.

329: integration of the PSD is from .025 Hz to 5Hz, so the most important (largest amplitude) components of TKE are potentially not included. This makes TKE difficult to estimate without some idea of the local "outer scale" where the spectral energy ceases to increase as frequency decreases. This is noted later (347), but with a confusing reference to 300m as the longest period in the k "measurement", since earlier in the paper 2420m was quoted as the longest interval in the 30s analysis intervals. However, no mention of the outer scale was made. Thus the k retrieval has a highly variable lower spatial scale with altitude, and the relevance of the k profiles is unclear.

We had used 300 m as it is the average distance travelled during the 30 s sample duration, but acknowledge that the airspeed at high altitude was much higher, leading to the 2420 m length of the longest segments. In the revised manuscript we use a constant statistical segment length. Although this may not fully resolve the largest eddies it should provide a consistent wavenumber range for which TKE is calculated. We did work at trying to implement an approach which spatially varied the segment length depending on the energy content, but due to the range of scales experienced during descent, we were not happy with the implementation and feel it needs further development before including in a publication.

339: shape of the "spectra" in Figure 8 is very strange. Part a) seems to have a noise floor near $10^{(-3)}$, but the noise floor is smaller (near $10^{(-6)}$) at a higher altitude (part b)! (Where pressure fluctuations are necessarily smaller). A noise floor is again seen near $10^{(-3)}$ in part c). Also, the spectral slope is too shallow in part c) as noted later in the paper. Could it be that this power spectral density is the noise figure of the sensor itself, and there is really no detectable signal at these low atmospheric pressures?

As noted above we had mislabeled Fig. 8 due to changing from presenting the results in the wavenumber domain to the frequency domain. The original figures were actually presenting the results in the waveunumber domain and we are unhappy that this error made its way into the final submitted manuscript. We also believe the reviewer is correct that the -1 slope is the noise figure of the sensor (or more accurately the combination of sensors used to determine the wind) and that the -1 slope indicates that there is no measurable turbulence present.

356: Buoyancy Reynolds number can be calculated after estimating epsilon, as a check on this assumption.

This assumption is not expected to be valid over the entire measurement range, and is checked/unvalidated by the power law fit discussed on lines 456-457. However, assuming it's validity is necessary to produce a dissipation rate estimate using this approach. Validity of this assumption is therefore reflected by the filled circles on Figures 9 and 10.

364: the $\kappa_1$ wavenumber component in (9) is the longitudinal component of the motion of the air relative to the sensor. This is only the longitudinal component of the vehicle ground velocity in the special case of zero mean wind (still air), or in the limit when the vehicle airspeed is much larger than the wind speed. This should be corrected to use the airspeed, and the course heading frame rotation should be replaced by one based on angle of attack and sideslip of the sensor relative to the relative wind vector.
366: again use of ground speed here in incorrect. Must be airspeed.

Thank you for raising this concern. Our initial approach was selected in a misguided attempt to minimize the effect of bias introduced by the dependence of Taylor's hypothesis on the wavenumber dependence of the wind velocity used for its application (discussed by Moin, JFM, 2009). Our experience in the ABL has indicated that neglecting the advection due to mean wind does not impact the resulting spectra, however these measurements are typically made in winds an order of magnitude smaller than the UAS ground speed. As pointed out by the reviewer, this is not necessarily the case for the measurements reported here. We therefore have revised the calculation of the longitudinal wavenumber spectrum in the current version of the manuscript to account for the advection due to mean wind. This includes adjusting the estimate of wavenumber from aircraft relative air speed and using the relative velocity component of the wind when calculating the spectra.

367: I don't understand the expression for $\Phi(\kappa_1)$.

This is discussed above and relates to the property that $\int_0^\infty F\,df$ and $\int_0^\infty \Phi\,d\kappa$ must both return the variance of the velocity component that they are calculated. We have updated the text to provide a better indicator.

Figure 9: Suggest plotting k and epsilon (or EDR) on a log scale to look for periodic artifacts (as noted earlier), and to make it easier to see the full range of these power function values.

We have plotted these figures on logarithmic axes in the revised manuscript and find no clear indication of artifacts due to athe periodic orbits.

Not clear how (of if) the noise floor/noise figure is removed in the qualified data fits.

We made no attempt to remove the noise floor from the data fits.

386: "It will be shown later that this enhanced EDR corresponds to measured fluctuations in velocity introduced by the presence of gravity waves at these altitudes". How? Gravity waves have a much large wavelength than could be influencing these epsilon estimates.

Unfortunately, the appearance of gravity waves appears to have been contaminated by the two-dimensional interpolation technique used to in the $\langle T \rangle'$ and $\langle w \rangle'$ analysis used to discern the presence of gravity waves. Although we believe there is still merit in the concept behind the approach, we could not come up with a suitable implementation in time to include in the revision. We have therefore removed the gravity wave discussion from the revised manuscript.

401: confirmation that the infrasound signal is due to turbulence is too strong a conclusion at this stage. Localized increases do not correspond to those in EDR.

This statement was driven by the increase observed as the aircraft enters the boundary layer and becomes immersed in the turbulence (as evident in the comparison of Figure 10 and Fig.12) and also the comparisons in Figures 14-16. However, we agree that this conclusion is weak, and that the infrasonic measurements need more development.

426: the "interesting features" in figures 12-14 show strong correlation with location on the flight path circle. This might be due to differences in structure in the atmosphere across the 5km circle diameter, but it might also be due to sensor signal dependence on the heading or attitude or airspeed of the vehicle, that is also periodic with location on the flight path (as noted above). Given that these "features" in the data persist over large altitude ranges (where e.g. shear and stability are expected to vary significantly), and the intermittent, sometimes contradictory correlations noted in the paper, it is difficult to consider the conclusions offered as more than optimistic interpretations of rather murky relationships. Too much is made of data that has not been thoroughly vetted.

We note that the statistics being presented are calculated over relatively short windows (of a length about 5% of the orbit circumference) and, as many of them require mean subtraction, they should be largely unaffected by any bias which may be introduced by aircraft heading.

When preparing Section 3.4 we were also concerned with the potential for bias due to interpolation and had ensured that there was at least one measurement point per interpolation grid cell. However, when revisiting these figures during the revision we found that the numerical interpolation scheme (which was an implementation of Delaunay triangulation) was creating cell values very different from the measurement point within the cell. Therefore, although not appearing to be due to position in the orbit, the reviewer's concerns were well-founded.

As Section 3.4 interpreted these figures in the context of $\langle \phi \rangle'$, and this approach used the interpolated data for background subtraction, we have removed this approach from the manuscript completely rewritten this section (which is now Section 3.5 in the revised manuscript). We still believe that there is merit in this visualization approach, however, since it does show interrelationship between statistical values determined at measurement points at the same azimuthal locations along the orbit. In the updated section, the contours are determined by direct triangulation between points and we have included the actual measurement locations and values in the revised figures to allow the reader to directly assess the impact of the triangulation on the contours.

449: Ri is used as a marker for stability in various places, which is confusing. Stability is indicated by N. Ri combines N with horizontal shear.

We have revised the manuscript by inclusion of square Brunt-Väisälä frequency $N^2$ and square shear frequency $S^2$ as they better reflect our intent.

441: an "identification" the source of observed EDR here is optimistic here, given the weakness of the "suggestions" seen in the data. Again, "wave" activity may be due to measurement anomalies that are periodic with vehicle motion.
451: given the strong horizontal advection, it is difficult to believe that turbulence features originating in the boundary layer could propagate into the stratosphere within the short 5km diameter of the (inertially fixed) helix of measurements.

As noted above, the waves and some of the vertical features are likely to have been introduced by the numerics of the interpolation scheme used and these interpretations have been removed. We have significantly updated section 3.4 and feel the the inclusion of $S^2$ and $N^2$ distributions make the connection between $EDR$ and shear/buoyancy easier to infer.

---

## Referee Report (RR1)

**Second Review of the manuscript "High-altitude atmospheric turbulence and infrasound measurements using a balloon-launched small unscrewed aircraft sustem" by A. N. Haghighi et al.**

**General comments:**

The authors have thoroughly revised their manuscript but there are still many shortcomings. Some additional information is unclear and not convincing. I think that the revised version cannot be published in its current form. In this second review, I focus on the main issues only.

(1) As outlined in the first review, the manuscript hesitates between the evaluation of a new technique and a scientific work on atmospheric processes, which makes the paper very unclear. The primary objective of the paper should be to provide all the information needed to give the reader a clear idea of the platform's strengths and weaknesses. It is not the case. For example, section 3.1 shows the comparisons between temperature, humidity and wind profiles obtained by Hidron and radiosondes. The differences and similarities between the two should be emphasized. Paragraph 4, page 14 (lines 354-364) is out of the topic, because it describes the temperature structure of the troposphere and stratosphere without considering the differences/consistencies between the UAV and radiosonde profiles. In addition, it is not clear why this description is useful, since it is not used afterwards.

(2) The description of Figure 7, page 17, focuses on the main trend of the wind profiles and on the presence of the jet-stream during flight 1, detected by both instruments. However, the authors do not discuss the "high-frequency"  fluctuations revealed by Hydron measurements, even though they are the most striking pattern of the three profiles (these fluctuations do not appear in the radiosonde data but their vertical resolution may be insufficient). Are they real or artefacts due to the helical pattern of the Hydron path, for example? Can they result from inertia gravity waves? Is a spectral analysis consistent with IGW? These questions, not addressed by the authors, are of prime importance, as their answer obviously conditions the quality of shear and Richardson number estimates.

(3) The newly introduced paragraph on stability conditions is unclear and not convincing. First, the text does not describe the red profiles (N2). We implicitly understand that they are obtained from the second method (described from line 407). The "noisy and spike" structure of N2 profiles and the large differences between the wind shear estimated from the two methods are suspect. Figure 9 shows Ri profiles for one method only (likely the method corresponding to the red profiles in Figure 8). Unfortunately, they are clearly not physical, as no results in the literature show such a stable atmosphere at a vertical resolution of 100 meters (or even lower). The authors should check the literature and their calculation methods.

(4) The expression of $\varepsilon$ (14), line 492 does not seem to be correct because it is equivalent to write $E_{ll}(k_l) = A$. As a consequence (?), there are inconsistencies in Figure 10. There are no substantial differences between TKE measured during flights 1, 2 and 3 (e.g. $\sim 0.1\ m^2 s^{-2}$ around 20 km). However, EDR(flight 1) is about ~5 times EDR(flight 2 and 3), i.e. $\varepsilon$(flight 1) is about $\sim 10^2$ larger. This is not consistent with TKE. The EDR profiles show an "envelope" that should correspond to the instrumental noise floor (apparently confirmed by a spectral slope close to 0 in figure 14e). The noise level was likely much more important during flight 1. During the present jet-stream, flight 1 should be more prone to turbulence, whereas Fig 10d seems to indicate the virtual absence of inertial turbulence. It is counter-intuitive and then suspect.

(5) The analysis in section 3.5 has been thoroughly revised, and considering the strong reduction of the resolution, this representation may show interesting trends. However, the description of the results are very difficult to follow. In addition, the results depend on the validity of the estimates of $N^2$, $S^2$ Ri and EDR described in the previous sections (e.g. the sudden "jump of $S^2$ above ~15 km is suspect). The authors conclude that all scenarios can occur (e.g. high turbulence with high Ri, low turbulence with high shear, etc.) but I believe that the reliability of the quantitative results is not high.

---

## Referee Report (RR2)

**Third Review of the manuscript "High-altitude atmospheric turbulence and infrasound measurements using a balloon-launched small unscrewed aircraft sustem" by A. N. Haghighi et al.**

**General comments:**

The authors' reply clarified points that helped me to better understand what had been done. The amount of work involved is indisputable, but some questions remain as to the reliability and interpretation of some results. Also, at this level of evaluation, more attention to form would have been appreciated, because there are many avoidable presentation errors. This gives the impression that the manuscript has been hastily revised and not proof-read. For example, there are many typos in the added text, and, e.g., in equation (2). Figure 9 caption does not correspond to the new version of the figure.

The manuscript has been improved but, from my point of view, still requires major revisions before publication.

1) The wind profiles measured from the onboard GPS during the ascent (now shown in figure 8) are *very important added value*s and provide convincing arguments for the quality of wind profiles measured by Hydron. They provide much more important information than NWS radiosondes (low resolution and large distance). However, they are only used for wind profile comparisons in Figure 8 and the comparison results are not mentioned in the abstract and conclusions. These additional GPS profiles should provide a good reference and can also be used to estimate wind shear at a vertical resolution of ~100 and ~1500 or 2500 m, since they are not affected by the (large) horizontal excursion as the Hydron. The wind comparison result is particularly good above 10 km for flight 3, with wind fluctuations of similar vertical wavelengths and amplitudes. Contrary to the authors' claims, this seems to suggest that the large horizontal excursion of the Hydron (and horizontal inhomogeneity) may not be the dominant factor explaining the shear (and likely N2) fluctuations at the vertical sampling of 100 m from time series. Also, the shear profiles in Figure 9 obtained at the vertical resolution of 100 m and supposed to be strongly "influenced" by the horizontal excursion (~1500 m) are quite typical of shear profiles estimated from balloon-borne radiosondes at a vertical resolution of 100 m (after applying a low-pass filter with a cut-off at 200 m on 10-m vertical resolution profiles). See the example below from a Vaisala radiosonde. This is another reason why I remain

skeptical about the interpretation of the results (lines 448-449) shown in figures 9 and 10, in terms of horizontal inhomogeneity.

[Figure]

**Wind shear (100 m)**
Vaisala sonde

In addition, if the effective vertical resolutions of $Ri_t$ and $Ri_z$ (and shear) are 100 m and 2500 m, respectively, the large difference between the two (figure 10) can be due to the (vertical) scale dependence of Ri (and shear) (*See: Balsley, B. B., Svensson, G., and Tjernström, M.: On the scale dependence of the gradient Richardson number in the residual layer, Bound.-Lay. Meteorol., 127, 57–72, 2008.*). Therefore, the difference in fluctuations at the two resolutions cannot be attributed to the (sole) horizontal inhomogeneity.

It turns out that I do not agree with the statement lines 481-485 because the authors refer to the estimates of $Ri_z$, always >>1 due to the poor vertical resolution (~2500 m). It is not adapted for the detection of shear instabilities commonly observed in the tropo-stratosphere (~a few hundred meter deep or less).

The fact that the vertical sampling is the same for the two profiles (Figure 9-10) is misleading: it gives the impression that the resolution is the same. The effective resolution should be given by using a clear terminology. For example: $Ri_{100}$ and $Ri_{2500}$. It should be used everywhere. Line 648 ("… with high Ri….") is ambiguous because the calculated Ri's cannot be directly compared to the thresholds (0.25 or 1).

Lines 410-415: It is not clear what the authors mean. The larger vertical wavelength of the fluctuations should be favorable to larger horizontal scale, and thus to horizontal

homogeneity. As a result, the aircraft's orbital trajectory should be less problematic under such conditions. In addition, the argument is weak: the wind profile during flight 2 shows larger amplitude fluctuations during the balloon measurements (ascent) between 5 and 8 km than during the Hydron measurements (descent).

In line 409, do the authors mean "…that are NOT evident in the NWS soundings"?

2) N is shown instead of $N^2$ in figure 9. The profiles contain negative values, which is not possible, because N is necessarily real positive (when $N^2 > 0$) or imaginary (when $N^2 < 0$). Can the authors explain how they calculated N in practice?

Lines 483-484 are not clear.

Line 505: "….caused by *inertial* turbulence".

Line 542: n must be equal to -5/3.

In Figure 12, the red lines should be limited to $k_l > 0.1$ because it was the threshold used for linear fitting.

Line 595: "… nearly constant with $\alpha$" (?)

In the conclusion section, Lines 646-649 comment on comparisons between Ri and EDR (the former figure 13), which are no longer described. Qualitative comparisons can only be made from Figures 14-15 and 16.

---

## Referee Report (RR3)

**Referee 3 Response to Manuscript Revisions**

Referee comments on the original manuscript are in black. Author responses are in blue. Referee responses to the revised manuscript and author responses are in green. (Apologies for the rough formatting of space).

We appreciate the time taken by all referees in providing insightful and detailed comments about our manuscript. Following the reviewers recommendations, numerous changes have been made in the analysis approach including: (1) Revisiting the conversion of the five-hole-probe wind estimate; (2) adjusting the statistical ensemble sized from being based on 30 s ensembles in time to 1500 m ensembles in space (with 750 m overlap between ensembles to retain spatial resolution); (3) revised the spectra calculation with the correct transformation from frequency to time domain; (4) updated the method used to calculate vertical gradients and adding Brunt-Väisälä and shear frequency measurements to the paper; and (5) addressing issues in the methodology used to generate contour plots. Of these changes, the largest impact on the results was the change made to the vertical gradient calculation, which impacted the gradient Richardson number, $Ri$, values and the revision to the contour plots. As a result of these changes, and other additional changes made to address specific comments made by the reviewers, we believe the revised version of the manuscript is more clear and complete than the version originally submitted. To help clarify where changes have been made in the revised version, we have highlighted all changes made using blue text.

Below, we respond to the individual comments made by the referee. To do so, we have reproduced the original review, with our comments provided in blue text.

**Reviewer(s)' Comments to Author:**
This paper presents some intriguing results using a new measurement platform for profiling the atmospheric column descending from about 20km. However, the quality of the observed data is not clear, given the periodic variations that may be a result of the periodic orbit of the gliding aircraft platform. As a result, the conclusions drawn about the viability of the sensing method and the relation to potential atmospheric structures and sources is tenuous, without furhter examination of the correlations between signal variations and platform motions.

As we were also concerned with apparent wind magnitude variations with altitude having a vertical wavelength similar to the altitude difference between successive aircraft orbits, we have thoroughly revisited the wind measurements and found several places where improvements could be implemented, including improving the time alignment between autopilot kinematic variables and payload sensors, identifying and correcting pitch and yaw probe misalignment (of less than 10˚) between aircraft and sensor coordinates, and discovering an error in probe rotation. However, these improvements only marginally modified the wind magnitude and did not affect the observed vertical profile in any meaningful manner.

We have also closely examined the dependence of the vertical profiles of wind magnitude with heading, as shown in Figure 1. The most notable similarity between vertical separation of orbits and vertical wavelengths in wind magnitude occurred during Flight 1. However examining successive orbits shows that the wavelengths are not identical, with the orbit vertical distance slightly longer. Therefore, if there is a bias in the measurement, it is not rigidly correlated to the heading.

We also perturbed different inputs into the wind estimation, and found that the vertical wavelengths in the wind estimate appears to be robust to these changes. In summary, we could find no conclusive evidence of bias in the measurements introduced by the aircraft heading and the periodic orbit.

Detailed comments:

Introduction: "The trajectory of the glider allowed for improved statistical convergence and higher spatial resolution of derived statistics measured by the in-situ sensors." Refers to balloon-borne measurements, but such improvements and higher spatial resolution were not demonstrated in the paper.

Here we were referring to the airspeed of the glider able to transect the flow at flight speeds over 20 m/s which, when compared to balloon-borne measurements, means that more wavelengths of turbulence can be measured over the same duration of sampling time. However, spatial resolution has different connotations and is not the correct term for what we are trying to describe. We have revised the text to better reflect our intended meaning.

Similarly: "which allowed the connection to be made between the locations of increased turbulence intensity and the source of its generation" was tenuous, only to the level of "consistent with".

[Figure]

Figure 1: Vertical profiles of wind magnitude from all three flights shown with horizontal lines indicating location where aircraft heading passed through 180˚. (left) Flight 1 (middle) Flight 2 (right) Flight 3

This is a fair assessment, we have revised the text accordingly.

The abstract still contains this same phrase. Also in the abstract, the sentence "By being able to transect the air, the glider allows for turbulence wavelengths to be sampled at a particular altitude, improving statistical convergence and spatial resolution of derived statistics from its in-situ sensors." is misleading, since the glider cannot remain at a particular altitude, and demonstration of improvement of convergence and spatial resolution remains missing in the paper, as noted in the previous comments above.

40: "with these results used to model the relationship between turbulence in the stratosphere as well as tropospheric activity" not clear: "as well as" vs. "and"?

Revised. 'and' should have been used.

85 "However, due to the transient nature of their Lagrangian flight trajectory, balloon-based approaches are not necessarily amenable to obtaining detailed statistical descriptions of turbulence at high altitudes." Why? Aircraft are also transient, and if GPS guided, only see what is advected past. Balloons with altitude profiling are not Lagrangian vertically, so also sample more than one parcel of air. The statement is vague: it depends what statistics are being evaluated.

True, and addresses the same point as the first detailed comment. We have revised the statement to be less vague.

115: "However, as balloons advect horizontally with the wind and are unable to maintain a fixed geospatial location, balloon-based approaches would benefit from complementary measurement approaches" is even more vague.
120: The scientific advantage of remaining over a fixed ground location is not self-evident, as the features of the atmosphere typically do not. Or is the advantage primarily operational in nature? Please articulate this point. Also, the transecting ability of a glider is not related to the helical profile, but to its horizontal airspeed. The sentence "These qualities facilitate the statistical analysis necessary for quantification of non-stationary properties." remains unsupported in the paper (that I can find).

90: "A glider offers advantages over traditional balloon launches by being able to maximize time at altitude

during its descent phase" Vertical rates for the glider vary from 5m/s to 1 m/s, very similar to descending balloons. "These qualities facilitate the statistical analysis necessary for quantification of non-stationary properties" is not supported by evidence in the paper.

We respectfully, but strongly, disagree on this point. Note that in 1000 m of altitude change, for the current experiments the orbit of the glider means that it samples approximately 15,000 m along its flight path, whereas a balloon will sample only the 1000 m. This is a significant difference in the amount of atmosphere and range of eddy sizes that are sampled over the same vertical distance. Note also that the current configuration of the aircraft means that the turbulent eddies are acquired at an order of magnitude higher temporal resolution as well. It therefore would be very difficult to reproduce the spectra of Figure 11 (and corresponding $k$ and $\varepsilon$ estimates), and azimuthal distributions of these statistical properties (as shown in Fig. 14) with a balloon, particularly over the wavenumber range and at the vertical resolution that the glider can measure.

We have added the above discussion to the conclusions to ensure that these points are not overlooked by other readers.

The objectionable phrase has apparently been removed from the introduction. The confusion arose because the vertical resolutions of both platforms are similar; the differences relate to the horizontal velocity.

Some confusion remains in conflating stationarity with homogeneity. Variations seen in a fast (e.g., 60 m/s) transect are likely to be due more to spatial inhomogeneity than temporal changes in statistics, provided that the overall time interval of the record is small. The higher glider airspeed speed allows larger spatial scales to be assessed in a shorter time, making the Taylor assumptions used in the turbulence estimation frequency-to-wavenumber conversion more likely to hold for the larger scale observations.

Because the glider transects are mostly horizontal, the changes measured are likely to be due more to horizontal inhomogeneity than vertical inhomogeneity. Balloons provide vertical profiles of parcels and cannot sample laterally. Their ability to ascend/descend slowly (few m/s) provides high spatial resolution (vertically). A glider with 60 m/s airspeed and a 15:1 glideslope, although descending at a similar rate (4 m/s here) could be considered as having a similar vertical resolution, provided that the variations observed are dominated by vertical variations, and not the lateral ones. E.g., for the 1500m record lengths used, is it more likely that data reflects the 100m vertical change, or the 1500m lateral change? The latter is supported by later discussion in the paper (around line 449), where time-successive vertical gradients are erratic, due to the large lateral changes, and where true vertical gradients are assessed by considering segments on successive turns on the helix spaced (vertically) about 2.5 km apart, leading to vertical resolutions on this order. This does not compare favorably to balloon vertical resolution.

A glider provides complementary measurements by sampling (mostly) laterally, and in this case with a very large helix diameter (10km) with an ability to evaluate large scale lateral inhomogeneities. Here with record lengths of 1500m, lateral spatial resolution, e.g. of turbulence quantities, is of this order.

Higher temporal resolution due to higher airspeed is not an inherent advantage, since the frequency-to-wavenumber conversion makes, say, epsilon, invariant. The effect on instrumentation, is, in fact unfavorable, since higher sample rates are required, and the inherent bandwidth of the sensors and plumbing can be a limiting factor for turbulence parameterization. E.g., with a 20 Hz sensor bandwidth, scale sizes greater than 2m can be observed at 60 m/s (assuming no noise floor issues). Preserving 1 decade of inertial range would require a record length of at least 20m, but with the noise floor seen in Figure 12 c), about 60m records would be required, providing a maximum of 60m

(mostly horizontal) resolution for turbulence parameters. A balloon moving at 4 m/s (with the same sensors) would have a (vertical) spatial resolution of 4m.

Thus, high-speed slant-path sampling has complex trade-offs that are not served by oversimplification. The paper could be improved by a more concrete discussion of the relative merits in fundamental terms, or refrain from vague comparisons. E.g., I don't see where the analysis records are defined for the Turbulence analysis. So it's hard to place the unquantified characterizations such as "high vertical resolution" (now in the Conclusion) in context. Likewise, "increased statistical convergence" would depend on a larger number of points in the data record, together with an assumption of statistical stationarity. The data record sizes were only discussed as covering a fixed horizontal distance, so the record sizes would be smaller at the higher altitudes, and the time intervals shorter. It would be helpful to describe these details in order to support the advantages discussed.

Difficulty of conducting UAS measurements of this type in the NAS was not discussed, nor the conditions under which the reported flights were allowed. Was this in restricted airspace? Under who's auspices? Or was this in the NAS under a COA?

The reviewer raises a good point as the current regulatory environment prevents these types of measurements from being routinely conducted. In the current experiment, flights were conducted in restricted airspace managed by the SpacePort America facility and coordinated with the nearby White Sands Missile range. We have added revisions to the manuscript to include these points.

I don't see where White Sands coordination is mentioned.

125: The iMET sensor specifications were not referenced. These accuracies and time constants tend to degrade at lower pressures and temperatures, and this was not indicated.

We have added a citation to the manufacturer's webpage where this information was obtained. Note that these sensors were previously flown on a similar platform and found to be consistent with model results (Schuyler TJ, Gohari SMI, Pundsack G, Berchoff D, Guzman MI. Using a Balloon-Launched Unmanned Glider to Validate Real-Time WRF Modeling. Sensors. 2019; 19(8):1914. https://doi.org/10.3390/s19081914).

Figure 3 would benefit from the addition of dimensions to the components pictured.

Dimensions have been added to the figures

150: "Comparison of calibrations with and without heating active indicated that there was no influence of probe heating on the five-hole-probe response characteristics." Not clear what response characteristics means: time constant? Calibration coefficients? Noise level?

We are referring to the calibration surfaces and have revised the text to be more precise.

153: "Each hole on the probe was connected to differential pressure transducers through 1.75 mm diameter flexible polymer tubing." What was the other port of the differential pressure sensor connected to? Presumably this was the "static pressure port", but this was not shown or described in the paper. How long was the tubing (this can have a detrimental effect frequency response of the air speed measurement, as noted later in the paper).

We have added these details to the text.

160: "Note that the during flight, the autopilot maintained flight speeds sufficient to produce pressure differences well within the range of the low-sensitivity transducers and hence only the readings from these sensors were used for this analysis." Please quantify the airspeeds obtained, and the corresponding average differential pressures.

We have added figures showing relative air velocities and noted the value of dynamic pressure in the text.

195: how was aircraft sideslip angle determined? How did the use of this affect the quality of the horizontal wind measurements?

While revisiting the wind measurement procedures, it was found that the probe was actually rotated 90 degrees relative to what the authors initially thought. This meant that it was actually the pitch holes that were disconnected for flights 2 and 3 and not that yaw holes. We also compared flight 1 data with and without the revisions required to calculate winds for flight 2 and found that the differences were negligible. The text has been revised the text accordingly.

The discussion of wind calculation for flights 2 and 3 in paragraph 254-262 is vague and disconcerting. The pressure lines disconnected are not specified in the text. The use of vertical aircraft speed from the "variometer" (presumably pressure altitude rate, i.e., inertial aircraft vertical velocity) does not reliably indicate vertical airspeed, just as inertial horizontal velocity does not indicate horizontal airspeed. "Flight 1 data was processed with the original and revised approach, and the impact on the results on the resulting wind velocity statistics found to be negligible". Which statistics? How negligible? Angle of attack, and hence vertical wind velocity

is likely to be significantly affected, but the horizontal wind may not depend on this very much. "although some differences in the frequency content of the vertical wind component could be expected." Why? What differences? Why is this important? Vertical wind velocity does not seem to be used later.

Generally, the details of this particular mutli-hole probe and its calibration and resulting accuracy were not provided. Can these be referenced from an earlier publication?

The probe used here is derived directly from the probes used in :

Barbieri, L. and Kral, S. T. and Bailey, S.C.C. and Frazier, A.E. and Jacob, J.D. and Reuder, J. and Brus, D. and Chilson, P.B. and Crick, C. and Detweiler, C. and others (2019) "Intercomparison of small unmanned aircraft system (sUAS) measurements for atmospheric science during the LAPSE-RATE campaign," *Sensors* 19(9), 2179.

and utilize calibration systems and approaches described in:

Witte, B.M., Singler, R.F. and Bailey, S.C.C. (2017) "Development of an Unmanned Aerial Vehicle for the Measurement of Turbulence in the Atmospheric Boundary Layer," *Atmosphere*, 8(10), 195.

Al-Ghussain, L. and Bailey, S. C. C. (2022) "Uncrewed Aircraft System Measurements of Atmospheric Surface-Layer Structure During Morning Transition," *Boundary Layer Meteorology*, v185, 229-258.

We have added these references to the revised manuscript.

Note, that these probes have also been successfully deployed in previous studies, including:

Bailey, S.C.C., Smith, S. W., Sama, M.P., Al-Ghussain, L. and de Boer, G. (2023) "Shallow katabatic flow in a complex valley: An observational case study leveraging uncrewed aircraft systems," *Boundary Layer Meteorology*, v186, 399–422.

Bailey, S.C.C., Sama, M.P., Canter, C.A., Pampolini, L.F, Lippay, Z.S., Schuyler, T.J., Hamilton, J.D., MacPhee, S.B., Rowe, I.S., Sanders, C.D., Smith, V.G., Vezzi, C.N., Wight, H.M., Hoagg, J.B., Guzman, M.I. and Suzanne Weaver Smith (2020) "University of Kentucky measurements of wind, temperature, pressure and humidity in support of LAPSE-RATE using multisite fixed-wing and rotorcraft unmanned aerial systems," *Earth System Science Data*, 12(3), 1759-1773.

Bailey S.C.C., Canter C.A., Sama M.P., Houston A.L. and Smith S.W. (2019) "Unmanned aerial vehicles reveal the impact of a total solar eclipse on the atmospheric surface layer" *Proceedings of the Royal Society A*, 47520190212.

so they, and their use, are not untested.

However, all these references relate to low altitude use. Has the probe been calibrated for use at the low pressures and temperatures encountered in this study? How do you know that the 20 Hz bandwidth (line 495) holds at high altitudes, since this depends on "viscous attenuation of the pressure fluctuations within the tubing" (line 252), and kinematic viscosity increases markedly at high altitudes.

205: How was the microphone mounted on the vehicle? Was it protected from dynamic pressure fluctuations? If so, how did this filter the infrasound pressure waves? Could aircraft motions (that are also dependent on ambient turbulence) influence these measurements?

The microphone was mounted rigidly within the nose of the aircraft, with the diaphragm facing forward. Being within the fuselage, the microphone was protected by dynamic pressure fluctuations. Note that infrasonic sound waves will be of the order of 30 m and larger, so attenuation of the sound waves by the fuselage is not expected in this configuration. Due to the rigid mounting of the probe in the aircraft, it is not anticipated that aircraft motion could influence the microphone, however we were not able to verify this assumption from the current set of measurements.

We have added this information to the revised manuscript.

How are winds calculated?

Winds were calculated based on the procedures described in lines 153 to 187 (lines 199 to 231 in the revised manuscript).

How is airspeed calculated? No plots of airspeed were provided. What was the airspeed as a function of altitude?

Airspeed was measured by both the aircraft's Pitot probe and the five-hole probe using the standard procedure of measuring dynamic pressure across total pressure (central hole) and static pressure and found to be in agreement between the two instruments. Plots of relative air velocity have been added to the revised manuscript.

220: temporal alignment can be intricate. How was this accomplished with this data? Was there a common time reference?

Additional clarification added to text. Initially we intended to use some of the statistics calculated by the payload and sent to autopilot via RS232 communication and recorded in the the aircraft telemetry stream, but found correlating the dynamic pressure measured from the aircraft pitot probe and measured from five hole probe to be a more reliable alignment indicator due to suspected buffering delays in the RS232 connection. Note that, although not mentioned in the manuscript for brevity, we were able to identify and remove a 0.005% difference in clocks between the two systems by windowing the correlation of the airspeed and five-hole probe. This corresponds to a 1 second difference in timing over the six hours of measurement.

230: what does "controlled landing" mean here? Manual landing (RC), or automatic landing (autopilot)?

Landings were conducted by the autopilot. This has been noted in the revised manuscript.

229: A portion of the descent from the 30km release seems to be very steep. There were also some very tight circles at isolated points in the first two descents. Why?

These flights were also test flights for the aircraft. During the flight, the operators conducted several tests of their systems which included adjusting the flight profile mid-flight and improving the response of the aircraft following release. Note that the steep release at 30 km is due to the requirement to achieve sufficient dynamic pressure to produce enough lift for controlled flight. The lower density means that the aircraft must fall a certain distance before the aircraft can travel to it's measurement location.

I don't see any revisions to this effect. It would be interesting to know the "measurement ceiling" of this system.

235: I think you mean UTC -6:00 here.

Correct. Revision made.

239: "Due to the configuration of the sensors on the aircraft". Vague. Please describe what about the configuration makes the sensor readings unreliable on ascent.

During ascent, they are in a stagnant region within the wing pillar wake and therefore not sufficiently aspirated to prevent self-heating and delayed air exchange with the environment. The result was that the $T$ and RH measurements had a significant altitude-dependent warm bias of 5 to 15 degrees K relative to descent and the radiosonde measurements. On descent, these sensors are introduced into the oncoming airflow and properly aspirated. We added more details to the revised manuscript.

289: "with backing"?

Revised.

295: central differencing between adjacent 30 sec averaged values?

That is correct, although we have updated the revised manuscript to use spatially regular, rather than temporally regular, segments.

As noted above, the effect on the record samples and time length would be helpful.

230: regression fit to a constant function over 150 sec? Central 30 sec interval with 2 intervals before and 2 after?

Prior to differencing we employ a 5 point moving average where the smoothed value at the central segment is found by averaging the values averaged within the central segment with average values from the 2 segments before and 2 segments after. We have added more details in the text.

271: "this is likely due to spatial heterogeneity in the atmospheric moisture concentration". Could also be due to an instrumentation anomaly.

It is possible. We have added more radiosonde data to these figures to better represent the spatial heterogeneity. The results suggest there might be some dry bias in the $RH$ sensor in cold temperatures.

The following statement "cloud conditions near Truth or Consequences, NM (near Spaceport America) were different" does not help. Different how? At what altitudes?

The ASOS reports only qualitative conditions (i.e. scattered, clear, overcast, etc.). However, the additional radiosonde profiles made the ASOS reported cloud unnecessary and this text was removed. We also now include satellite imagery in Appendix A, which provides a more nuanced illustration of the different humidity conditions which could be expected.

276: compare well given the spatial offset and the local weather conditions, and if the local periodic variations are ignored. These variations are suspiciously periodic with altitude, raising questions about artifacts from the platform airspeed/attitude/descent rate that may be varying with the same period. (See the related comments about Ri later). Some evidence should be provided that these results are not correlated with aircraft motions.

This was addressed when responding to the initial comment. The additional radiosonde profiles in the revised manuscript also provide increased context for the wind fluctuations observed.

The radiosonde wind magnitudes seem highly smoothed/decimated with straight lines connecting large changes in altitude. So much so that they don't really corroborate the "periodic" variations in the Hydron winds of concern. I would think that standard 1Hz cadence radiosondes would provide higher resolution.

300: Although it probably does not make much difference, z in this formula should be altitude MSL, not AGL. Recommend that MSL be used throughout for consistency and for interpretation of the results. Also, I can't seem to find the altitude of the ground at the launch location.

True, this was a typo as we made our calculation using MSL. However, based off of Reviewer 2's comments, we have removed the altitude dependent $g$ calculation from the revised manuscript. We have also added the m.s.l. altitudes of the launch and recovery location manuscript.

303: The Ri profiles seem to have highly periodic excursions with altitude. Could these be at the same period as the orbits the plane executes on the descent? That is, how do we know this is not an artifact of the sensors or the periodic motion of the platform? It would also be good to see how this correlates with the bank angle of the plane, since this will not be constant in wind. It will be difficult to take the results at face value without careful checking for motion/attitude/descent rate artifacts from the platform. Similarly, the Ri values seem suspiciously low, with < 0.25 values for much of the flight. Are these low values periodic anomalies in the measurements?

We have extensively updated the text addressing this variability and present an improved method to calculate Richardson number. See additional discussion above regarding the dependence of wind velocity on the orbits.

455: "The vertical gradient of each quantity was then found by finding the nearest statistical segments in the positive and negative z directions which minimized the difference in α". This is difficult to unpack (took me a while). I think you are saying that helix segments one turn apart, above or below each other, are selected for each alpha. This could be clarified better.

Figure10: Rit is erratic and very small, Riz is less erratic but very large! As noted, this mismatch is due to the mismatch in Si vs Sz, since Nz essentially smooths Ni. But the reasoning behind the difference is S also applies to N, so the issue with S is "most likely" due to something else. Thus it is hard to believe either Ri value, or either S value.

323: how does $< u >$ differ from $< U >$ used earlier?

We use $u$, $v$, and $w$ to indicate the components of the wind vector having a projected horizontal magnitude of

$U$. These distinctions were defined on line 183 in the original submission with $U$ specifically defined in equation 3 (now equation 7 in the revised manuscript) .

327: what were the subintervals and overlap used in the Welch method? The "spectra" in Figure 8 seem to have a 40s period fundamental frequency, so this is confusing.

We applied the Welch method without using only three subintervals and a 50% overlap. We have also verified that the application of Welch's method does not impact the shape of the resulting velocity spectrum. The 40s fundamental period is due to mislabeling of the figure, as discussed later.

$F_{uu}$ and $Phi_{uu}$ are both used for the power spectral density (please be consistent), and these are incorrectly called the "frequency spectrum".

Please note that we are using colloquial language used in contemporary fundamental turbulence literature whereby the power spectral density is referred to as the velocity spectrum, or more broadly the energy spectrum, or shortened to simply the spectrum depending on context. Furthermore, these "spectra" are often distinguished as to whether their dependence is in the frequency domain or wavenumber domain by referring to them as the "frequency spectrum" or "wavenumber spectrum" respectively (see, for example, Pope (2000) "Turbulent Flows", Cambridge University Press.) We made the distinction in the original manuscript between $F_{uu}$ ( $f$ ) (i.e. t∫he frequency∫spectrum) and $\Phi_{uu}(\kappa)$ (i.e. the wavenumber spectrum) as these are separate functions since both $\int_0^\infty F\,df$ and $\int_0^\infty \Phi\,d\kappa$ must both return the variance of the velocity component that they are calculated for (e.g. $u$ in the above case). As $\kappa \approx 2\pi\,f\,/V_{rel}$ this means that $\Phi \approx F\,(V_{rel}/2\pi)$. We realize that this distinction is confounded by the fact that the original figure 5 was mislabeled as $\Phi_{uu}$ instead of $F_{uu}$ for two of the subfigures, but this was a typographical error and not intentionally inconsistent. In the revised manuscript we strive to be more clear that we are referring to the velocity spectrum, and try to be more clear with the distinction between $F$ ( $f$ ) and $\Phi(\kappa)$. Reviewer 2 also did not like the use of $\Phi(\kappa)$, so we have now replaced that with $E$ ($\kappa$). We also now present the wavenumber spectra instead of the frequency spectra in the revised manuscript as it was more relevant for the discussion of determining the inertial subrange slope.

Because of the confusion in the literature, it would be helpful to distinguish power spectral density from amplitude spectra, power spectra, etc., since this makes a difference in the quantitative results. Note the Welch method can also be used for smoothing the amplitude spectrum, and the discrete Fourier transform can use different normalizations. To make it clear what was done, it would be good to include the expressions for the Fourier transforms used, or provide a specific reference.

328: Was the Hanning window variance-preserving?

Yes. We confirmed it is variance preserving prior to implementing it.

329: integration of the PSD is from .025 Hz to 5Hz, so the most important (largest amplitude) components of TKE are potentially not included. This makes TKE difficult to estimate without some idea of the local "outer scale" where the spectral energy ceases to increase as frequency decreases. This is noted later (347), but with a confusing reference to 300m as the longest period in the k "measurement", since earlier in the paper 2420m was quoted as the longest interval in the 30s analysis intervals. However, no mention of the outer scale was made. Thus the k retrieval has a highly variable lower spatial scale with altitude, and the relevance of the k profiles is unclear.

We had used 300 m as it is the average distance travelled during the 30 s sample duration, but acknowledge that the airspeed at high altitude was much higher, leading to the 2420 m length of the longest segments. In the revised manuscript we use a constant statistical segment length. Although this may not fully resolve the largest eddies it should provide a consistent wavenumber range for which TKE is calculated. We did work at trying to implement an approach which spatially varied the segment length depending on the energy content, but due to the range of scales experienced during descent, we were not happy with the implementation and feel it needs further development before including in a publication.

339: shape of the "spectra" in Figure 8 is very strange. Part a) seems to have a noise floor near $10^{(-3)}$, but the noise floor is smaller (near $10^{(-6)}$ at a higher altitude (part b)! (Where pressure fluctuations are necessarily smaller). A noise floor is again seen near $10^{(-3)}$ in part c). Also, the spectral slope is too shallow in part c) as noted later in the paper. Could it be that this power spectral density is the noise figure of the sensor itself, and there is really no detectable signal at these low atmospheric pressures?

As noted above we had mislabeled Fig. 8 due to changing from presenting the results in the wavenumber domain to the frequency domain. The original figures were actually presenting the results in the waveunumber domain and we are unhappy that this error made its way into the final submitted manuscript. We also believe the reviewer is correct that the -1 slope is the noise figure of the sensor (or more accurately the combination of sensors used to determine the wind) and that the -1 slope indicates that there is no measurable turbulence present.

The noise floor noted above was a flat level, not a f^-1 slope. This is typical of electronic Johnson noise in sensors. This can be seen at high frequencies in the new Figure 12 b) (at a level of about 10^-3) and c) (at a level of about 10^-2), which now makes more sense with the increase in altitude. Were the spectra in Fig. 12 cases

deemed Kolmogorov turbulence? If so, it would be good to show spectra for those that did not so the noise floor issue can be clarified against your fitting/qualification procedure.

356: Buoyancy Reynolds number can be calculated after estimating epsilon, as a check on this assumption.

This assumption is not expected to be valid over the entire measurement range, and is checked/unvalidated by the power law fit discussed on lines 456-457. However, assuming it's validity is necessary to produce a dissipation rate estimate using this approach. Validity of this assumption is therefore reflected by the filled circles on Figures 9 and 10.

Anomalous spectral shapes can return a f^-5/3 fit. It's the standard deviation of fit that is telling, and this not reported. Lines 456-457 in the revised manuscript concern N and S, not epsilon. The case of flight 1 in figure 12 is strange in that it has EDR values higher than many points in flights 2 and 3, yet these values are not qualified as turbulent. Also seems strange that the EDR is higher but TKE is smaller for flight 1. This does not make sense. How do the spectral shapes compare? Seems the sensor direction in flight 1 has been fixed from the original paper, but are there any other sensor system differences between flights? Did w' make a significant difference in the TKE calculations for flights 2 and 3, given the comments earlier that w' is likely to be incorrect?

364: the $\kappa_1$ wavenumber component in (9) is the longitudinal component of the motion of the air relative to the sensor. This is only the longitudinal component of the vehicle ground velocity in the special case of zero mean wind (still air), or in the limit when the vehicle airspeed is much larger than the wind speed. This should be corrected to use the airspeed, and the course heading frame rotation should be replaced by one based on angle of attack and sideslip of the sensor relative to the relative wind vector.
366: again use of ground speed here in incorrect. Must be airspeed.

Thank you for raising this concern. Our initial approach was selected in a misguided attempt to minimize the effect of bias introduced by the dependence of Taylor's hypothesis on the wavenumber dependence of the wind velocity used for its application (discussed by Moin, JFM, 2009). Our experience in the ABL has indicated that neglecting the advection due to mean wind does not impact the resulting spectra, however these measurements are typically made in winds an order of magnitude smaller than the UAS ground speed. As pointed out by the reviewer, this is not necessarily the case for the measurements reported here. We therefore have revised the calculation of the longitudinal wavenumber spectrum in the current version of the manuscript to account for the advection due to mean wind. This includes adjusting the estimate of wavenumber from aircraft relative air speed and using the relative velocity component of the wind when calculating the spectra.

367: I don't understand the expression for $\Phi(\kappa_1)$.

This is discussed above and relates to the property that $\int_0^\infty F \, df$ and $\int_0^\infty \Phi \, d\kappa$ must both return the variance of the velocity component that they are calculated. We have updated the text to provide a better indicator.

Figure 9: Suggest plotting k and epsilon (or EDR) on a log scale to look for periodic artifacts (as noted earlier), and to make it easier to see the full range of these power function values.

We have plotted these figures on logarithmic axes in the revised manuscript and find no clear indication of artifacts due to athe periodic orbits.

Not clear how (of if) the noise floor/noise figure is removed in the qualified data fits.

We made no attempt to remove the noise floor from the data fits.

The noise floor can corrupt the estimated slope, so its removal can often retrieve epsilon values that were previously rejected. This could help to explain the absence of qualified values at the higher altitudes (ala line 552),

386: "It will be shown later that this enhanced EDR corresponds to measured fluctuations in velocity introduced by the presence of gravity waves at these altitudes". How? Gravity waves have a much large wavelength than could be influencing these epsilon estimates.

Unfortunately, the appearance of gravity waves appears to have been contaminated by the two-dimensional interpolation technique used to in the $\langle T \rangle'$ and $\langle w \rangle'$ analysis used to discern the presence of gravity waves. Although we believe there is still merit in the concept behind the approach, we could not come up with a suitable implementation in time to include in the revision. We have therefore removed the gravity wave discussion from the revised manuscript.

401: confirmation that the infrasound signal is due to turbulence is too strong a conclusion at this stage. Localized increases do not correspond to those in EDR.

This statement was driven by the increase observed as the aircraft enters the boundary layer and becomes immersed in the turbulence (as evident in the comparison of Figure 10 and Fig.12) and also the comparisons in Figures 14-16. However, we agree that this conclusion is weak, and that the infrasonic measurements need more development.

570: "with the range used selected due to finding that lower frequency acoustic content better correlated with the EDR values measured with the five-hole-probe when compared to the higher frequency acoustic content, which tended to contain additional signal noise." This is problematic: seems like you are cherry picking aspects of the infrasound signal that correlate with EDR, in order to confirm that infrasound signals

indicate turbulence (EDR). What is the nature of the "additional signal noise"?

577: despite agreeing that "this conclusion is weak" above, the line "providing an initial confirmation of the presence of infrasonic sound generation by turbulence" is still present in the paper.

426: the "interesting features" in figures 12-14 show strong correlation with location on the flight path circle. This might be due to differences in structure in the atmosphere across the 5km circle diameter, but it might also be due to sensor signal dependence on the heading or attitude or airspeed of the vehicle, that is also periodic with location on the flight path (as noted above). Given that these "features" in the data persist over large altitude ranges (where e.g. shear and stability are expected to vary significantly), and the intermittent, sometimes contradictory correlations noted in the paper, it is difficult to consider the conclusions offered as more than optimistic interpretations of rather murky relationships. Too much is made of data that has not been thoroughly vetted.

We note that the statistics being presented are calculated over relatively short windows (of a length about 5% of the orbit circumference) and, as many of them require mean subtraction, they should be largely unaffected by any bias which may be introduced by aircraft heading.

When preparing Section 3.4 we were also concerned with the potential for bias due to interpolation and had ensured that there was at least one measurement point per interpolation grid cell. However, when revisiting these figures during the revision we found that the numerical interpolation scheme (which was an implementation of Delaunay triangulation) was creating cell values very different from the measurement point within the cell. Therefore, although not appearing to be due to position in the orbit, the reviewer's concerns were well-founded.

As Section 3.4 interpreted these figures in the context of $\langle \phi \rangle'$, and this approach used the interpolated data for background subtraction, we have removed this approach from the manuscript completely rewritten this section (which is now Section 3.5 in the revised manuscript). We still believe that there is merit in this visualization approach, however, since it does show interrelationship between statistical values determined at measurement points at the same azimuthal locations along the orbit. In the updated section, the contours are determined by direct triangulation between points and we have included the actual measurement locations and values in the revised figures to allow the reader to directly assess the impact of the triangulation on the contours.

449: Ri is used as a marker for stability in various places, which is confusing. Stability is indicated by N. Ri combines N with horizontal shear.

We have revised the manuscript by inclusion of square Brunt-Vӓisӓlä frequency $N^2$ and square shear frequency $S^2$ as they better reflect our intent.

441: an "identification" the source of observed EDR here is optimistic here, given the weakness of the "suggestions" seen in the data. Again, "wave" activity may be due to measurement anomalies that are periodic with vehicle motion.
451: given the strong horizontal advection, it is difficult to believe that turbulence features originating in the boundary layer could propagate into the stratosphere within the short 5km diameter of the (inertially fixed) helix of measurements.

As noted above, the waves and some of the vertical features are likely to have been introduced by the numerics of the interpolation scheme used and these interpretations have been removed. We have significantly updated section 3.4 and feel the the inclusion of $S^2$ and $N^2$ distributions make the connection between $E\,DR$ and shear/buoyancy easier to infer.

592: "We therefore also only examine the behavior of other quantities in terms of relative trends in these values, rather than specific thresholds." Vague.

595: "Examining Flight 1, the distribution of measured Nz shown in Fig. 14a indicates that the static stability conditions are nearly constant with altitude." Stability is significantly increasing with altitude! This was also stated earlier in line 469.

655: "were actually produced by the aircraft passing through the same structure more than once." This is not clear from the discussion in that section. Pointing out a specific example or two would help to support this comment.

662: Despite this ambiguity, these initial flights suggest that the sUAS measurements suggest the potential exists" Please reword.

666: "the flight pattern allows for increased statistical convergence due to the larger volume of air sampled over a particular altitude range." How does a larger sample volume automatically provide "increased statistical convergence"? This is likely to produce more variation in measured quantities. Also, the conclusion is not a natural place to introduce a specific argument.

---

## Referee Report (RR4)

**Reviewer 3 Response to Authors**

Comments are inserted into the Author's latest Response to Referee 3 document, preserving the original colors: black for reviewer comments and blue for author responses. Reviewer comments for this latest revision are colored in green. Items without additional green comments are acceptable in the revision.

This version provides more information on the filtering scheme for multi-hole probe pressure measurements, but the noise-subtraction approach remains highly suspect. Within this, however, is a basis for determining the noise floor of the measurements, which could then be used to select spectral data above the floor by a suitable factor for turbulence characterization. The estimation process for vertical wind remains unclear, particularly for flights 2 and 3 where one of the pressure sensors was disconnected. As this is not really needed for the results of the paper, perhaps this could be omitted if it cannot be clarified.

We appreciate the time taken by the referees to once again review our manuscript and provide comments and suggestions for its improvement.
   Below, we respond to the individual comments made by the referee on this version of the manuscript. To do so, we have reproduced the original review, with our comments provided in blue text.

   This paper has been improved by a tighter focus and omission of many previous vague and unsupported statements. Most of the arguments from the previous reviews have been settled by removal of the sections in question, or by clarification in this revision. Remaining issues are noted below.

   Comments:

   – Line 10: "nominal vertical resolution on the order of 1 m during was achieved" Please re-word.

      We have reworded this statement to 'a nominal vertical resolution'.

   – Line 15: "and broadband response measured within boundary layer turbulence". Do you mean "broadband winds"?

      We have reworded this statement to "The low-frequency response of the infrasonic microphone was found to corre- late to long wavelength wind velocity fluctuations measured at high altitude, and broadband frequency response of the microphone was also measured within boundary layer turbulence.'

   – Line 27: "predict stratospheric and high altitude turbulence" High altitude is different than stratospheric?

      We have reworded this statement to 'predict stratospheric and upper-tropospheric turbulence'.

   – Line 114: "which allowed it to measure a wide range of turbulence scales horizontally at high vertical resolution." Confusing/misleading. Implies that horizontal and vertical effects can be separated, or that vertical variations dominate horizontal ones. Statement should be qualified accordingly.

      We have reworded this statement to 'which allowed it to measure over a distance of up to 50 m horizontally for a 1 m change of altitude'.

   – Line 211: "to check for any Reynolds-number dependence of $C_\beta$, $C_a$ and $C_q$." Were there any?

      We have added a statement that none was found.

   – Line 214: "flexible polymer tubing" could be almost anything. Usually, these tubes must be semi-rigid to properly propagate the pressure variations. Please specify.

      The manufacturer sells this product as PVC plastic tubing with a trade name of Tygon. We have updated the text to include this information.

   – Line 259: "angle between the true airspeed determined from the aircraft's pitot probe, and the vertical velocity determined by the aircraft's variometer". No idea what this means. Pitot does not provide an angle, and variometer (pressure altitude rate, akin to inertial velocity) is not the same as the vertical component of relative wind.

      When the autopilot manufacturer was queried as to how the angle of attack is determined, the description above is verbatim to the response we received. We have contacted them again and have updated the description of how $a$ was found, specifically from a combination of the true airspeed determined from the aircraft's pitot probe, the vertical speed

determined by a Kalman filter fusion of the static pressure rate of change, vertical acceleration, and global positioning system velocity, along with the aircraft orientation measured by the autopilot gyroscopes which was used to transform between inertial and body-fixed coordinates.

We have updated the text accordingly.

Text still incorrectly implies that vertical aircraft speed is the same as vertical wind, e.g. lines 325-326: "This orientation data allowed transformation of the vertical speed from inertial to body-fixed coordinates". How do you get angle of attack from vertical (body) velocity and pitch angle and pitot airspeed? Pitch angle is not the same as angle of attack, and flight path angle is not either, unless you assume vertical air motion is zero. The description of "a combination of factors" is still very vague, making it hard to gauge the integrity of the data. Later, lines 419-421, implies (incorrectly) that the vertical wind is found from the sine of the pitch angle multiplied by the airspeed. It would be much clearer if the equations used to make these estimates and error bounds were provided.

– Lines 293-301: Sensor and signal amplification/conditioning and quantization noises are random processes, so cannot be measured before-hand and subtracted from signals measured later. The noise removal process described is as likely to corrupt the measurement as it is to clean it up. Even if "noise" means biases, these are likely to be variable with temperature, so could not conceivably be obtained from pre-flight sampling.

This would generally be true, except that much of the high frequency noise in the pressure transducer signal is quasi-periodic noise which we believe to be introduced by a switching voltage regulator, rather than more stochastic noise sources described above. In this revision, we have added additional exposition about the noise characteristics as well as evidence that the approach we have devised is successful in reducing the impact of the quasi-periodic noise, and evidence that the noise was independent of atmospheric conditions. Specifically:

1. Excerpts from time series illustrating the noise content.

2. Frequency spectra showing that the frequency content of the transducers in an inactive environment chamber did not change between atmospheric and low pressure/low temperature conditions.

3. Frequency spectra for all altitudes, which shows that the high frequency content of the transducer signal does not change during the flight for frequencies higher than the signal from the five hole probe

4. A comparison of excerpts of the pressure time series taken during flight showing that the filter appears to successfully remove electrical noise from the pressure signal.

5. A comparison of frequency spectra before and after filtering showing that the high frequency content is reduced, and that the pressure signal more accurately reproduces a -5/3 slope in the square-root of the corresponding dynamic pressure signal (i.e. in the resulting velocity signal).

Please also note that this noise filtering approach was added in response to a previous comment that stated 'Not clear how (of if) the noise floor/noise figure is removed in the qualified data fits. The noise floor can corrupt the estimated slope, so its removal can often retrieve epsilon values that were previously rejected. This could help to explain the absence of qualified values at the higher altitudes,' As the noise floor in the wind velocity was a function of numerous inputs, we could not devise a hands-free, unbiased, approach to consistently identify or remove it from the wind estimate. Therefore, we introduced this filtering approach to at least remove some of the noise introduced by the pressure transducers. We found that its impact on the statistics presented in the remainder of the paper was negligible.

The noise shown in Figure 4(a) does indeed appear to have a set of spikes, with a repetition frequency of about 1 Hz. The fact that there is no prominent, persistent spectral component near 1 Hz (Fig 4(b), but the noise spectrum is very broad, suggests that these spikes are not regular. Read: unpredictable in amplitude and phase, even on a short time scale. Hence subtraction of a pre-recorded signal, in the time domain or frequency domain, cannot be expected to remove noise in measured signals. To the extent that this subtraction reduces the spectral component amplitudes, it can be expected to reduce signal variation in the time domain, and appear "filtered", but both signal and noise are affected equally. At frequencies where the signal component amplitudes are large, the subtraction of a small noise changes the signal very little. But at frequencies where the component amplitudes are not large relative to the noise, then both signal and noise are altered greatly, sometimes reducing and sometimes increasing the component magnitudes, depending on the relative phase. Because of this random effect, it would be better to only accept spectral components for analysis that are significantly larger than the noise spectrum, e.g. by 1 order of magnitude. All others are corrupted by noise, and doubly corrupted by the filtering technique proffered here. A test for this corruption would be to compare time domain signals pointwise before and after filtering at a scale where the frequencies of interest (e.g. up to 20 Hz) can be seen. The spectra of Fig 4(f) show another way to assess this, but it would be helpful to show the noise spectrum, too. If the noise spectrum is well below the signal spectrum up to where the inertial cascade is fit, then this indicates that that spectrum should not be unduly influenced by noise, or the filtering method. But how representative is this one case? Otherwise, it is very difficult to believe that the resulting modified signals are reasonably free of artifacts, since the basic filtering approach does not make good sense. Another approach would be to remove the spikes (if they are the main problem) by an outlier removal method in the time domain. Lastly, the hypothesis that this noise is due to the (switching) power supply seems unlikely, given that these switch at hundreds of kilohertz, and, due to the filtering inherent in the technique, produce supply line noise primarily at the fundamental switching frequency. So, even in the presence of aliasing, this would not be expected to produce such spiky noise. Are there high-current loads in the avionics that have narrow pulses at about 1 Hz, e.g. telemetry transmissions?

– Lines 340-341: "uncertainty in wind magnitude was found to be most dependent on the yaw angle" How? Why? Do the wind excursions in Figure 4 correlate with yaw angle?

The uncertainty dependence of wind magnitude on aircraft attitude is introduced into the wind estimate during the coordinate transformation between body-fixed and inertial coordinate systems. During flight, the yaw angle can vary from 0 to 360 degrees, whereas the sideslip, angle of attack, pitch and roll angles are near zero. The result is that the greatest contribution to the $u/v$ wind components during transformation is the true airspeed being multiplied by the sine/cosine of yaw and cosine of the pitch. The result is that during an orbit, the horizontal velocity components will have high sensitivity to yaw error at both $0°$ degrees and $90°$. For a similar reason, the vertical component of velocity is most sensitive to error in pitch.

This argument oversimplifies the wind estimation problem. The vector conversion of relative wind from body to inertial coordinates is equally sensitive to yaw errors at all yaw angles. It is the vector combination of relative wind and inertial velocity in the wind triangle that produces sensitivity variation on the orbit, since when the plane has the highest inertial velocity (flying downwind), its rate of change of attitude is greatest on the circle, and this can expose errors due to timing mismatches in the various sensors. The paper uses a complicated timing recovery scheme, since the data is not time stamped with a common reference, so this may be where to look for periodic wind estimate excursions. Do they occur around this point on the circle? Or do they occur where the orbit nears the jet? Or are they at 0 and 90 deg? (I can't tell from the paper).

As illustrated by the uncertainty bounds shown in Figure 4, the wind excursions cannot be explained by error in yaw. Note also that prior to preparation of this manuscript, we conducted an informal perturbation study to determine if these excursions could be explained by measurement error and found that for them to be removed from the wind profile, the yaw error would not only have to be non-monotonically dependent on altitude (i.e. not attributable to sensor drift), but it would have to be so high that it could only be explained by failure of the system.
I think you mean Figure 5. Your uncertainty analysis only models the random forms of error, not the systematic ones such as that mentioned in the response above. So it is premature to say that these are covered by the doubling mentioned in the text. The real test would be to plot the winds as a function of azimuth on the circle, as well as noting the mean wind direction on each circle and the location of the jet, to see if aircraft motions are correlated with the measurements.

We have added the first statement to the manuscript, the second statement is left out due to the informal nature of the perturbation analysis that was conducted (the uncertainty analysis presented in the manuscript being the more rigorous

approach).

More rigorous, yes, given the assumptions, but perhaps leaving out the largest effects!

– Lines 400-401: "and we assume that the characteristics of the atmosphere within these segments are horizontally homogeneous (i.e. they are a function of z only)". A very loaded assumption to make with no justification!

This statement was not thought to be as loaded as as the reviewer is implying, due existing consensus that the intrinsic stability in the stratosphere and upper troposphere will promote horizontal homogeneity (as exemplified by one of the other reviewers being insistent that variability in the measured profiles must be due to measurement error or vertical stratification rather than any horizontal heterogeneity). As the aircraft approaches the surface and the radius of the orbits becomes smaller, this assumption also becomes less restrictive.

We have added our rationale for this assumption in the revised manuscript.

"Promote horizontal homogeneity" does not mean that the layer structures are perfectly horizontal and homogeneous over arbitrary distances. Does this extend over hundreds of m, km, tens of km? Gravity wave activity, for one, can upset this ideal situation, as intimated later from the data in lines 537-539, possibly contradicting this assumption. Also, nearer the surface, does the orbit radius decrease faster than the scale of horizontal variations? Basically, there is still no quantitative support for this assumption, and it should not be treated as common or obvious.

– Lines 440-441: "the HiDRON measurements do contain short wavelength fluctuations" How short? Please quantify.

We have added a statement that the wavelengths we are referring to are the ones on the order of 1 km or less.

– Line 444: "These low frequency waves may be bias in the wind estimate introduced by the orbital path". Indeed, it seems like they may correspond to one turn on the helix, so may be an anomaly in the wind retrieval. Comparison to the vertical period of the helix would be important here.

We have added some quantification and discussion of these values to the text.

The only "quantification" I see is the statement "that the periodicity is shorter than the pitch of the helical flight path". How much shorter? Does this hold for all orbits?

– Line 464: "backing with altitude"?

Backing winds are winds that change direction counterclockwise with height. We have revised the statement to remove the use of backing and simply state the wind direction changes.

– Line 470: Should note that an accurate assessment of TKE also depends on capturing the lowest wavenumber components in the inertial subrange, since they contain the largest energy per wavenumber. Did the PSDs over the statistical intervals exhibit a roll-off or flattening of the $f^{-5/3}$ slope at low wavenumbers, indicating the outer scales were captured?

A note to this effect has been added to the revised manuscript. The low frequency content more often increased, rather than rolled off, as described in the wavelet analysis, and discussion of the time series. This was also the justification for the selected segment length.

– Line 473: "over a specified frequency, $f$, range". What was the range?

The frequency range determination process was described in the manuscript shortly after this statement. We have revised the text by removal of the statement above so that the question of frequency range does not come up until it is actually described.

– Line 478: "which will reach a minimum at the frequency where the noise has a greater contribution to the integration than the signal". Why?

Compensated, or "pre-multiplied," spectra are commonly employed in turbulent boundary layer studies because they facilitate the visualization of energy spectra on semi-logarithmic axes. Specifically because they allow for a clearer representation of the frequency/wavenumber dependence of the relative contribution of each frequency/wavenumber to the overall energy content. This is because, for example,

$$d\langle u^2 \rangle = F_{uu}df = fF_{uu}d(\log f). \tag{1}$$

Hence when $f\,Fd(\log f)$ begins to increase on a semi-logarithmic plot at high frequencies, this indicates a frequency range where the energy content increases with $f$. Given that universal equilibrium range turbulence will decrease in energy content with $f$, the minimum in $f\,Fd(\log f)$ indicates a frequency at which the noise begins to have a greater contribution to the variance than the turbulence content.

However, the use of the compensated spectrum is a procedural detail and in hindsight distracts from the overall point of this processing step, which is to determine at frequency the contribution to overall variance increases with increasing $f$, rather than decreases. The same result could have been achieved using the un-compensated frequency spectrum, and we only used the compensated spectrum to simplify visualizing the frequency-dependence of $\langle u^2 \rangle$, $\langle v^2 \rangle$ and $\langle w^2 \rangle$.

As including the detailed exposition above would distract from this point, while adding little value to the overall intent of its inclusion, we have simplified the discussion to simply state that the upper bound of this range was determined by identifying the frequency where the noise has a greater contribution to the integration than the signal than the velocity fluctuations.

I thought the explanation above was fine. But why make the detail vague by leaving it out?

– Line 478-479: "filter frequency was consistent with the probe's frequency response in the boundary layer and varied between 1 Hz and 20 Hz above the boundary layer". Does "filter frequency" correspond with the "frequency range" in line 473? By "probe's frequency response" hear do you mean "probe bandwidth"? How did this vary above the boundary layer?

By filter frequency, we meant the upper bound of the range described in the previous statement. We have modified 'filter frequency' to read upper bound of the frequency range.

Yes we do mean probe's bandwidth. These terms are interchangeable and, as we use frequency response at numerous points throughout the paper prior to using it in this sentence, would prefer not to change it to 'probe bandwidth'. We have clarified that we are referring to the maximum frequency response, as frequency response in general can also refer to the Bode plot of the probe's response to excitation.

A frequency response (amplitude curve as a function of frequency) does not have a "maximum frequency of response". This is a smoothly varying function with no lower bound, so some standard point on this curve is picked to indicate "bandwidth" of response. This is typically the -3dB roll-off frequency. This bandwidth (particularly if you say 3dB bandwidth) would be widely understood, whereas your "maximum frequency response" could be confused with other things, such as the frequency where the response is maximum.

As noted in the text, the frequency at which the noise exceeded the signal varied between 1 Hz and 20 Hz. We do not describe a trend above the boundary layer since no trends were evident, being dependent on the presence of low-frequency energy content. We have altered the text to now read 'with the higher upper frequency bounds corresponding to instances where there was increased low frequency content in $F_{uu}$, $F_{vv}$ and $F_{ww}$.'

The frequency where the noise exceeds the signal is entirely dependent on the turbulent energy in the flow, so no trend (say with altitude) would be expected.

– Line 514-516: "due to the statistical segment length used for averaging, the value of $\langle k \rangle$ will be biased to wavelengths smaller than the statistical segment length" Why? Do you mean smaller than the outer scale (as noted for Line 470 above)? Is this what "and therefore may not completely describe the actual energy content of the turbulence" is alluding to? This could be said much more clearly.

That is exactly what we meant. Due to the segment size of $\sim 3$ km we cannot resolve wavenumber content larger than the length of the statistical segment (or frequency content below $O(0.01)$ Hz). The point of the above statement is to note that this is insufficient to capture any sort of outer scale/low frequency energy contribution below of wavelengths longer than the segment length. We have changed this statement to note that, in addition to the implicit assumptions made when calculating $\langle \varepsilon \rangle$, the method used to calculate $\langle k \rangle$ reflects only the energy content corresponding wavelengths smaller than the statistical segment length (or frequencies higher than the the inverse of the time taken to traverse that segment length).

– Figure 10: It would be helpful to draw a $\langle k \rangle^{3/2}$ line on the plot for reference.

This line has been added.

– Line 524: "Above the boundary layer turbulence $\langle k \rangle$ and $\langle EDR \rangle$ are largely in agreement" This is hard so see, since $\langle k \rangle$ should be proportional to $\langle EDR \rangle^2$ but Figure 11 compares $\langle k \rangle$ to $\langle EDR \rangle$.

We have changed this figure and corresponding references in the text to $\langle EDR \rangle^2$

– Line 535: "Nyquist frequency of the minimum probe response". No idea what this means. Nyquist relates to the sampling frequency, not the frequency response.

We have changed this sentence to refer half the maximum frequency response of the probe.

– Lines 538-540: "Noticeable in Figs. 11b, d, and f is the significant long wavelength content for $\kappa \ell < 0.003$ (wavelengths larger than 2 km) when $z > 10$ km." I don't see this. It would help to show a color bar. Looks to me like there is significant long wavelength content below .0003 rad/m over all altitudes. And why does the plot have a curved boundary on the left and a straight boundary on the right? Seems that the "time" variable discussed in the wavelet transform is really altitude z here. Correct? This whole discussion is very terse for readers unfamiliar with wavelets.

The long wavelength content for $\kappa \ell < 0.003$ when $z > 10$ km is more evident when plotting the wavelet coefficient on non-logarithmic contours (which does a poor job of visualizing the short wavelength distribution of the coefficient), or when using a non-colorblind friendly colormap.

We have modified this figure by changing the colormap, added a colorbar, and adjusted the horizontal axis to better constrain the wavenumbers to those less than an orbit. However, the difference below 10 km and above 10 km is still subtle, therefore we have clarified within the text that the contributions for altitudes greater than 10 km are most noticeable for Flights 2 and 3

The wavelet transform is calculated in the time/frequency domain, but since altitude is a function of time (which we had tried to indicate by referring to $z(t)$), it is then plotted as a function of altitude. We have rephrased this sentence to be more clear.

The curved boundary on the left of the figure is due to the time-frequency nature of the wavelet transform. At the start and end of the time series, there is insufficient information to resolve the low frequency content. However, towards the central part of the time series, the maximum low frequency content can be resolved, resulting in the curved boundary on the left (low frequency) end of the figure. The right of the figure, representing the high frequency content, is not subject to such resolution issues and therefore has a straighter boundary (although in this presentation, since the frequency has been transformed to wavenumber using Taylor's hypothesis, the highest wavenumber resolved is a function of the airspeed, which decreases with altitude, resulting in the slanted boundary on the right of the figure.)

We have updated the text to provide more description of the wavelet transform and its features.

– Lines 541-547: What are the implications of these observations from the wavelet transform? Why is the frequency content behind turbulent parameterization important? Usually this is constrained by the inertial cascade.

Our primary rationalization for examining the low frequency content is that it helps us to explain the differences between $k$ and $EDR$, highlighting that $EDR$ does not capture the low wavenumber content. This analysis also helps us to justify and understand the use of the turbulent kinetic energy estimate using lower frequency energy content than used for the initial $\langle k \rangle$ estimate.

We have modified this paragraph to better highlight some of the above points.

– Lines 557-558: "and therefore is attributed to increased atmospheric absorption due to the increase in molecular mean free path with altitude". Seems this could also be due to the decrease in coupling coefficient to the microphone diaphragm due to lower density.

It is not clear to us how the density can affect the efficiency of conversion of mechanical energy to electrical energy. Assuming the intention was to describe the decrease in mechanical forcing which could be expected due to lower density, we then would expect the variance of pressure measured by the microphone to scale with $P^2$, rather than $P$. However, we found that it does not scale with $P^2$. In addition to various other normalizations, we also tried scaling the variance of the microphone signal with $\rho c$ (corresponding to the expected change in sound intensity with altitude), which was also was unsuccessful. The only scaling we found that provided some success was the normalization of the the variance of microphone signal with $P$, as noted in the paper, and this result was consistent with the expected attenuation due to increase in mean free path, following the discussion presented in the cited reference (Bass 2007).

– Line 562: "The altitude attenuation will be dependent on the local temperature as well". Why?

In the paragraph before the referenced statement we had attributed the altitude dependence to the mean free path, which is a function of pressure and temperature. We have added this statement explicitly to the above sentence.

Lines 565-566: "the resulting infrasonic amplitude profile can be observed to strongly correlate with [k]". Depends what you mean by "correlate". The variations in $[k]$ are not correlated with the normalized infrasound variance, only the large scale means seem to correlate. The infrasound signal looks like a LP filtered version of the TKE. Seems the infrasound signal (even filtered at 20Hz) should be able to follow the variations in $[k]$ that occur over km of vertical intervals. Why does it not?

The use of the word correlate to indicate correspondence was a poor choice and has been changed. The $[k]$ measurement is an in-situ measurement, whereas the infrasonic microphone is a remote measurement. Therefore there is no reason to expect exact correlation between the two measurement approaches. Indeed, at least ideally, the infrasonic microphone will detect the turbulence before the aircraft enters it, acting as a filter to the 'spikiness' of the $[k]$ measured by the in-situ sensor. Not only that, but the microphone will also detect any sound generated by nearby turbulent patches that the aircraft does not fly through. The net result can thus be expected to be a low-pass version of the $[k]$ profile, as the sound propagation will be be omnidirectional and emitted from numerous locations, whereas the $[k]$ can be expected to be constrained to stratified vertical layers.

We have added these discussion points to the revised manuscript.

– Line 569: Do you mean $[\sigma]_{LF}$ instead of $[\sigma_f]$? Also, I don't understand why these two would increase at the same rate if high frequency energy is primarily increasing, as supposed as being "most likely".

Yes, we meant $[\sigma]_{LF}$ and have fixed this typo.

Rate is probably the wrong word for what we are trying to describe. We have reworded this statement to try to more clearly describe our rationale for why the ratio $[\sigma^2]_{LF}/[\sigma^2]$ will remain constant in the boundary layer.

Lines 713-715: garbled revised sentence.

– Line 575: "This is due to the helical flight path". It is really due to the small flight path angle. A steep helix would not be as susceptible to horizontal gradients.

We were referring to the specific helical flight path flown during these flights, not helical flight paths in general. We have reworded the statement to 'This is due to the shallow glide slope of the particular helical flight path flown in these experiments'.

– Line 585: "These values were then re-interpolated to each statistical segment." Don't know what this means, exactly. Averaged for each segment? Interpolated how? And why were the 200m smoothed data averaged again (binned) at 100m intervals?

In this context, the smoothing is applied as a low-pass filter, and therefore the time-series post-smoothing has the same number of data points as pre-smoothed. The bin averaging over 100 m intervals was done to facilitate the differencing across the 100 m interval used for gradient calculation (i.e. equivalent to downsampling the signal). In the past we calculated a difference across $\Delta z = 100$ m for each point in the time series, but a different reviewer took exception to this approach in a previous revision of this manuscript, stating it obscures the vertical resolution of the differencing. Finally, since the $\Delta z = 100$ m downsampled data points at which gradients were calculated do not coincide with the $z$ locations of the statistical segments over which $\langle \theta_v \rangle$ was calculated, we had to interpolate the gradients to locations of the statistical segments. It is a messy process, but seems to be necessary to achieve $Ri$ profiles similar to what is observed from radiosonde measurements.

– Line 601-602: "effectively reproduces the N2 profiles calculated along the flight path". Only if by "effectively" you mean a highly smoothed version of N2.

Yes, we meant that (particularly when compared to the $S^2$ profiles) this approach produced a highly smoothed version of the $N^2$ profile determined with other approach. We have reworded this statement to 'effectively reproduces the trend of the $N^2$ vertical profiles calculated along the flight path.'

– Line 617: "Wind profiles were in good agreement with the available National Weather Service radiosonde profiles". Please quantify "good". Likewise, quantify "best comparison" in the next sentence.

We have added a discussion quantifying the difference in the body of the paper while discussing the wind profiles and summarized this in the last section.

– Lines 619-620: "over a large horizontal wavelength range with high vertical resolution". Confusing/misleading. Implies that horizontal and vertical effects can be separated, or that vertical variations dominate horizontal ones. Statement should be qualified accordingly (as was done in the body).

We have reworded this statement to 'over a large horizontal distance relative to the vertical distance traveled.'

– Line 637: "fror example". Typo.

This typo has been fixed.

– Line 643: "Additional flight patterns can also be designed with tighter helical descent can be designed". Awkward.

Reworded to 'Additional flight patterns can also be designed with tighter helical descent'

– Appendix A: "all transducers" is vague. Wind retrieval is a combination of many transducers, including relative wind, attitude, and inertial velocity. It would be clearer to use the more specific nomenclature from the body of the paper.

We have removed the line legend and reference the line colors in the figure caption. To use the nomenclature from the body of the paper would have required shrinking the font size below acceptable limits (at least for the graphing software that we used when generating the bulk of the figures used in the paper).

– Line 601-602: "effectively reproduces the N2 profiles calculated along the flight path". Only if by "effectively" you mean a highly smoothed version of N2.

Yes, we meant that (particularly when compared to the $S^2$ profiles) this approach produced a highly smoothed version of the $N^2$ profile determined with other approach. We have reworded this statement to 'effectively reproduces the trend of the $N^2$ vertical profiles calculated along the flight path.'

– Line 617: "Wind profiles were in good agreement with the available National Weather Service radiosonde profiles". Please quantify "good". Likewise, quantify "best comparison" in the next sentence.

We have added a discussion quantifying the difference in the body of the paper while discussing the wind profiles and summarized this in the last section.

– Lines 619-620: "over a large horizontal wavelength range with high vertical resolution". Confusing/misleading. Implies that horizontal and vertical effects can be separated, or that vertical variations dominate horizontal ones. Statement should be qualified accordingly (as was done in the body).

We have reworded this statement to 'over a large horizontal distance relative to the vertical distance traveled.'

– Line 637: "fror example". Typo.

This typo has been fixed.

– Line 643: "Additional flight patterns can also be designed with tighter helical descent can be designed". Awkward.

Reworded to 'Additional flight patterns can also be designed with tighter helical descent'

– Appendix A: "all transducers" is vague. Wind retrieval is a combination of many transducers, including relative wind, attitude, and inertial velocity. It would be clearer to use the more specific nomenclature from the body of the paper.

We have removed the line legend and reference the line colors in the figure caption. To use the nomenclature from the body of the paper would have required shrinking the font size below acceptable limits (at least for the graphing software that we used when generating the bulk of the figures used in the paper).

---

## Author Response (AR2)

**Response to Referee 1**

We appreciate the time taken by the referees to once again read our manuscript and provide comments and suggestions for its improvement. We have made additional revisions to the manuscript and to help clarify where changes have been made in the revised version, we have highlighted all changes made using blue text.

The only major revisions made to the current draft were: (1) the addition of an uncertainty analysis section; (2) the addition of velocity measured by the onboard GPS during balloon ascent, interpreted as a measurement of wind velocity, with associated discussion of the resulting comparison; (3) removing the moving average step during calculation of vertical gradients and presenting $N$ and $S$ instead of $N^2$ and $S^2$ to provide a presentation of these quantities which may be more familiar to readers; and (4) removal of the comparison of $Ri$ and $EDR$ as the comparison did not provide much value and the authors felt the need to reduce the length of the paper after adding an additional 9 pages of content since the original submission. Most of the remaining changes were minor textual changes to better highlight and clarify certain points brought up by the referees.

Below, we respond to the individual comments made by the referee. To do so, we have reproduced the original review, with our comments provided in blue text.

Haghighi et al. present an exciting new measurement system that allows to sample the atmosphere up to 25 km with high resolution. The manuscript has been significantly improved. The plots look more trustworthy, but some questions remain, especially with regards to the basic wind estimation.

Comments:

– I lost confidence in the calibration of the wind measurement system. When you write that yaw and pitch alignment were corrected now, I wonder how this was done.

As noted in the manuscript the "additional estimate and correction of the misalignment of the probe axis and aircraft's body-frame axis was conducted following the approach described in" Al-Ghussain and Bailey, (2021) Atmospheric Measurement Techniques. Given that this is a somewhat lengthy procedure, we preferred to just reference the previously peer-reviewed and published work rather than reproduce the details here. Note that this technique is essentially an extension of the flight calibration approach described by van den Kroonenberg et al, but is instead applied a posteriori using a slightly different optimization approach.

I assumed this had been done even before the first manuscript.

Unfortunately, we did not to conduct these corrections initially following the flight campaign as the approach had only been recently published and the codes had not yet been generalized for datasets other than the ones used in the paper. Although we had intended to apply these corrections, focus of subsequent work was on data analysis and we did not revisit determination of the velocity vector until receiving reviewer feedback from the original submission.

Have Lenschow maneuvers for in-flight calibration been done?

Typical in-flight calibration maneuvers require flight at a constant altitude, which was not possible with this aircraft as it is a glider.

Have not only the misalignment angles but also the airspeed factor been determined? This is essential.

The Al-Ghussain and Bailey correction approach also accounts for influence of aircraft blockage effects on the airspeed measurement. We have added mention of this in the revised paper. Note that for other studies (using powered sUAS) we have compared the in-flight maneuver calibration approach to the Al-Ghussain and Bailey approach and found little difference between the corrected results (although this comparison was only used for internal validation and not published).

– Please show some verification that all the angles and airspeed measurements are correct now, including static pressure.

Although we had initially discarded all data taken during ascent as the aircraft sensors (five-hole-probe, humidity, and temperature) were not configured to perform measurements on ascent, we have since realized that an estimate of the horizontal wind speed can be provided by the ground speed of the aircraft during the balloon ascent. We have included

45   these wind speed and direction profiles for additional validation of the five-hole-probe measurements made during descent and the results will hopefully restore the referee's confidence in the five-hole-probe measurements. Note that we do not independently measure static pressure, except via the aircraft's altimeter. However, we have compared the airspeed produced by treating the five-hole-probe as a Pitot-static probe to that measured by the aircraft's Pitot-static probe and the comparison was favorable.

50   – An uncertainty propagation for the wind algorithm with flow probes was developed by van den Kroonenberg et al. (2008) and could be used to identify the uncertainties. It would be good to include the uncertainty in some of the plots.

   We have added a new section on uncertainty analysis to the revised manuscript, in it we implement a Monte-Carlo-based estimate for uncertainty propagation which is very similar conceptually to the approach used by van den Kroonenberg et al. (2008) but by implementing it as a Monte-Carlo analysis we do not rely on a single reference state and can also
55   include the effects of coupling between different error sources.

   – Appendix A, Fig. A1: The NOAA maps are not readable at the given resolution.

   We have updated this figure with a higher resolution version.

**Response to Referee 2**

We appreciate the time taken by the referees to once again read our manuscript and provide comments and suggestions for its improvement. We have made additional revisions to the manuscript and to help clarify where changes have been made in the revised version, we have highlighted all changes made using blue text.

The only major revisions made to the current draft were: (1) the addition of an uncertainty analysis section; (2) the addition of velocity measured by the onboard GPS during balloon ascent, interpreted as a measurement number of wind velocity, with associated discussion of the resulting comparison; (3) removing the moving average step during calculation of vertical gradients and presenting $N$ and $S$ instead of $N^2$ and $S^2$ to provide a presentation of these quantities which may be more familiar to readers; and (4) removal of the comparison of $Ri$ and $EDR$ as the comparison did not provide much value and the authors felt the need to reduce the length of the paper after adding an additional 9 pages of content since the original submission. Most of the remaining changes were minor textual changes to better highlight and clarify certain points brought up by the referees.

Below, we respond to the individual comments made by the referee. To do so, we have reproduced the original review, with our comments provided in blue text.

The authors have thoroughly revised their manuscript but there are still many shortcomings. Some additional information is unclear and not convincing. I think that the revised version cannot be published in its current form. In this second review, I focus on the main issues only.

1. As outlined in the first review, the manuscript hesitates between the evaluation of a new technique and a scientific work on atmospheric processes, which makes the paper very unclear. The primary objective of the paper should be to provide all the information needed to give the reader a clear idea of the platform's strengths and weaknesses. It is not the case. For example, section 3.1 shows the comparisons between temperature, humidity and wind profiles obtained by Hidron and radiosondes. The differences and similarities between the two should be emphasized. Paragraph 4, page 14 (lines 354-364) is out of the topic, because it describes the temperature structure of the troposphere and stratosphere without considering the differences/consistencies between the UAV and radiosonde profiles. In addition, it is not clear why this description is useful, since it is not used afterwards.

The intent with the original structure of the manuscript was to not only describe the measurement technique, but also illustrate some of the statistical analysis which can be conducted with the data from the measurement technique, which we believe to be a strength of the platform. We also note that in the section of text the reviewer is referring to includes "With the exception of $\langle RH \rangle$, the HiDRON H2 measurements are broadly consistent with the trends and values indicated by the radiosonde data, although the HiDRON H2 temperature shows a noticeable warm bias compared to the radiosonde above $z = 16$ km that appears most noticeable for Flight 3. Note that no density correction was applied to this sensor measurement to account for the reduced convective heat transfer at these altitudes." which we would say addresses the point of comparison between the HiDRON and radiosonde measurements and identfication of weaknesses in the platform which needs to be addressed in future flights.

As the focus of the paper is intended to be on the turbulence statistics calculated from the HiDRON, the section being referenced had originally been written with the intent to provide a general atmospheric conditions during the measurements, including stability conditions, as reflected in the lapse rate and Richardson number and mean wind profiles, order to provide the reader with context for the turbulence statistics discussed in the later section. However, the revisions requested by the previous round of reviews has somewhat muddled this structure, as comments suggested that we add additional comparisons to the temperature and humidity profiles, move certain points of discussion, and add additional details and statistical metrics, the result being that the description of atmospheric conditions interpreted from the temperature profiles highlighted by the referee now appears out of context and the original narrative structure has become disordered. We have removed this now 'out of topic' paragraph in the revised manuscript but understand that this is only

intended as an example of the reviewers disagreement with the organization of the manuscript. We have made some minor edits throughout the manuscript to try to restore, and better explain, the original narrative structure.

2. The description of Figure 7, page 17, focuses on the main trend of the wind profiles and on the presence of the jet-stream during flight 1, detected by both instruments. However, the authors do not discuss the "high-frequency"  fluctuations revealed by Hydron measurements, even though they are the most striking pattern of the three profiles (these fluctuations do not appear in the radiosonde data but their vertical resolution may be insufficient). Are they real or artefacts due to the helical pattern of the Hydron path, for example? Can they result from inertia gravity waves? Is a spectral analysis consistent with IGW? These questions, not addressed by the authors, are of prime importance, as their answer obviously conditions the quality of shear and Richardson number estimates.

We note several things here. First, the 'high frequency fluctuations' were initially intended to be addressed by Section 3.5, where we had pointed out that "When the measurements are examined in this manner, some of the scatter in the vertical profiles was be found to be associated with horizontal heterogeneity in the measured statistics". Again, this connection may have become lost in the revised manuscript due to additional content inserted at the request of the referees. Second, these fluctuations are not evident in the radiosonde data simply because of the lack of vertical resolution in the radiosonde wind profiles we were able to obtain. We tried to find higher resolution publicly-available data for this region and this time of day, but were unable to do so. However, although we had initially discarded all data taken during ascent as the aircraft sensors (five-hole-probe, humidity, and temperature) were not configured to perform measurements on ascent, we have since realized that an estimate of the horizontal wind speed can be provided by the GPS-measured ground speed of the aircraft during ascent. This result shows the same 'high frequency fluctuations' as measured by the five-hole probe on descent, and we have added a corresponding discussion of the oscillations observed in the profiles, which also notes that similar behavior is commonly observed in high resolution radiosonde launches. In light of these observations, we feel that adding additional gravity wave analysis would not add any value to the paper, while significantly increasing its length.

3. The newly introduced paragraph on stability conditions is unclear and not convincing. First, the text does not describe the red profiles (N2). We implicitly understand that they are obtained from the second method (described from line 407). The "noisy and spike" structure of N2 profiles and the large differences between the wind shear estimated from the two methods are suspect. Figure 9 shows Ri profiles for one method only (likely the method corresponding to the red profiles in Figure 8). Unfortunately, they are clearly not physical, as no results in the literature show such a stable atmosphere at a vertical resolution of 100 meters (or even lower). The authors should check the literature and their calculation methods.

First, we would like to point out that the profiles in Figures 8 and 9 are identified as $N_t^2$ (black) and $N^2$ (red), and $S_t^2$ (black) and $S^2$ (red) in the legend. This may not have been clear as a quirk in our graphing program results in the legend text needing to be smaller than desireable to position the legend within the whitespace of the figure. We have updated the legends with larger text where possible. We do note that the difference between $N_t^2$ and $N^2$ was provided in the text with the differences between the two being the method of with "The profiles of $N^2$ and $S^2$ calculated in this manner (using central differencing between time-adjacent statistical segments) are referred to as $N_t^2$ and $S_t^2$" where the subscripted $t$ is intended to indicate that the calculation is conducted using differences in time. The profiles of $N_z$, and $S_z$ on Figures 8 and 9 are calculated using a technique whereby we calculate the vertical gradients using the data points which were the 'nearest neighbors' in the vertical direction. Hence providing a more true estimate of the local vertical gradients of wind and temperature. In the revised manuscript, we have updated the text to use $N_z^2$, $S_z^2$ and $Ri_z$ to avoid further confusion.

We have also done this as the distinction between calculation approaches is an important one since, as the HiDRON is a glider, for every 1 km of vertical descent there is roughly 15 km of horizontal travel. This may not have been clear from the trajectory illustrations provided in Figure 6, as the vertical axis in km was not to scale with the longitude and latitude on the horizontal axes, thus causing the orbits to appear 'tighter' than they were in reality. We have revised Figure 6 with the km axis to scale with the latitude/longitude to provide a more accurate 3D rendition of the flight profile. We have also made several small revisions in the text to highlight the same point.

We feel the distinction in these calculation approaches is relevant as horizontal heterogeneity can can play an outsized impact in the calculation of $N^2$, $S^2$ and $Ri$ and the spiralling flight path will be much more sensitive to horizontal heterogeneity than measurement approaches designed to only measure the vertical profile. One downside of this approach is that, due to the size of the orbit, the vertical resolution is of the order of this calculation is of the order of the vertical distance between orbits, roughly 1.5 km. This is why there is an appearance of 'clearly not physical result' as the reviewer is misinterpreting the resolution as being much higher than it is. This misinterpretation is likely because the vertical gradient can be calculated for every 1500 m long (mostly horizontal) statistical window, which results the appearance of higher vertical resolution when plotted as a vertical profile. This is one of the reasons we believe that these measurements are better described using the approach provided in Section 3.5.

We apologize that these discussion points were unclear in the revised manuscript, and have made further revisions to try to make the distinction between these approaches, and the reasoning behind implementing them, and their implications, more clear. Finally, we have removed the moving average step in the calculation of $S_t^2$, $N_t^2$ and $Ri_t$ and plot the results in terms of $S$ and $N$ instead of $N^2$ and $S^2$ to attempt to better bring the results of these calculations more in line with the referee's expectations.

4. The expression of $\varepsilon$ (14), line 492 does not seem to be correct because it is equivalent to write $E_{ll}(k_l) = A$. As a consequence (?), there are inconsistencies in Figure 10. There are no substantial differences between TKE measured during flights 1, 2 and 3 (e.g. ∼0.1 m$^2$ s$^{-2}$ around 20 km). However, EDR(flight 1) is about ∼5 times EDR(flight 2 and 3), i.e. $\varepsilon$ (flight 1) is about $10^2$ larger. This is not consistent with TKE. The EDR profiles show an "envelope" that should correspond to the instrumental noise floor (apparently confirmed by a spectral slope close to 0 in figure 14e). The noise level was likely much more important during flight 1. During the present jet-stream, flight 1 should be more prone to turbulence, whereas Fig 10d seems to indicate the virtual absence of inertial turbulence. It is counter-intuitive and then suspect.

There was indeed a typo in equation (14) and have corrected it in the revised manuscript. We have verified that the calculation itself was correct. As we mention in the manuscript "Note that the approach used is only an approximation of $\varepsilon$ as the inertial subrange only rarely follows the -5/3 slope, hence it will provide a non-zero value even if no turbulence is present, and therefore some caution is required when interpreting these $EDR$ profiles beyond being a qualitative indication of the presence of turbulence in the form of localized regions of relatively high $EDR$. " We generally agree with the referee's remaining assessment, which is why we had introduced the spectral slope an additional metric for the identification of regions of turbulence. We note that, as this is an in-situ approach we can expect the sensors to experience increased noise due to the impact of ionizing radiation on the sensor's electronic components.

As for the influence (or lack thereof) of the jet stream on turbulence production, we admit that we were also surprised by this result. We do note however that the time scales of the shear associated with the mean shear of the jet stream ($\sim 13000m/25\ m\ s^{-1} \approx 500\ s$) corresponds to a square shear frequency on the order of $10^{-6}$ which is comparable to what was measured for all three flights with $S_z^2$ over vertical scales of 1.5 km and lower than that measured by $S_t^2$ at much smaller spatial scales. Hence we would hypothesize that the influence of the jet-stream was not strong enough at our measurement location to produce additional shear production, as the jet steam was centered over El Paso (EPZ), 150 km to the south (see Appendix A in the manuscript and the EPZ radiosonde profiles provided in Figure 8). Noting also that the air was relatively dry ($RH < 40\%$), and the measured lapse rate less than the dry adiabatic lapse rate, the buoyant-production would also be reduced for Flight 1 when compared to Flights 2 and 3 (as reflected in the lower $N_z^2$ and $N_t^2$ values).

5. The analysis in section 3.5 has been thoroughly revised, and considering the strong reduction of the resolution, this representation may show interesting trends. However, the description of the results are very difficult to follow. In addition, the results depend on the validity of the estimates of N$^2$, S$^2$ Ri and EDR described in the previous sections (e.g. the sudden "jump of S$^2$ above ∼15 km is suspect). The authors conclude that all scenarios can occur (e.g. high turbulence with high Ri, low turbulence with high shear, etc.) but I believe that the reliability of the quantitative results is not high.

We feel that this section is relevant given the glide slope of the aircraft. Particularly as we found that a not insignificant amount of the fluctuations measured in the vertical profiles can be attributed to horizontal heterogeneity, and this becomes apparent given the presentation provided in Section 3.5. The text in that section is largely intended to draw the reader's attention to certain large-scale features that are evident in the visualizations and not intended to make any concrete inferences about the nature of the turbulence. Particularly, as we note again, that much of the disconnect between $Ri_z$, $S_z$, $N_z$ and the turbulence-related quantities is likely due to the significantly reduced vertical resolution of this approach. Although this point had been raised in the conclusions already, we have made several additions to the text to better reiterate it.

---

## Author Response (AR3)

**Response to Referee 2**

We appreciate the time taken by the referees to review our manuscript and provide comments and suggestions for its improvement. We have taken some time to essentially reset and rewrite the manuscript which, although the measurements and results are largely unchanged from our initial submission, has resulted in what we believe to be a much cleaner submission with more direct focus on the measurement technique and much less attention spent on interpretation of the results.

Below, we respond to the individual comments made by the referee on the last version of the manuscript. To do so, we have reproduced the original review, with our comments provided in blue text.

The authors' reply clarified points that helped me to better understand what had been done. The amount of work involved is indisputable, but some questions remain as to the reliability and interpretation of some results. Also, at this level of evaluation, more attention to form would have been appreciated, because there are many avoidable presentation errors. This gives the impression that the manuscript has been hastily revised and not proof-read. For example, there are many typos in the added text, and, e.g., in equation (2). Figure 9 caption does not correspond to the new version of the figure. The manuscript has been improved but, from my point of view, still requires major revisions before publication.

The referee is correct, the previous revisions of this manuscript were hurried, leading to numerous mistakes and inconsistencies that we have done our best to fix in this version.

Comments:

– The wind profiles measured from the onboard GPS during the ascent (now shown in figure 8) are very important added values and provide convincing arguments for the quality of wind profiles measured by Hydron. They provide much more important information than NWS radiosondes (low resolution and large distance). However, they are only used for wind profile comparisons in Figure 8 and the comparison results are not mentioned in the abstract and conclusions. These additional GPS profiles should provide a good reference and can also be used to estimate wind shear at a vertical resolution of ~100 and ~1500 or 2500 m, since they are not affected by the (large) horizontal excursion as the Hydron. The wind comparison result is particularly good above 10 km for flight 3, with wind fluctuations of similar vertical wavelengths and amplitudes. Contrary to the authors' claims, this seems to suggest that the large horizontal excursion of the Hydron (and horizontal inhomogeneity) may not be the dominant factor explaining the shear (and likely N2) fluctuations at the vertical sampling of 100 m from time series. Also, the shear profiles in Figure 9 obtained at the vertical resolution of 100 m and supposed to be strongly "influenced" by the horizontal excursion ( 1500 m) are quite typical of shear profiles estimated from balloon-borne radiosondes at a vertical resolution of 100 m (after applying a low-pass filter with a cut-off at 200 m on 10-m vertical resolution profiles). See the example below from a Vaisala radiosonde. This is another reason why I remain skeptical about the interpretation of the results (lines 448-449) shown in figures 9 and 10, in terms of horizontal inhomogeneity.

Although we still believe that there is some horizontal heterogeneity picked up in these measurements, for the sake of expediency the revised manuscript no longer addresses horizontal heterogeneity (except as a potential complication when interpreting the results).

In addition, if the effective vertical resolutions of $Ri_t$ and $Ri_z$ (and shear) are 100 m and 2500 m, respectively, the large difference between the two (figure 10) can be due to the (vertical) scale dependence of Ri (and shear) (See: Balsley, B. B., Svensson, G., and Tjernström, M.: On the scale dependence of the gradient Richardson number in the residual layer, Bound.-Lay. Meteorol., 127, 57–72, 2008.). Therefore, the difference in fluctuations at the two resolutions cannot be attributed to the (sole) horizontal inhomogeneity.

It turns out that I do not agree with the statement lines 481-485 because the authors refer to the estimates of $Ri_z$, always »1 due to the poor vertical resolution (~2500 m). It is not adapted for the detection of shear instabilities commonly observed in the tropo-stratosphere (~a few hundred meter deep or less).

The fact that the vertical sampling is the same for the two profiles (Figure 9-10) is misleading: it gives the impression that the resolution is the same. The effective resolution should be given by using a clear terminology. For example: $R_{100}$ and $Ri_{2500}$. It should be used everywhere. Line 648 ("... with high Ri...") is ambiguous because the calculated Ri's cannot be directly compared to the thresholds (0.25 or 1).

50      We agree that the large difference in the magnitude of $Ri$ estimates are due to the large differences in vertical resolution. We note that the aircraft measurements will be much more sensitive to the horizontal gradients than to the vertical gradients due to the shallow glide slope, which could be contributing to the fluctuations in $S^2$. However, in the revised version we have trimmed down the section on Richardson number calculation and no longer discuss variability in the results as relating to horizontal heterogeneity.

55      – Lines 410-415: It is not clear what the authors mean. The larger vertical wavelength of the fluctuations should be favorable to larger horizontal scale, and thus to horizontal homogeneity. As a result, the aircraft's orbital trajectory should be less problematic under such conditions. In addition, the argument is weak: the wind profile during flight 2 shows larger amplitude fluctuations during the balloon measurements (ascent) between 5 and 8 km than during the Hydron measurements (descent).

60      'the larger vertical wavelength of the fluctuations should be favorable to larger horizontal scale, and thus to horizontal homogeneity' would require isotropy in the velocity fluctuations, which is not likely at these large scales.

     – In line 409, do the authors mean "...that are NOT evident in the NWS soundings..."?

     This was indeed a typo that should have read 'than are evident in the NWS soundings' instead of 'that are evident in the NWS soundings'.

65      N is shown instead of $N^2$ in figure 9. The profiles contain negative values, which is not possible, because N is necessarily real positive (when $N^2 > 0$) or imaginary (when $N^2 < 0$). Can the authors explain how they calculated N in practice?

     We failed to mention in the previous version that we had preserved the sign of $N^2$ when presenting $N$, i.e. $N = \text{sgn}(N^2)\sqrt{|N^2|}$. To avoid this ambiguity in the current revision we have reverted to presenting $N^2$ and $S^2$ instead of $N$ and $S$.

70      – Lines 483-484 are not clear.

     The revised manuscript no longer includes this statement

     – Line 505: "...caused by inertial turbulence...".

     This depends on your definition of turbulence. For example, laminar flows can certainly provide non-zero values of $k$ as typically defined (e.g. laminar vortex shedding, laminar waves, low speed velocity fluctuations introduced by changes

75      in boundary conditions, etc. will all produce oscillations/unsteadiness about the mean value that can produce $k > 0$). If one defines *all* unsteadiness as being turbulence, then the suggested revision would be correct. However, this is not true in the strictest sense and the original statement that "elevated values of $k$ may not necessarily correspond to velocity fluctuations caused by turbulence." is more general and therefore we elected to keep it in place.

     – Line 542: n must be equal to -5/3.

80      Only if we are assuming all locations contain inertial turbulence. As was stated later, the fitted value of $n$ being close to -5/3 ($\pm 10\%$) was used as a metric to identify the presence of inertial turbulence. However, this is not relevant as the revised manuscript no longer attempts to distinguish between turbulent and non-turbulent events and forces the fit to use $n = -5/3$ to simplify the calculation of $\varepsilon$.

     – In Figure 12, the red lines should be limited to $f > 0.1$ because it was the threshold used for linear fitting.

85      The revised manuscript no longer includes these figures.

     – Line 595: "... nearly constant with $\alpha$" (?)

     The revised manuscript no longer includes the figure being referred to.

- In the conclusion section, Lines 646-649 comment on comparisons between Ri and EDR (the former figure 13), which are no longer described. Qualitative comparisons can only be made from Figures 14-15 and 16.

90    The revised manuscript has a completely rewritten Summary and Conclusions section.

**Response to Referee 3**

We appreciate the time taken by the referees to review our manuscript and provide comments and suggestions for its improvement. We have taken some time to essentially reset and rewrite the manuscript which, although the measurements and results are largely unchanged from our first submission, has resulted in what we believe to be a much cleaner submission with more direct focus on the measurement technique and much less attention spent on interpretation of the results.

Below, we respond to the individual comments made by the referee on the last version of the manuscript. As the referee's comments to the last revision was appended to our response of their review of our original submission, we have only included comments referring to the most recent submission. Furthermore, we have preserved the entire comment/response chain below. To try to minimize any confusion regarding the sequence of the discussion, we have included our initial response in green and our current responses are included in blue. We hope that this is not too confusing.

1. Similarly: "which allowed the connection to be made between the locations of increased turbulence intensity and the source of its generation" was tenuous, only to the level of "consistent with".

   This is a fair assessment, we have revised the text accordingly.

   The abstract still contains this same phrase.

   We have rewritten the abstract in the revised manuscript.

   Also in the abstract, the sentence "By being able to transect the air, the glider allows for turbulence wavelengths to be sampled at a particular altitude, improving statistical convergence and spatial resolution of derived statistics from its in-situ sensors." is misleading, since the glider cannot remain at a particular altitude, and demonstration of improvement of convergence and spatial resolution remains missing in the paper, as noted in the previous comments above.

   We have rewritten the abstract in the revised manuscript. A more nuanced response to this comment is also detailed below.

2. "However, due to the transient nature of their Lagrangian flight trajectory, balloon-based approaches are not necessarily amenable to obtaining detailed statistical descriptions of turbulence at high altitudes." Why? Aircraft are also transient, and if GPS guided, only see what is advected past. Balloons with altitude profiling are not Lagrangian vertically, so also sample more than one parcel of air. The statement is vague: it depends what statistics are being evaluated.

   True, and addresses the same point as the first detailed comment. We have revised the statement to be less vague.

   115: "However, as balloons advect horizontally with the wind and are unable to maintain a fixed geospatial location, balloon-based approaches would benefit from complementary measurement approaches" is even more vague.

   See below regarding Eulerian vs Lagrangian observations.

   120: The scientific advantage of remaining over a fixed ground location is not self-evident, as the features of the atmosphere typically do not. Or is the advantage primarily operational in nature? Please articulate this point. Also, the transecting ability of a glider is not related to the helical profile, but to its horizontal airspeed. The sentence "These qualities facilitate the statistical analysis necessary for quantification of non-stationary properties." remains unsupported in the paper (that I can find).

   We were not aware that this was not self-evident. Particularly since most (if not all?) numerical models of the atmosphere are conducted on Eulerian grids, it would seem intuitive that observations intended to improve our ability to model the atmosphere should therfore be conducted as close to a fixed geospatial reference location as possible (i.e. the UAS stays within a spatial scale corresponding to a numerical grid dimension). Furthermore, studies of turbulence in a Lagrangian frame of reference introduce different statistical behavior than Eulerian studies of turbulence (see, for example, Federico Toschi and Eberhard Bodenschatz (2009) "Lagrangian Properties of Particles in Turbulence" Annual Review of Fluid Mechanics 2009 41:1, 375-404). For example, it has been long recognized that Eulerian and Lagrangian integral scales are not interchangeable when modeling turbulent dispersion.

90: "A glider offers advantages over traditional balloon launches by being able to maximize time at altitude during its descent phase" Vertical rates for the glider vary from 5 m/s to 1 m/s, very similar to descending balloons. "These qualities facilitate the statistical analysis necessary for quantification of non-stationary properties" is not supported by evidence in the paper.

We respectfully, but strongly, disagree on this point. Note that in 1000 m of altitude change, for the current experiments the orbit of the glider means that it samples approximately 15,000 m along its flight path, whereas a balloon will sample only the 1000 m. This is a significant difference in the amount of atmosphere and range of eddy sizes that are sampled over the same vertical distance. Note also that the current configuration of the aircraft means that the turbulent eddies are acquired at an order of magnitude higher temporal resolution as well. It therefore would be very difficult to reproduce the spectra of Figure 11 (and corresponding $k$ and $\varepsilon$ estimates), and azimuthal distributions of these statistical properties (as shown in Fig. 14) with a balloon, particularly over the wavenumber range and at the vertical resolution that the glider can measure. We have added the above discussion to the conclusions to ensure that these points are not overlooked by other readers.

The objectionable phrase has apparently been removed from the introduction. The confusion arose because the vertical resolutions of both platforms are similar; the differences relate to the horizontal velocity.

Some confusion remains in conflating stationarity with homogeneity. Variations seen in a fast (e.g., 60 m/s) transect are likely to be due more to spatial inhomogeneity than temporal changes in statistics, provided that the overall time interval of the record is small. The higher glider airspeed speed allows larger spatial scales to be assessed in a shorter time, making the Taylor assumptions used in the turbulence estimation frequency-to-wavenumber conversion more likely to hold for the larger scale observations.

We agree. But we also note that most fundamental theories of turbulence are constructed in the wavenumber/spatial domain, hence there are intrinsic benefits to measuring turbulence in the spatial domain compared to the time domain (particularly as there is reduced reliance on Taylor's frozen flow hypothesis). True, this does require an assumption of horizontal statistical homogeneity in the vertical distance traversed by the aircraft while gliding, but this assumption is no less restrictive than the assumption of stationarity of statistics over the same vertical distance when examining behavior in the time domain. Furthermore, not only are larger spatial scales assessed in a shorter time, but also more of the smaller spatial scales are assessed. As noted, a balloon ascends/descends at roughly the same speeds as the glider, hence this is the rationale behind our statement that there will be increased statistical convergence (better averages of the scales smaller than the horizontal distance traversed.

Because the glider transects are mostly horizontal, the changes measured are likely to be due more to horizontal inhomogeneity than vertical inhomogeneity. Balloons provide vertical profiles of parcels and cannot sample laterally. Their ability to ascend/descend slowly (few m/s) provides high spatial resolution (vertically). A glider with 60 m/s airspeed and a 15:1 glideslope, although descending at a similar rate (4 m/s here) could be considered as having a similar vertical resolution, provided that the variations observed are dominated by vertical variations, and not the lateral ones. E.g., for the 1500m record lengths used, is it more likely that data reflects the 100m vertical change, or the 1500m lateral change? The latter is supported by later discussion in the paper (around line 449), where time successive vertical gradients are erratic, due to the large lateral changes, and where true vertical gradients are assessed by considering segments on successive turns on the helix spaced (vertically) about 2.5 km apart, leading to vertical resolutions on this order. This does not compare favorably to balloon vertical resolution.

A glider provides complementary measurements by sampling (mostly) laterally, and in this case with a very large helix diameter (10km) with an ability to evaluate large scale lateral inhomogeneities.

We agree on these points as well. The only thing that we should have clarified is that our comparison assumes horizontal homogeneity of averaged statistics. We are not fully convinced that this assumption is valid, however we have had some difficulty convincing other referees that some of the features we are seeing in the data can be attributed to horizontal heterogeneity at large scales. This is something that we hope to investigate further in follow-on investigations.

Here with record lengths of 1500m, lateral spatial resolution, e.g. of turbulence quantities, is of this order.

In our experience, resolution typically refers to the smallest scales that can be resolved, and not the largest scales that can be resolved. This difference in interpretation may explain some of the confusion with regards the statements in the previous manuscript version.

Higher temporal resolution due to higher airspeed is not an inherent advantage, since the frequency-to-wavenumber conversion makes, say, epsilon, invariant. The effect on instrumentation, is, in fact unfavorable, since higher sample rates are required, and the inherent bandwidth of the sensors and plumbing can be a limiting factor for turbulence parameterization.

Note that it has long been understood that Taylor's frozen flow hypothesis is a poor assumption for large scales (Moin P. Revisiting Taylor's hypothesis. Journal of Fluid Mechanics. 2009) hence the frequency-to-wavenumber invariance may not be strictly true and measurements where it can be avoided/minimized should advantageous. We do agree that higher relative velocities requires higher bandwidth sensors, hence our efforts to quantify the bandwidth of the sensor used here. We would like to point out, though, that many in-situ velocity sensing devices have a nonlinear response to velocity and hence it is actually advantageous for sensitivity and signal-to-noise ratio for relative velocities to be higher rather than lower. This is the case for five-hole-probes such as used in this study, for which $\Delta P = 0.5\rho\Delta V^2$. Hence, a $\pm 1$ m/s velocity fluctuation results in a 144 Pa pressure difference at 60 m/s relative velocity whereas a $\pm 1$ m/s fluctuation at 2 m/s relative velocity results in only a 5 Pa pressure difference being produced. Constant temperature anemometers also have a nonlinear response but tend to be more sensitive at lower velocities. However, they have other issues including increased sensivity to electrical noise, calibration drift, and require calibration typically using a pitot probe (at least in wind tunnel studies).

E.g., with a 20 Hz sensor bandwidth, scale sizes greater than 2m can be observed at 60 m/s (assuming no noise floor issues). Preserving 1 decade of inertial range would require a record length of at least 20m, but with the noise floor seen in Figure 12 c), about 60m records would be required, providing a maximum of 60m (mostly horizontal) resolution for turbulence parameters. A balloon moving at 4 m/s (with the same sensors) would have a (vertical) spatial resolution of 4m.

We are confused a bit here, but believe it caused by differences in how we define resolution. As noted above, the bandwidth disadvantage at higher velocities can be offset by higher signal-to-noise ratio and lower relative velocities can actually be disadvantageous depending on the type of sensor used.

Again, from our perspective, being able to sample more wavelengths over the same 4 m vertical ascent distance is an advantage, not a disadvantage, for reasons noted above. For a balloon to capture the same wavelength range over that 4 m of ascent would require the same relative velocity as the glider, i.e. 60 m/s relative winds assuming the validity of the frequency-to-wavenumber invariance assumption. For 10 m/s relative winds, it would take 6 times longer to sample the same wavelength range (i.e. a decade) of turbulence. For the same rise/descent rate, the balloon would therefore have travelled 24 m, compared to the aircraft's 4 m, thus resulting in higher resolution in the aircraft measurement. Of course, again, this assumes horizontal homogeneity in the turbulence statistics. This comparison of course assumes that a balloon is rising/descending and not neutrally buoyant, for which very long wavelengths can be captured at a single altitude, something a glider UAS could not do.

Thus, high-speed slant-path sampling has complex trade-offs that are not served by oversimplification. The paper could be improved by a more concrete discussion of the relative merits in fundamental terms, or refrain from vague comparisons.

Although we are enjoying the discussion here, our initial submission assumed that many of the points we made above would be intuitive for the readership of AMT and therefore not require lengthy exposition. Detailed evidence-based demonstrations of these points would require different experiments than the ones conducted here, which motivates future efforts. Therefore, for the sake of expediency (as this manuscript has languished overly long in review), we have elected to take the second option with the current revision and removed any vague comparisons to balloon-based measurements.

E.g., I don't see where the analysis records are defined for the Turbulence analysis. So it's hard to place the unquantified characterizations such as "high vertical resolution" (now in the Conclusion) in context. Likewise, "increased statistical

convergence" would depend on a larger number of points in the data record, together with an assumption of statistical stationarity. The data record sizes were only discussed as covering a fixed horizontal distance, so the record sizes would be smaller at the higher altitudes, and the time intervals shorter. It would be helpful to describe these details in order to support the advantages discussed.

230    These points are largely addressed above. Specifically, we consider 'high vertical resolution' as being the large wavelength range that is sampled for a comparatively small vertical distance. We also consider 'statistical convergence' from the perspective of the number of samples of a particular wavelength of turbulence that can be sampled over a particular vertical distance (e.g. a 60 m sample would be able to average more 1 m wavelengths than a 10 m sample). The UAS also provides advantages for 'statistical stationarity' as it can sample more wavelengths in a shorter period of time, therefore

235    reducing the time the turbulence has to evolve temporally. However, we acknowledge that if we were to define these terms with respect to the sheer number of samples acquired in time, rather than wavelengths, the data record size is conflated with the relative velocity and such comparisons become more complex.

3. Difficulty of conducting UAS measurements of this type in the NAS was not discussed, nor the conditions under which the reported flights were allowed. Was this in restricted airspace? Under who's auspices? Or was this in the NAS under

240    a COA?

The reviewer raises a good point as the current regulatory environment prevents these types of measurements from being routinely conducted. In the current experiment, flights were conducted in restricted airspace managed by the SpacePort America facility and coordinated with the nearby White Sands Missile range. We have added revisions to the manuscript to include these points.

245    I don't see where White Sands coordination is mentioned.

We did not include it as we felt it was superfluous to the fact the flight was conducted in restricted airspace used. However, we have added this note into the revised manuscript.

4. how was aircraft sideslip angle determined? How did the use of this affect the quality of the horizontal wind measurements?

250    While revisiting the wind measurement procedures, it was found that the probe was actually rotated 90 degrees relative to what the authors initially thought. This meant that it was actually the pitch holes that were disconnected for flights 2 and 3 and not that yaw holes. We also compared flight 1 data with and without the revisions required to calculate winds for flight 2 and found that the differences were negligible. The text has been revised the text accordingly.

The discussion of wind calculation for flights 2 and 3 in paragraph 254-262 is vague and disconcerting. The pressure lines

255    disconnected are not specified in the text. The use of vertical aircraft speed from the "variometer" (presumably pressure altitude rate, i.e., inertial aircraft vertical velocity) does not reliably indicate vertical airspeed, just as inertial horizontal velocity does not indicate horizontal airspeed. "Flight 1 data was processed with the original and revised approach, and the impact on the results on the resulting wind velocity statistics found to be negligible". Which statistics? How negligible? Angle of attack, and hence vertical wind velocity 6 is likely to be significantly affected, but the horizontal

260    wind may not depend on this very much. "although some differences in the frequency content of the vertical wind component could be expected." Why? What differences? Why is this important? Vertical wind velocity does not seem to be used later.

We have provided a much more detailed comparison of the substitution of the aircraft's angel of attack for the probe's angle of attack in the revised manuscript.

265    5. Generally, the details of this particular mutli-hole probe and its calibration and resulting accuracy were not provided. Can these be referenced from an earlier publication?

The probe used here is derived directly from the probes used in:

Barbieri, L. and Kral, S. T. and Bailey, S.C.C. and Frazier, A.E. and Jacob, J.D. and Reuder, J. and Brus, D. and Chilson, P.B. and Crick, C. and Detweiler, C. and others (2019) "Intercomparison of small unmanned aircraft system (sUAS)

270    measurements for atmospheric science during the LAPSE-RATE campaign," Sensors 19(9), 2179.

and utilize calibration systems and approaches described in: Witte, B.M., Singler, R.F. and Bailey, S.C.C. (2017) "Development of an Unmanned Aerial Vehicle for the Measurement of Turbulence in the Atmospheric Boundary Layer," Atmosphere, 8(10), 195.

Al-Ghussain, L. and Bailey, S. C. C. (2022) "Uncrewed Aircraft System Measurements of Atmospheric Surface-Layer Structure During Morning Transition," Boundary Layer Meteorology, v185, 229-258.

We have added these references to the revised manuscript. Note, that these probes have also been successfully deployed in previous studies, including:

Bailey, S.C.C., Smith, S. W., Sama, M.P., Al-Ghussain, L. and de Boer, G. (2023) "Shallow katabatic flow in a complex valley: An observational case study leveraging uncrewed aircraft systems," Boundary Layer Meteorology, v186, 399–422.

Bailey, S.C.C., Sama, M.P., Canter, C.A., Pampolini, L.F, Lippay, Z.S., Schuyler, T.J., Hamilton, J.D., MacPhee, S.B., Rowe, I.S., Sanders, C.D., Smith, V.G., Vezzi, C.N., Wight, H.M., Hoagg, J.B., Guzman, M.I. and Suzanne Weaver Smith (2020) "University of Kentucky measurements of wind, temperature, pressure and humidity in support of LAPSE-RATE using multisite fixed-wing and rotorcraft unmanned aerial systems," Earth System Science Data, 12(3), 1759-1773.

Bailey S.C.C., Canter C.A., Sama M.P., Houston A.L. and Smith S.W. (2019) "Unmanned aerial vehicles reveal the impact of a total solar eclipse on the atmospheric surface layer" Proceedings of the Royal Society A, 47520190212.

so they, and their use, are not untested.

However, all these references relate to low altitude use. Has the probe been calibrated for use at the low pressures and temperatures encountered in this study? How do you know that the 20 Hz bandwidth (line 495) holds at high altitudes, since this depends on "viscous attenuation of the pressure fluctuations within the tubing" (line 252), and kinematic viscosity increases markedly at high altitudes.

As I am sure the referee is aware, conducting a wind tunnel directional calibration at the conditions experienced at high altitude would require specialized facilities that we did not have access to. We did verify operation of the equipment at low temperatures and pressures in an environment chamber prior to installation in the HiDRON (although we could not conduct velocity/directional calibrations in this chamber). However, we also repeated the calibration at the lowest velocity our tunnel can produce, as well as at conditions matching flight dynamic pressure, to check the Reynolds number independence of the calibration (as well as with and without the probe heat). However, prior to installation in the HiDRON we had not conducted the frequency response test in the environment chamber. This test was conducted since the last revision and the results are discussed in the revised manuscript.

6. A portion of the descent from the 30km release seems to be very steep. There were also some very tight circles at isolated points in the first two descents. Why?

These flights were also test flights for the aircraft. During the flight, the operators conducted several tests of their systems which included adjusting the flight profile mid-flight and improving the response of the aircraft following release. Note that the steep release at 30 km is due to the requirement to achieve sufficient dynamic pressure to produce enough lift for controlled flight. The lower density means that the aircraft must fall a certain distance before the aircraft can travel to it's measurement location.

I don't see any revisions to this effect. It would be interesting to know the "measurement ceiling" of this system.

We have added a statement to this effect in our current revision. To our knowledge the aircraft's ceiling is not yet known, as the flights reported here were the highest altitudes the aircraft has been tested at to date.

7. central differencing between adjacent 30 sec averaged values?

That is correct, although we have updated the revised manuscript to use spatially regular, rather than temporally regular, segments.

As noted above, the effect on the record samples and time length would be helpful.

We have included more rationale behind our selection of record length in the revised manuscript.

8. compare well given the spatial offset and the local weather conditions, and if the local periodic variations are ignored. These variations are suspiciously periodic with altitude, raising questions about artifacts from the platform airspeed/attitude/descent rate that may be varying with the same period. (See the related comments about Ri later). Some evidence should be provided that these results are not correlated with aircraft motions.

This was addressed when responding to the initial comment. The additional radiosonde profiles in the revised manuscript also provide increased context for the wind fluctuations observed.

The radiosonde wind magnitudes seem highly smoothed/decimated with straight lines connecting large changes in altitude. So much so that they don't really corroborate the "periodic" variations in the Hydron winds of concern. I would think that standard 1Hz cadence radiosondes would provide higher resolution.

We agree. We tried to find the NWS data at standard 1Hz but could only find them with the resolution provided here, hence our use of the phrase 'publicly available' when referring to the NWS radiosonde data (as these data are likely to exist in a repository that we are unaware of). We also looked at ERA reanalysis data, but it had similar resolution to the radiosonde data we found. We note that the manuscript does contains a comparison to wind profiles measured during ascent by the aircraft's GPS that provide favorable comparison.

9. The Ri profiles seem to have highly periodic excursions with altitude. Could these be at the same period as the orbits the plane executes on the descent? That is, how do we know this is not an artifact of the sensors or the periodic motion of the platform? It would also be good to see how this correlates with the bank angle of the plane, since this will not be constant in wind. It will be difficult to take the results at face value without careful checking for motion/attitude/descent rate artifacts from the platform. Similarly, the Ri values seem suspiciously low, with < 0.25 values for much of the flight. Are these low values periodic anomalies in the measurements?

We have extensively updated the text addressing this variability and present an improved method to calculate Richardson number. See additional discussion above regarding the dependence of wind velocity on the orbits.

"The vertical gradient of each quantity was then found by finding the nearest statistical segments in the positive and negative z directions which minimized the difference in $\alpha$". This is difficult to unpack (took me a while). I think you are saying that helix segments one turn apart, above or below each other, are selected for each alpha. This could be clarified better.

This was want we were saying, we have simplified the description in the revised manuscript.

Figure10: Rit is erratic and very small, Riz is less erratic but very large! As noted, this mismatch is due to the mismatch in Si vs Sz, since Nz essentially smooths Ni. But the reasoning behind the difference is S also applies to N, so the issue with S is "most likely" due to something else. Thus it is hard to believe either Ri value, or either S value.

As can be observed by comparing existing profiles of wind and temperature, it is common for there to be more rapid changes in wind speed and direction than in temperature. This is also the case in the current study. We should therefore expect higher sensitivity of $S^2$ to the vertical scale used for calculating gradients than $N^2$. Hence the reasoning for these differences should not be expected to apply equivalently to $N^2$ and $S^2$.

We have struggled with the sensitivity of $Ri$ to the details of its calculation, but the method applied in the previous and current revision is similar to profiles presented in radiosonde literature. We have revised the discussion of $N^2$ and $S^2$ in the current revision to try to better reflect the sensitivity of the results to the method used for calculation.

10. $F_{uu}$ and $\Phi_{uu}$ are both used for the power spectral density (please be consistent), and these are incorrectly called the "frequency spectrum".

Please note that we are using colloquial language used in contemporary fundamental turbulence literature whereby the power spectral density is referred to as the velocity spectrum, or more broadly the energy spectrum, or shortened to simply the spectrum depending on context. Furthermore, these "spectra" are often distinguished as to whether their dependence is in the frequency domain or wavenumber domain by referring to them as the "frequency spectrum" or "wavenumber spectrum" respectively (see, for example, Pope (2000) "Turbulent Flows", Cambridge University Press.)

We made the distinction in the original manuscript between $F_{uu}(f)$ (i.e. the frequency spectrum) and $\Phi_{uu}(\kappa)$ (i.e. the wavenumber spectrum) as these are separate functions since both $\int_0^\infty F_{uu}df$ and $\int_0^\infty \Phi_{uu}d\kappa$ must both return the variance of the velocity component that they are calculated for (e.g. $\overline{u^2}$ in the above case). As $\kappa \approx 2\pi f/V_{rel}$ this means that $\Phi \approx FV_{rel}/(2\pi)$. We realize that this distinction is confounded by the fact that the original figure 5 was mislabeled as $\Phi_{uu}$ instead of $F_{uu}$ for two of the subfigures, but this was a typographical error and not intentionally inconsistent. In the revised manuscript we strive to be more clear that we are referring to the velocity spectrum, and try to be more clear with the distinction between $F(f)$ and $\Phi(\kappa)$. Reviewer 2 also did not like the use of $\Phi(\kappa)$, so we have now replaced that with $E(\kappa)$. We also now present the wavenumber spectra instead of the frequency spectra in the revised manuscript as it was more relevant for the discussion of determining the inertial subrange slope.

Because of the confusion in the literature, it would be helpful to distinguish power spectral density from amplitude spectra, power spectra, etc., since this makes a difference in the quantitative results. Note the Welch method can also be used for smoothing the amplitude spectrum, and the discrete Fourier transform can use different normalizations. To make it clear what was done, it would be good to include the expressions for the Fourier transforms used, or provide a specific reference.

The calculation was conducted using the Matlab function, pwelch, and have added that information to the revised manuscript. However, the documentation does not provide details of the analytical form of the calculation. That said, during this and previous studies we have extensively compared the output of the pwelch function to the frequency spectra calculated via FFT, and have confirmed that integration of the resulting one-sided frequency spectrum returns the variance of the signal (e.g. Saddoughi and Veeravalli, JFM, 1994, Pope, Turbulent Flows, 2000). The only difference we've noted is the additional smoothing due to the averaging windows, which we minimized by selection of window size in the present calculation (as also mentioned by the referee), which we found slightly reduced scatter in the corresponding results and was was why we elected to use the Matlab function rather than the results from the FFT calculation.

11. shape of the "spectra" in Figure 8 is very strange. Part a) seems to have a noise floor near $10^{(-3)}$, but the noise floor is smaller (near $10^{(-6)}$ at a higher altitude (part b)! (Where pressure fluctuations are necessarily smaller). A noise floor is again seen near $10^{(-3)}$ in part c). Also, the spectral slope is too shallow in part c) as noted later in the paper. Could it be that this power spectral density is the noise figure of the sensor itself, and there is really no detectable signal at these low atmospheric pressures?

As noted above we had mislabeled Fig. 8 due to changing from presenting the results in the wavenumber domain to the frequency domain. The original figures were actually presenting the results in the wavenunumber domain and we are unhappy that this error made its way into the final submitted manuscript. We also believe the reviewer is correct that the -1 slope is the noise figure of the sensor (or more accurately the combination of sensors used to determine the wind) and that the -1 slope indicates that there is no measurable turbulence present.

The noise floor noted above was a flat level, not a $f^{-1}$ slope. This is typical of electronic Johnson noise in sensors. This can be seen at high frequencies in the new Figure 12 b) (at a level of about $10^{-3}$) and c) (at a level of about $10^{-2}$), which now makes more sense with the increase in altitude. Were the spectra in Fig. 12 cases 14 deemed Kolmogorov turbulence? If so, it would be good to show spectra for those that did not so the noise floor issue can be clarified against your fitting/qualification procedure.

It was not clear to us where you were seeing 'flat' energy roll off, particularly given the amount of scatter observed in spectra. Our -1 slope comment was more related to the fit to the range $0.1 < f < 20$ Hz) which was returning a -1 slope instead of a -5/3 slope when the energy content at high frequencies dropped below the noise floor (therefore reducing the average roll-off of the spectrum). However, this discussion is no longer relevant as the current revision has made several updates to the method used to find $\varepsilon$. Specifically, we now allow the fit range to vary by identifying the noise floor, and we fix the slope of the fit at -5/3 such that $\varepsilon$ is now the only fitted parameter. We have also removed the individual spectra from the revised manuscript, replacing them with a wavelet transform of the entire flight time series.

12. Buoyancy Reynolds number can be calculated after estimating epsilon, as a check on this assumption.

This assumption is not expected to be valid over the entire measurement range, and is checked/unvalidated by the power law fit discussed on lines 456-457. However, assuming it's validity is necessary to produce a dissipation rate estimate using this approach. Validity of this assumption is therefore reflected by the filled circles on Figures 9 and 10.

Anomalous spectral shapes can return a $f^{-5/3}$ fit. It's the standard deviation of fit that is telling, and this not reported. Lines 456-457 in the revised manuscript concern N and S, not epsilon. The case of flight 1 in figure 12 is strange in that it has EDR values higher than many points in flights 2 and 3, yet these values are not qualified as turbulent. Also seems strange that the EDR is higher but TKE is smaller for flight 1. This does not make sense. How do the spectral shapes compare? Seems the sensor direction in flight 1 has been fixed from the original paper, but are there any other sensor system differences between flights? Did w' make a significant difference in the TKE calculations for flights 2 and 3, given the comments earlier that w' is likely to be incorrect?

We agree with the reference to anomalous spectral shapes and value of $R^2$ for fit accuracy. The lines referenced were the prior submission and hence why they weren't relevant to the revision. As noted above, we have adjusted our $\varepsilon$ calculation to integrate only to the noise floor, which has addressed many of the concerns noted above. With regards to $w'$, as shown in Appendix A in the revised manuscript, the impact on $w'$ of replacing the disconnected transducer was not as large as could be expected and did not make a significant impact on the TKE calculation.

13. Not clear how (of if) the noise floor/noise figure is removed in the qualified data fits.

We made no attempt to remove the noise floor from the data fits.

The noise floor can corrupt the estimated slope, so its removal can often retrieve epsilon values that were previously rejected. This could help to explain the absence of qualified values at the higher altitudes (ala line 552),

In the latest revision we have removed the noise floor from the data fits. This process is explained in the revised manuscript. We found that it did 'clean up' the profiles of $\varepsilon$ a little bit.

14. confirmation that the infrasound signal is due to turbulence is too strong a conclusion at this stage. Localized increases do not correspond to those in EDR.

This statement was driven by the increase observed as the aircraft enters the boundary layer and becomes immersed in the turbulence (as evident in the comparison of Figure 10 and Fig.12) and also the comparisons in Figures 14-16. However, we agree that this conclusion is weak, and that the infrasonic measurements need more development.

570: "with the range used selected due to finding that lower frequency acoustic content better correlated with the EDR values measured with the five-hole-probe when compared to the higher frequency acoustic content, which tended to contain additional signal noise."

This is problematic: seems like you are cherry picking aspects of the infrasound signal that correlate with EDR, in order to confirm that infrasound signals 16 indicate turbulence (EDR). What is the nature of the "additional signal noise"? 577: despite agreeing that "this conclusion is weak" above, the line "providing an initial confirmation of the presence of infrasonic sound generation by turbulence" is still present in the paper.

We have revised the infrasonic microphone analysis and find that normalizing the low-pass filtered microphone response by the unfiltered response we are able to get good agreement between the long wavelength wind velocity variance and the infrasonic (sub 20 Hz) acoustic energy. This threshold was selected to conform with the typical definition of infrasonic frequencies.

15. 592: "We therefore also only examine the behavior of other quantities in terms of relative trends in these values, rather than specific thresholds." Vague.

This section is no longer included in the revised manuscript.

16. 595: "Examining Flight 1, the distribution of measured Nz shown in Fig. 14a indicates that the static stability conditions are nearly constant with altitude." Stability is significantly increasing with altitude! This was also stated earlier in line 469.

This was a typo and should have read constant with aziumuth. Regardless, this section is no longer included in the revised manuscript.

17. 655: "were actually produced by the aircraft passing through the same structure more than once." This is not clear from the discussion in that section. Pointing out a specific example or two would help to support this comment.

This section is no longer included in the revised manuscript.

18. 662: Despite this ambiguity, these initial flights suggest that the sUAS measurements suggest the potential exists" Please reword.

The Summary and Conclusions section has been rewritten.

19. 666: "the flight pattern allows for increased statistical convergence due to the larger volume of air sampled over a particular altitude range." How does a larger sample volume automatically provide "increased statistical convergence"? This is likely to produce more variation in measured quantities. Also, the conclusion is not a natural place to introduce a specific argument.

The Summary and Conclusions section has been rewritten.

---

## Author Response (AR4)

**Response to Referee 3**

We appreciate the time taken by the referees to once again review our manuscript and provide comments and suggestions for its improvement.

Below, we respond to the individual comments made by the referee on this version of the manuscript. To do so, we have reproduced the original review, with our comments provided in blue text.

This paper has been improved by a tighter focus and omission of many previous vague and unsupported statements. Most of the arguments from the previous reviews have been settled by removal of the sections in question, or by clarification in this revision. Remaining issues are noted below.

Comments:

– Line 10: "nominal vertical resolution on the order of 1 m during was achieved" Please re-word.

We have reworded this statement to 'a nominal vertical resolution'.

– Line 15: "and broadband response measured within boundary layer turbulence". Do you mean "broadband winds"?

We have reworded this statement to "The low-frequency response of the infrasonic microphone was found to correlate to long wavelength wind velocity fluctuations measured at high altitude, and broadband frequency response of the microphone was also measured within boundary layer turbulence.'

– Line 27: "predict stratospheric and high altitude turbulence" High altitude is different than stratospheric?

We have reworded this statement to 'predict stratospheric and upper-tropospheric turbulence'.

– Line 114: "which allowed it to measure a wide range of turbulence scales horizontally at high vertical resolution." Confusing/misleading. Implies that horizontal and vertical effects can be separated, or that vertical variations dominate horizontal ones. Statement should be qualified accordingly.

We have reworded this statement to 'which allowed it to measure over a distance of up to 50 m horizontally for a 1 m change of altitude'.

– Line 211: "to check for any Reynolds-number dependence of $C_\beta$, $C_\alpha$ and $C_q$." Were there any?

We have added a statement that none was found.

– Line 214: "flexible polymer tubing" could be almost anything. Usually, these tubes must be semi-rigid to properly propagate the pressure variations. Please specify.

The manufacturer sells this product as PVC plastic tubing with a trade name of Tygon. We have updated the text to include this information.

– Line 259: "angle between the true airspeed determined from the aircraft's pitot probe, and the vertical velocity determined by the aircraft's variometer". No idea what this means. Pitot does not provide an angle, and variometer (pressure altitude rate, akin to inertial velocity) is not the same as the vertical component of relative wind.

When the autopilot manufacturer was queried as to how the angle of attack is determined, the description above is verbatim to the response we received. We have contacted them again and have updated the description of how $\alpha$ was found, specifically from a combination of the true airspeed determined from the aircraft's pitot probe, the vertical speed determined by a Kalman filter fusion of the static pressure rate of change, vertical acceleration, and global positioning system velocity, along with the aircraft orientation measured by the autopilot gyroscopes which was used to transform between inertial and body-fixed coordinates.

We have updated the text accordingly.

– Lines 293-301: Sensor and signal amplification/conditioning and quantization noises are random processes, so cannot be measured before-hand and subtracted from signals measured later. The noise removal process described is as likely to corrupt the measurement as it is to clean it up. Even if "noise" means biases, these are likely to be variable with temperature, so could not conceivably be obtained from pre-flight sampling.

This would generally be true, except that much of the high frequency noise in the pressure transducer signal is quasi-periodic noise which we believe to be introduced by a switching voltage regulator, rather than more stochastic noise sources described above. In this revision, we have added additional exposition about the noise characteristics as well as evidence that the approach we have devised is successful in reducing the impact of the quasi-periodic noise, and evidence that the noise was independent of atmospheric conditions. Specifically:

1. Excerpts from time series illustrating the noise content.

2. Frequency spectra showing that the frequency content of the transducers in an inactive environment chamber did not change between atmospheric and low pressure/low temperature conditions.

3. Frequency spectra for all altitudes, which shows that the high frequency content of the transducer signal does not change during the flight for frequencies higher than the signal from the five hole probe

4. A comparison of excerpts of the pressure time series taken during flight showing that the filter appears to successfully remove electrical noise from the pressure signal.

5. A comparison of frequency spectra before and after filtering showing that the high frequency content is reduced, and that the pressure signal more accurately reproduces a -5/3 slope in the square-root of the corresponding dynamic pressure signal (i.e. in the resulting velocity signal).

Please also note that this noise filtering approach was added in response to a previous comment that stated 'Not clear how (of if) the noise floor/noise figure is removed in the qualified data fits. The noise floor can corrupt the estimated slope, so its removal can often retrieve epsilon values that were previously rejected. This could help to explain the absence of qualified values at the higher altitudes,' As the noise floor in the wind velocity was a function of numerous inputs, we could not devise a hands-free, unbiased, approach to consistently identify or remove it from the wind estimate. Therefore, we introduced this filtering approach to at least remove some of the noise introduced by the pressure transducers. We found that its impact on the statistics presented in the remainder of the paper was negligible.

– Lines 340-341: "uncertainty in wind magnitude was found to be most dependent on the yaw angle" How? Why? Do the wind excursions in Figure 4 correlate with yaw angle?

The uncertainty dependence of wind magnitude on aircraft attitude is introduced into the wind estimate during the coordinate transformation between body-fixed and inertial coordinate systems. During flight, the yaw angle can vary from 0 to 360 degrees, whereas the sideslip, angle of attack, pitch and roll angles are near zero. The result is that the greatest contribution to the $u/v$ wind components during transformation is the true airspeed being multiplied by the sine/cosine of yaw and cosine of the pitch. The result is that during an orbit, the horizontal velocity components will have high sensitivity to yaw error at both $0°$ degrees and $90°$. For a similar reason, the vertical component of velocity is most sensitive to error in pitch.

As illustrated by the uncertainty bounds shown in Figure 4, the wind excursions cannot be explained by error in yaw. Note also that prior to preparation of this manuscript, we conducted an informal perturbation study to determine if these excursions could be explained by measurement error and found that for them to be removed from the wind profile, the yaw error would not only have to be non-monotonically dependent on altitude (i.e. not attributable to sensor drift), but it would have to be so high that it could only be explained by failure of the system.

We have added the first statement to the manuscript, the second statement is left out due to the informal nature of the perturbation analysis that was conducted (the uncertainty analysis presented in the manuscript being the more rigorous approach).

– Lines 400-401: "and we assume that the characteristics of the atmosphere within these segments are horizontally homogeneous (i.e. they are a function of z only)". A very loaded assumption to make with no justification!

This statement was not thought to be as loaded as as the reviewer is implying, due existing consensus that the intrinsic stability in the stratosphere and upper troposphere will promote horizontal homogeneity (as exemplified by one of the other reviewers being insistent that variability in the measured profiles must be due to measurement error or vertical stratification rather than any horizontal heterogeneity). As the aircraft approaches the surface and the radius of the orbits becomes smaller, this assumption also becomes less restrictive.

We have added our rationale for this assumption in the revised manuscript.

– Lines 440-441: "the HiDRON measurements do contain short wavelength fluctuations" How short? Please quantify.

We have added a statement that the wavelengths we are referring to are the ones on the order of 1 km or less.

– Line 444: "These low frequency waves may be bias in the wind estimate introduced by the orbital path". Indeed, it seems like they may correspond to one turn on the helix, so may be an anomaly in the wind retrieval. Comparison to the vertical period of the helix would be important here.

We have added some quantification and discussion of these values to the text.

– Line 464: "backing with altitude"?

Backing winds are winds that change direction counterclockwise with height. We have revised the statement to remove the use of backing and simply state the wind direction changes.

– Line 470: Should note that an accurate assessment of TKE also depends on capturing the lowest wavenumber components in the inertial subrange, since they contain the largest energy per wavenumber. Did the PSDs over the statistical intervals exhibit a roll-off or flattening of the $f^{-5/3}$ slope at low wavenumbers, indicating the outer scales were captured?

A note to this effect has been added to the revised manuscript. The low frequency content more often increased, rather than rolled off, as described in the wavelet analysis, and discussion of the time series. This was also the justification for the selected segment length.

– Line 473: "over a specified frequency, $f$, range". What was the range?

The frequency range determination process was described in the manuscript shortly after this statement. We have revised the text by removal of the statement above so that the question of frequency range does not come up until it is actually described.

– Line 478: "which will reach a minimum at the frequency where the noise has a greater contribution to the integration than the signal". Why?

Compensated, or "pre-multiplied," spectra are commonly employed in turbulent boundary layer studies because they facilitate the visualization of energy spectra on semi-logarithmic axes. Specifically because they allow for a clearer representation of the frequency/wavenumber dependence of the relative contribution of each frequency/wavenumber to the overall energy content. This is because, for example,

$$d\langle u^2\rangle = F_{uu}df = fF_{uu}d(\log f). \tag{1}$$

Hence when $fFd(\log f)$ begins to increase on a semi-logarithmic plot at high frequencies, this indicates a frequency range where the energy content increases with $f$. Given that universal equilibrium range turbulence will decrease in energy content with $f$, the minimum in $fFd(\log f)$ indicates a frequency at which the noise begins to have a greater contribution to the variance than the turbulence content.

However, the use of the compensated spectrum is a procedural detail and in hindsight distracts from the overall point of this processing step, which is to determine at frequency the contribution to overall variance increases with increasing $f$,

rather than decreases. The same result could have been achieved using the un-compensated frequency spectrum, and we only used the compensated spectrum to simplify visualizing the frequency-dependence of $\langle u^2 \rangle$, $\langle v^2 \rangle$ and $\langle w^2 \rangle$.

As including the detailed exposition above would distract from this point, while adding little value to the overall intent of its inclusion, we have simplified the discussion to simply state that the upper bound of this range was determined by identifying the frequency where the noise has a greater contribution to the integration than the signal than the velocity fluctuations.

– Line 478-479: "filter frequency was consistent with the probe's frequency response in the boundary layer and varied between 1 Hz and 20 Hz above the boundary layer". Does "filter frequency" correspond with the "frequency range" in line 473? By "probe's frequency response" hear do you mean "probe bandwidth"? How did this vary above the boundary layer?

By filter frequency, we meant the upper bound of the range described in the previous statement. We have modified 'filter frequency' to read upper bound of the frequency range.

Yes we do mean probe's bandwidth. These terms are interchangeable and, as we use frequency response at numerous points throughout the paper prior to using it in this sentence, would prefer not to change it to 'probe bandwidth'. We have clarified that we are referring to the maximum frequency response, as frequency response in general can also refer to the Bode plot of the probe's response to excitation.

As noted in the text, the frequency at which the noise exceeded the signal varied between 1 Hz and 20 Hz. We do not describe a trend above the boundary layer since no trends were evident, being dependent on the presence of low-frequency energy content. We have altered the text to now read 'with the higher upper frequency bounds corresponding to instances where there was increased low frequency content in $F_{uu}$, $F_{vv}$ and $F_{ww}$.'

– Line 514-516: "due to the statistical segment length used for averaging, the value of $\langle k \rangle$ will be biased to wavelengths smaller than the statistical segment length" Why? Do you mean smaller than the outer scale (as noted for Line 470 above)? Is this what "and therefore may not completely describe the actual energy content of the turbulence" is alluding to? This could be said much more clearly.

That is exactly what we meant. Due to the segment size of $\sim 3$ km we cannot resolve wavenumber content larger than the length of the statistical segment (or frequency content below $O(0.01)$ Hz). The point of the above statement is to note that this is insufficient to capture any sort of outer scale/low frequency energy contribution below of wavelengths longer than the segment length. We have changed this statement to note that, in addition to the implicit assumptions made when calculating $\langle \varepsilon \rangle$, the method used to calculate $\langle k \rangle$ reflects only the energy content corresponding wavelengths smaller than the statistical segment length (or frequencies higher than the the inverse of the time taken to traverse that segment length).

– Figure 10: It would be helpful to draw a $\langle k \rangle^{3/2}$ line on the plot for reference.

This line has been added.

– Line 524: "Above the boundary layer turbulence $\langle k \rangle$ and $\langle EDR \rangle$ are largely in agreement" This is hard so see, since $\langle k \rangle$ should be proportional to $\langle EDR \rangle^2$ but Figure 11 compares $\langle k \rangle$ to $\langle EDR \rangle$.

We have changed this figure and corresponding references in the text to $\langle EDR \rangle^2$

– Line 535: "Nyquist frequency of the minimum probe response". No idea what this means. Nyquist relates to the sampling frequency, not the frequency response.

We have changed this sentence to refer half the maximum frequency response of the probe.

– Lines 538-540: "Noticeable in Figs. 11b, d, and f is the significant long wavelength content for $\kappa \ell < 0.003$ (wavelengths larger than 2 km) when $z > 10$ km." I don't see this. It would help to show a color bar. Looks to me like there is significant long wavelength content below .0003 rad/m over all altitudes. And why does the plot have a curved boundary on the left

and a straight boundary on the right? Seems that the "time" variable discussed in the wavelet transform is really altitude z here. Correct? This whole discussion is very terse for readers unfamiliar with wavelets.

The long wavelength content for $\kappa\ell < 0.003$ when $z > 10$ km is more evident when plotting the wavelet coefficient on non-logarithmic contours (which does a poor job of visualizing the short wavelength distribution of the coefficient), or when using a non-colorblind friendly colormap.

We have modified this figure by changing the colormap, added a colorbar, and adjusted the horizontal axis to better constrain the wavenumbers to those less than an orbit. However, the difference below 10 km and above 10 km is still subtle, therefore we have clarified within the text that the contributions for altitudes greater than 10 km are most noticeable for Flights 2 and 3

The wavelet transform is calculated in the time/frequency domain, but since altitude is a function of time (which we had tried to indicate by referring to $z(t)$), it is then plotted as a function of altitude. We have rephrased this sentence to be more clear.

The curved boundary on the left of the figure is due to the time-frequency nature of the wavelet transform. At the start and end of the time series, there is insufficient information to resolve the low frequency content. However, towards the central part of the time series, the maximum low frequency content can be resolved, resulting in the curved boundary on the left (low frequency) end of the figure. The right of the figure, representing the high frequency content, is not subject to such resolution issues and therefore has a straighter boundary (although in this presentation, since the frequency has been transformed to wavenumber using Taylor's hypothesis, the highest wavenumber resolved is a function of the airspeed, which decreases with altitude, resulting in the slanted boundary on the right of the figure.)

We have updated the text to provide more description of the wavelet transform and its features.

– Lines 541-547: What are the implications of these observations from the wavelet transform? Why is the frequency content behind turbulent parameterization important? Usually this is constrained by the inertial cascade.

Our primary rationalization for examining the low frequency content is that it helps us to explain the differences between $k$ and $EDR$, highlighting that $EDR$ does not capture the low wavenumber content. This analysis also helps us to justify and understand the use of the turbulent kinetic energy estimate using lower frequency energy content than used for the initial $\langle k \rangle$ estimate.

We have modified this paragraph to better highlight some of the above points.

– Lines 557-558: "and therefore is attributed to increased atmospheric absorption due to the increase in molecular mean free path with altitude". Seems this could also be due to the decrease in coupling coefficient to the microphone diaphragm due to lower density.

It is not clear to us how the density can affect the efficiency of conversion of mechanical energy to electrical energy. Assuming the intention was to describe the decrease in mechanical forcing which could be expected due to lower density, we then would expect the variance of pressure measured by the microphone to scale with $P^2$, rather than $P$. However, we found that it does not scale with $P^2$. In addition to various other normalizations, we also tried scaling the variance of the microphone signal with $\rho c$ (corresponding to the expected change in sound intensity with altitude), which was also was unsuccessful. The only scaling we found that provided some success was the normalization of the the variance of microphone signal with $P$, as noted in the paper, and this result was consistent with the expected attenuation due to increase in mean free path, following the discussion presented in the cited reference (Bass 2007).

– Line 562: "The altitude attenuation will be dependent on the local temperature as well". Why?

In the paragraph before the referenced statement we had attributed the altitude dependence to the mean free path, which is a function of pressure and temperature. We have added this statement explicitly to the above sentence.

– Lines 565-566: "the resulting infrasonic amplitude profile can be observed to strongly correlate with [k]". Depends what you mean by "correlate". The variations in [k] are not correlated with the normalized infrasound variance, only the large

scale means seem to correlate. The infrasound signal looks like a LP filtered version of the TKE. Seems the infrasound signal (even filtered at 20Hz) should be able to follow the variations in $[k]$ that occur over km of vertical intervals. Why does it not?

The use of the word correlate to indicate correspondence was a poor choice and has been changed. The $[k]$ measurement is an in-situ measurement, whereas the infrasonic microphone is a remote measurement. Therefore there is no reason to expect exact correlation between the two measurement approaches. Indeed, at least ideally, the infrasonic microphone will detect the turbulence before the aircraft enters it, acting as a filter to the 'spikiness' of the $[k]$ measured by the in-situ sensor. Not only that, but the microphone will also detect any sound generated by nearby turbulent patches that the aircraft does not fly through. The net result can thus be expected to be a low-pass version of the $[k]$ profile, as the sound propagation will be be omnidirectional and emitted from numerous locations, whereas the $[k]$ can be expected to be constrained to stratified vertical layers.

We have added these discussion points to the revised manuscript.

– Line 569: Do you mean $[\sigma]_{LF}$ instead of $[\sigma_f]$? Also, I don't understand why these two would increase at the same rate if high frequency energy is primarily increasing, as supposed as being "most likely".

Yes, we meant $[\sigma]_{LF}$ and have fixed this typo.

Rate is probably the wrong word for what we are trying to describe. We have reworded this statement to try to more clearly describe our rationale for why the ratio $[\sigma^2]_{LF}/[\sigma^2]$ will remain constant in the boundary layer.

– Line 575: "This is due to the helical flight path". It is really due to the small flight path angle. A steep helix would not be as susceptible to horizontal gradients.

We were referring to the specific helical flight path flown during these flights, not helical flight paths in general. We have reworded the statement to 'This is due to the shallow glide slope of the particular helical flight path flown in these experiments'.

– Line 585: "These values were then re-interpolated to each statistical segment." Don't know what this means, exactly. Averaged for each segment? Interpolated how? And why were the 200m smoothed data averaged again (binned) at 100m intervals?

In this context, the smoothing is applied as a low-pass filter, and therefore the time-series post-smoothing has the same number of data points as pre-smoothed. The bin averaging over 100 m intervals was done to facilitate the differencing across the 100 m interval used for gradient calculation (i.e. equivalent to downsampling the signal). In the past we calculated a difference across $\Delta z = 100$ m for each point in the time series, but a different reviewer took exception to this approach in a previous revision of this manuscript, stating it obscures the vertical resolution of the differencing. Finally, since the $\Delta z = 100$ m downsampled data points at which gradients were calculated do not coincide with the $z$ locations of the statistical segments over which $\langle \theta_v \rangle$ was calculated, we had to interpolate the gradients to locations of the statistical segments. It is a messy process, but seems to be necessary to achieve $Ri$ profiles similar to what is observed from radiosonde measurements.

– Line 601-602: "effectively reproduces the N2 profiles calculated along the flight path". Only if by "effectively" you mean a highly smoothed version of N2.

Yes, we meant that (particularly when compared to the $S^2$ profiles) this approach produced a highly smoothed version of the $N^2$ profile determined with other approach. We have reworded this statement to 'effectively reproduces the trend of the $N^2$ vertical profiles calculated along the flight path.'

– Line 617: "Wind profiles were in good agreement with the available National Weather Service radiosonde profiles". Please quantify "good". Likewise, quantify "best comparison" in the next sentence.

We have added a discussion quantifying the difference in the body of the paper while discussing the wind profiles and summarized this in the last section.

– Lines 619-620: "over a large horizontal wavelength range with high vertical resolution". Confusing/misleading. Implies that horizontal and vertical effects can be separated, or that vertical variations dominate horizontal ones. Statement should be qualified accordingly (as was done in the body).

255 We have reworded this statement to 'over a large horizontal distance relative to the vertical distance traveled.'

– Line 637: "fror example". Typo.

This typo has been fixed.

– Line 643: "Additional flight patterns can also be designed with tighter helical descent can be designed". Awkward.

Reworded to 'Additional flight patterns can also be designed with tighter helical descent'

260 – Appendix A: "all transducers" is vague. Wind retrieval is a combination of many transducers, including relative wind, attitude, and inertial velocity. It would be clearer to use the more specific nomenclature from the body of the paper.

We have removed the line legend and reference the line colors in the figure caption. To use the nomenclature from the body of the paper would have required shrinking the font size below acceptable limits (at least for the graphing software that we used when generating the bulk of the figures used in the paper).

**Response to Referee 4**

We appreciate the time taken by the referee to review our manuscript and provide comments and suggestions for its improvement.

Below, we respond to the individual comments made by the referee on this version of the manuscript. To do so, we have reproduced the original review, with our comments provided in blue text.

This article presents an exciting new gliding UAS platform for conducting potentially low-cost meteorological observations up to 30 [km] MSL. A calibrated five-hole pressure probe was employed to quantify turbulence characteristics and assess the effectiveness of an infrasonic microphone to qualitatively observe atmospheric turbulence. While the discussions presented in the article adequately support the effectiveness of the platform's capabilities to observe stratospheric environments reliably, the limited discussion detailing the processing of data needs further refinement before publication.

- 98: "Within the atmospheric boundary layer, the infrasound energy from ground-based arrays has been found to correspond to the turbulent kinetic energy in the atmosphere, particularly when buoyantly-produced 100 convective turbulence is present (Cuxart et al., 2015). The infrasound energy levels were also found to increase in the presence of elevated jets or turbulence above the measurement height, which was thought to be caused by the sound generated at higher altitudes reaching the microphones."

  "found to correspond to turbulence" how? A working hypothesis relating infrasound measurements to turbulence is warranted if any quantitative assessments of turbulence characteristics are to be derived from the infrasonic microphone measurements.

  The authors of the cited study specifically show that the infrasound energy measured by integration of the energy spectrum of the recorded microphone signal in the range 0.01 Hz to 15 Hz increases with the turbulent kinetic energy measured by a co-located sonic anemometer. The authors refer to the microphone signal amplitude as being a surrogate for turbulent kinetic energy but the relationship was non-linear, and the authors of the cited study related the turbulent kinetic energy measured by the sonic anemometer to the voltage content in the microphone signal, not the sound pressure level or pressure itself (which will vary with make and manufacture). Hence we described this relationship as a qualitative correspondence rather than a correlation or quantitative relationship.

- The objective of using an infrasound instrument for turbulence characterization is unclear here. Do the authors intend to simply use infrasound measurements for qualitative turbulence detection? Or quantify turbulence characteristics? A decisive discussion helps clarify the objectives of using the infrasound instrument.

  The motivation for the inclusion of the microphone in this study was indeed for qualitative turbulence detection, motivated by the potential usage of infrasonic microphones to detect clear air turbulence. The in-situ sensors on the UAS were intended to provide the quantification of the local turbulence characteristics allowing a comparison of the microphone measurements to the quantified turbulence. This is because the use of infrasonic microphones for clear air turbulence detection is still in its infancy, with little known about how this particular microphone might respond at altitude, or whether a signal indicating the presence of turbulence can even be detected. Given that the difference in remote vs in-situ sensing modalities are very different, we are still at the point of qualitatively connecting measured infrasonic energy to turbulent kinetic energy.

  We have re-written this section to try to be more clear.

- 220-224: "During flight, the autopilot maintained flight speeds sufficient to produce pressure differences well within the range of the low-sensitivity transducers (i.e. the dynamic pressure was maintained between 100 Pa and 200 Pa) which exceeded the range of the high sensitivity transducer connected to $\Delta P_1$. Hence, only the readings from the low-sensitivity sensors were used for data analysis. However, the high sensitivity transducers provided a means to estimate the uncertainty of the pressure measurement, as will be described later."

If I understand this correctly, the wording suggests that the high-sensitivity pressure transducer was used for uncertainty estimation only and not for scientific analysis. This raises questions about the increasing instrument noise floor with altitude. Have the authors modeled/empirically identified the five-hole probe instrument's noise characteristics as a function of altitude? Was the high-sensitivity transducer saturated frequently in flight and mostly the data unusable?

Specifically, the high-sensitivity $\Delta P_1$ transducer (which is sensitive to dynamic pressure) was saturated during the entire forward flight portion of the experiment, making recovery of any relative wind velocity vector impossible, The high-sensitivity $\Delta P_{32}$ and $\Delta P_{54}$ transducers, which are sensitive to $\alpha$ and $\beta$, were not saturated, since the $\alpha$ and $\beta$ angles are close to zero throughout the entire flight. As described in the uncertainty analysis section, this allowed us to compare the high- and low-sensitivity $\Delta P_{32}$ and $\Delta P_{54}$ transducer pressure readings during the entire flight and use the difference between them for uncertainty estimation. We have updated the paragraph referenced above to clarify.

As will be discussed later in this response, the noise was found to be independent of altitude.

– 243: "The time-dependent horizontal wind velocity magnitude and direction could then found from" modified to "The time-dependent horizontal wind velocity magnitude and direction could then be found from"

This has been corrected. Thank you.

– Figure A1. "Figures showing comparison of (a) horizontal wind velocity magnitude, (b) horizontal wind direction, and (c) vertical component of wind velocity calculated using all transducers to find Q, $\alpha$, and $\beta$ and using only two transducers to calculate Q and $\beta$ with $\alpha$ determined from the aircraft angle of attack measurement. Comparison of resulting (a) $\langle u'2\rangle$, (b) $\langle v'2\rangle$, and (c) $\langle w'2\rangle$ Reynolds stress tensor components." I believe that the figure caption has mislabelled tiles "(a) $\langle u'2\rangle$, (b) $\langle v'2\rangle$, and (c) $\langle w'2\rangle$". Shouldn't it be "(d) $\langle u'2\rangle$, (e) $\langle v'2\rangle$, and (cf) $\langle w'2\rangle$"?

This has been corrected. Thank you.

– 293 - 298: "To minimize the impact of electrical noise introduced into the pressure signals by the sensors and data acquisition system, during post-processing a background noise subtraction procedure was conducted on the digitized voltage signals prior to scaling them to Pascals. This process involved identifying a 5 minute long segment of the signal measured prior to balloon launch when a cover was present over the five-hole-probe (Fig. 1a) and the infrasonic signal was quiescent. This portion of the time series was assumed to be representative of the background electrical noise and therefore subtracted from the full time series in the Fourier domain in 5 minute long segments."

The authors assume, without adequate justification, that the instrument noise characteristics are independent of flight dynamics. This choice, without a discussion or proper justification, is speculative and questionable, and the representativeness of the instrument noise measured pre-flight in quiescent conditions warrants reevaluation.

Further, it is claimed that electrical noise is "minimized" without a presentation/discussion identifying/stating (with references to studies in literature if any) the implications of noise on the measurements or derived data products.

In this revision, we have added additional justification that the approach we have devised is successful in reducing the impact of high frequency periodic noise content that was present. Specifically, the revised manuscript now includes:

1. Excerpts from time series illustrating the noise content.

2. Frequency spectra showing that the frequency content of the transducers in an inactive environment chamber did not change between atmospheric and low pressure/low temperature conditions.

3. Frequency spectra for all altitudes, which shows that the high frequency content of the transducer signal does not change during the flight for frequencies higher than the signal from the five hole probe

4. A comparison of excerpts of the pressure time series taken during flight showing that the filter appears to successfully remove electrical noise from the pressure signal.

5. A comparison of frequency spectra before and after filtering showing that the high frequency content is reduced, and that the pressure signal more accurately reproduces a -5/3 slope in the square-root of the corresponding dynamic pressure signal (i.e. in the resulting velocity signal).

Note also that despite this evidence indicating that the electrical noise removal was successful, we found that its impact on the statistics presented in the remainder of the paper was negligible. However, the noise filtering approach was added in response to a prior reviewer who felt that not having such an approach would bias the velocity spectra in the inertial subrange.

– The discussion of pressure measurement errors due to noise is out-of-place in section 2.2.1. It is recommended that the authors discuss the instrument noise/characteristics in section 2.2.1 instead.

We had initially included this discussion in 2.2.4 as it cannot be determined if the noise source is the transducers, data acquisition system, or embedded computer and the embedded computer is the final link in this chain. However, given that the approach had its greatest impact on the five-hole-probe pressure measurement, we have moved this discussion to 2.2.1 as suggested.

– 359: "Comprise" modified to "compromise".

This has been corrected. Thank you.

– 382: "This latter constraint is introduced since $\Delta P_1$ exceeded the transducer's range shortly after the aircraft was released and started its flight towards the helical orbit location." Is the transducer under consideration here the high- or low-sensitivity transducer? Please clarify.

We have clarified by rewriting this statement to read 'This latter constraint is introduced since even the low-sensitivity $\Delta P_1$ exceeded the transducer's range shortly after the aircraft was released and started its flight towards the helical orbit location. By the time the aircraft reached the orbit location, $\Delta P_1$ had returned to a range measurable by the low-sensitivity transducer, although it never reduced to a value measurable by the high-sensitivity $\Delta P_1$ transducer.'

– 401: "In order to decrease the vertical spacing between statistical values, each segment is overlapped with its neighbor by 50%, thereby decreasing the spacing of statistical quantities to 75 m vertically)." It is unclear as to where the bracketed text begins. Please clarify.

The bracket has been removed.

– Figure 9: The horizontal wind data presented here exhibits periodic motions (on visual inspection) on 1 km vertical scales (Ex: 5 - 10 km in Flight 2). It is not uncommon to expect periodic artifacts in horizontal wind estimates derived from flights following periodic/helical tracks. It is recommended that the authors present a brief discussion to clarify the quality of the estimated horizontal winds with emphasis on the impact (if any) of periodic artifacts related to the flight trajectory on the horizontal wind components. Any evidence showing that the periodic artifacts (if any are present) are uncorrelated to flight dynamics will aid in resolving questions on the quality of horizontal wind estimates.

We are familiar with this issue with winds measured by orbiting UAVs which, from our experience, is typically introduced through error propagation and amplification in the conversion from body-fixed, to inertial coordinate systems, with error in the magnetometer reading the greatest culprit. This concern has also been raised by several other referees in early versions of this manuscript. To address this issue, prior to preparation of the initial submission and over the course of several revisions we have:

1. compared periodicity in the wind estimate to that of the helical flight path's pitch and observed that they do not exactly correspond to the orbital pitch, which means that if there is a bias it is altitude dependent However, the similarity is close enough that we do not want to highlight the difference in the manuscript as it is weak justification. Note also that the agreement between periodicity and orbital pitch is really only evident during Flight 1, as the aircraft passes through the edge of the jet stream.

2. noted that very similar periodicity is evident in the ABQ and TUS radiosonde measurements. The introduction of multiple radiosondes into the comparison also shows that the HiDRON measurement trends are within the variability in the radiosonde wind measurements. However, these radiosonde measurements are not coincident/co-located, nor are they high resolution measurements, indeed it appears that the wind measurements in the radiosonde

395 profiles were downsampled, which may artificially introduce the periodicity observed. Therefore we do not feel confident that this is sufficient evidence that this periodicity is natural and not measurement bias.

3. introduced the uncertainty analysis which shows that the observed periodicity in the winds far exceeds the expected uncertainty, but this analysis only accounts for known errors and cannot incorporate unknown biases (i.e. magnetometer drift). However, prior to preparation of this manuscript, we conducted an informal perturbation study to
400 determine if the periodicity in the wind estimate could be explained by measurement error but found that for them to be removed from the wind profile, the yaw error would not only have to be non-monotonically dependent on altitude (i.e. if it is due to sensor drift, it would have to be drifting significantly away from and then back toward the correct value), and the magnitude of the error would have to be so high that it would be noticeable by the ground operators. Given the informality of the perturbation analysis and that we cannot know beyond a reasonable doubt
405 that there wasn't drift/failure of the autopilot yaw measurement to this degree of error, we elected only to include the formal uncertainty analysis.

4. added in the wind estimated by the aircraft's drift measured by GPS on ascent. These are only roughly co-located measurements (see Figure 6), but there are certain parts of each of the profiles that quite strongly agree between the two wind estimates. Most notably at high altitudes during Flight 3, where the ascent/descent phases of the flight
410 were most co-located and coincident. Introduction of this comparison appears to have done the most to assuage the previous reviewers' concerns.

Most of these arguments have already been noted in the text, although relatively weak when taken individually, as a whole they provide us with some confidence in the wind estimates. The net result is that the discussion we have provided in the manuscript acknowledges that there may be bias introduced by the flight path. However, we intentionally keep the
415 discussion vague since we cannot be 100% certain that the wind estimate is/isn't affected by this bias.

We also hypothesize that it is possible that there there may be significant horizontal heterogeneity in the winds at these altitudes given the amount of mean kinetic energy, large-scale shear, and static stability present, which could lead to large-scale quasi-two-dimensional inactive instabilities at the edge of the jet stream. The glider will be much more sensitive to such horizontal heterogeniety than the balloon-based measurements, which will drift with the air parcel
420 and therefore not experience as much horizontal shear. However, at least one previous reviewer was resistant to these suggestions, particularly as we have no evidence that such horizontal heterogeneity is present. We therefore have limited this point to a suggestion of its possibility in the text.

In the revised manuscript we have added some additional exposition to the text in the discussion of uncertainty about the source of periodic artifacts and in the discussion of the wind estimates about the possibility that periodicity in the wind
425 profiles can be attributed to this source of uncertainty.

– 438: "In general, the wind magnitude and direction measured by the HiDRON H2 are within the bounds provided by the radiosonde soundings, with the wind direction measured during descent producing good agreement with that reported by the radiosondes and by the GPS on ascent." This statement is vague. It is not clear what the constitutes "good" agreement here especially when the nearest radiosonde sources for comparison are spatially separated on scales as large as 100s of
430 km.

We have added a discussion quantifying the difference in the body of the paper while discussing the wind profiles and summarized this in the last section.

– Section 3.3: Infrasonic Detection of Turbulence. The discussion presented in this section suggests that the authors use the infrasonic microphone for purely qualitative analyses. It is vaguely stated that the measured infrasonic amplitude
435 correlates to TKE and no further insights were offered to reinforce the effectiveness of utilizing the instrument for turbulence observations. It is recommended that the authors provide comments on the effectiveness of the infrasonic microphone data for any quantitative turbulence characterization.

As we do not have a comprehensive spatio-temporal map of the sound-generating turbulence during these flights, we cannot yet provide a quantitative empirical connection between the turbulence in the atmosphere and the infrasonic

440    signal measured by the microphone. Although there are ongoing efforts seeking to produce this relationship, we cannot include them in this manuscript. We therefore have to relegate our discussion and conclusions to a qualitative evaluation of the microphone response, and have tried to rework the relevant sections of the manuscript accordingly.

---

## Author Response (AR5)

**Response to Referee 3**

We once again appreciate the time taken by the referee to further review our manuscript and provide comments and suggestions for its improvement.

Below, we respond to the individual comments made by the referee on this version of the manuscript. As the new referee comments are integrated into the original review, the original review by referee is provided in black text, the original response by authors is provided in blue text, the latest review by Referee 3 is provided in green text, and the response from the authors to these latest comments is provided in red text. Note that below we have removed portions of the original response which Referee 3 has found acceptable

This paper has been improved by a tighter focus and omission of many previous vague and unsupported statements. Most of the arguments from the previous reviews have been settled by removal of the sections in question, or by clarification in this revision. Remaining issues are noted below.

Reviewer comments for this latest revision are colored in green. Items without additional green comments are acceptable in the revision.

This version provides more information on the filtering scheme for multi-hole probe pressure measurements, but the noise-subtraction approach remains highly suspect. Within this, however, is a basis for determining the noise floor of the measurements, which could then be used to select spectral data above the floor by a suitable factor for turbulence characterization. The estimation process for vertical wind remains unclear, particularly for flights 2 and 3 where one of the pressure sensors was disconnected. As this is not really needed for the results of the paper, perhaps this could be omitted if it cannot be clarified.

Comments:

– Line 259: "angle between the true airspeed determined from the aircraft's pitot probe, and the vertical velocity determined by the aircraft's variometer". No idea what this means. Pitot does not provide an angle, and variometer (pressure altitude rate, akin to inertial velocity) is not the same as the vertical component of relative wind.

When the autopilot manufacturer was queried as to how the angle of attack is determined, the description above is verbatim to the response we received. We have contacted them again and have updated the description of how $\alpha$ was found, specifically from a combination of the true airspeed determined from the aircraft's pitot probe, the vertical speed determined by a Kalman filter fusion of the static pressure rate of change, vertical acceleration, and global positioning system velocity, along with the aircraft orientation measured by the autopilot gyroscopes which was used to transform between inertial and body-fixed coordinates.

We have updated the text accordingly.

Text still incorrectly implies that vertical aircraft speed is the same as vertical wind, e.g. lines 325-326: "This orientation data allowed transformation of the vertical speed from inertial to body-fixed coordinates". How do you get angle of attack from vertical (body) velocity and pitch angle and pitot airspeed? Pitch angle is not the same as angle of attack, and flight path angle is not either, unless you assume vertical air motion is zero. The description of "a combination of factors" is still very vague, making it hard to gauge the integrity of the data. Later, lines 419-421, implies (incorrectly) that the vertical wind is found from the sine of the pitch angle multiplied by the airspeed. It would be much clearer if the equations used to make these estimates and error bounds were provided.

Although we believe we understand the calculation, we do not have access to the exact form of these equations being used (being contained within the manufacturer's software), therefore we cannot reproduce them within this manuscript. However, we confirmed that the calculation does indeed assume the vertical velocity of the air relative to the ground is negligible relative to the vertical velocity of the aircraft relative to the ground. We had thought that this assumption was introduced in the prior version of the manuscript but now realize this was not the case. Thank you for identifying this omission.

We have revised this manuscript to include this statement. Note that, with this assumption, $\alpha$ can calculated following the standard form used in flight dynamics.

- Lines 293-301: Sensor and signal amplification/conditioning and quantization noises are random processes, so cannot be measured before-hand and subtracted from signals measured later. The noise removal process described is as likely to corrupt the measurement as it is to clean it up. Even if "noise" means biases, these are likely to be variable with temperature, so could not conceivably be obtained from pre-flight sampling.

This would generally be true, except that much of the high frequency noise in the pressure transducer signal is quasi-periodic noise which we believe to be introduced by a switching voltage regulator, rather than more stochastic noise sources described above. In this revision, we have added additional exposition about the noise characteristics as well as evidence that the approach we have devised is successful in reducing the impact of the quasi-periodic noise, and evidence that the noise was independent of atmospheric conditions. Specifically:

1. Excerpts from time series illustrating the noise content.

2. Frequency spectra showing that the frequency content of the transducers in an inactive environment chamber did not change between atmospheric and low pressure/low temperature conditions.

3. Frequency spectra for all altitudes, which shows that the high frequency content of the transducer signal does not change during the flight for frequencies higher than the signal from the five hole probe

4. A comparison of excerpts of the pressure time series taken during flight showing that the filter appears to successfully remove electrical noise from the pressure signal.

5. A comparison of frequency spectra before and after filtering showing that the high frequency content is reduced, and that the pressure signal more accurately reproduces a -5/3 slope in the square-root of the corresponding dynamic pressure signal (i.e. in the resulting velocity signal).

Please also note that this noise filtering approach was added in response to a previous comment that stated 'Not clear how (of if) the noise floor/noise figure is removed in the qualified data fits. The noise floor can corrupt the estimated slope, so its removal can often retrieve epsilon values that were previously rejected. This could help to explain the absence of qualified values at the higher altitudes,' As the noise floor in the wind velocity was a function of numerous inputs, we could not devise a hands-free, unbiased, approach to consistently identify or remove it from the wind estimate. Therefore, we introduced this filtering approach to at least remove some of the noise introduced by the pressure transducers. We found that its impact on the statistics presented in the remainder of the paper was negligible.

The noise shown in Figure 4(a) does indeed appear to have a set of spikes, with a repetition frequency of about 1 Hz. The fact that there is no prominent, persistent spectral component near 1 Hz (Fig 4(b), but the noise spectrum is very broad, suggests that these spikes are not regular. Read: unpredictable in amplitude and phase, even on a short time scale. Hence subtraction of a pre-recorded signal, in the time domain or frequency domain, cannot be expected to remove noise in measured signals. To the extent that this subtraction reduces the spectral component amplitudes, it can be expected to reduce signal variation in the time domain, and appear "filtered", but both signal and noise are affected equally. At frequencies where the signal component amplitudes are large, the subtraction of a small noise changes the signal very little. But at frequencies where the component amplitudes are not large relative to the noise, then both signal and noise are altered greatly, sometimes reducing and sometimes increasing the component magnitudes, depending on the relative phase. Because of this random effect, it would be better to only accept spectral components for analysis that are significantly larger than the noise spectrum, e.g. by 1 order of magnitude. All others are corrupted by noise, and doubly corrupted by the filtering technique proffered here. A test for this corruption would be to compare time domain signals pointwise before and after filtering at a scale where the frequencies of interest (e.g. up to 20 Hz) can be seen. The spectra of Fig 4(f) show another way to assess this, but it would be helpful to show the noise spectrum, too.

We agree with this commentary, and were actually surprised that the process worked as well as it did. We also note that while vetting this procedure, we did indeed closely compare time domain signals before and after filtering at a scale where the frequency content on the order of 20 Hz is evident seen and did not observe evidence of corruption. However, we obviously could not provide such a detailed comparison in the revised manuscript as it would be impossible to do so over the length of the entire flight. Also, selectively presenting a 1 s portion of the signal showing no corruption provides little value as evidence that such corruption was not present elsewhere.

This is why we provided the spectra of Fig. 4(c) which shows *all* one-sided power spectra calculated from the flight data, and compares them to the spectrum which is used to represent background noise, as the referee is suggesting we do in Fig. 4(f).

The difference between Fig. 4(c) and Fig. 4(f) is that for Fig. 4(f) the pressure signal has been transformed to an approximated velocity signal prior to calculation of the spectrum, specifically to illustrate how the agreement with the -5/3 inertial subrange slope is slightly improved with inclusion of the filtering process. For this figure, a single spectrum has been isolated to maintain readabilty of the figure. However, given the request of the referee, we have added the corresponding background noise spectrum to the revised manuscript so that a reader can compare the magnitude of the noise to useful signal if so desired.

If the noise spectrum is well below the signal spectrum up to where the inertial cascade is fit, then this indicates that that spectrum should not be unduly influenced by noise, or the filtering method. But how representative is this one case? Otherwise, it is very difficult to believe that the resulting modified signals are reasonably free of artifacts, since the basic filtering approach does not make good sense.

Referring to Fig. 4(c) and, as should be evident throughout the manuscript, there are portions of the flight where there is little-to-no high frequency fluctuations in the air to measure (e.g. where there is no turbulence to measure). In these instances, there is no reason to expect the signal content to increase above the background noise spectrum (i.e. all the signal is in the DC portion of the signal, which is subtracted out when calculating the energy spectrum). In these cases, the time-domain signal can fully be expected to be corrupted by the noise subtraction process, but since there was no information contained within the high frequency content of the signal (only noise), this corruption would have little impact on the measurement results. Where there *is* useful information in the signal then that information is typically over an order of magnitude above the background noise spectrum.

Another approach would be to remove the spikes (if they are the main problem) by an outlier removal method in the time domain.

This would be a challenging approach to implement in practice. As illustrated in Fig. 4d, the noise is smaller than the signal fluctuations and most outlier detection schemes use the standard deviation of the signal to detect outliers. Hence, this process would require a significant amount of effort to implement correctly.

Lastly, the hypothesis that this noise is due to the (switching) power supply seems unlikely, given that these switch at hundreds of kilohertz, and, due to the filtering inherent in the technique, produce supply line noise primarily at the fundamental switching frequency. So, even in the presence of aliasing, this would not be expected to produce such spiky noise. Are there high-current loads in the avionics that have narrow pulses at about 1 Hz, e.g. telemetry transmissions?

Our thought was that the high-frequency switching noise might somehow be aliasing into the sampled lower frequencies. However, we are not confident in this assumption, as it would require the noise to bypass the anti-aliasing filters, which is why in the manuscript itself, we only refer to the noise being present in the power supply line since we measure the noise in this signal. We cannot attribute the noise source to the avionics, since it appears even when the system is outside the aircraft (e.g. as evident by comparing the noise spectrum from the environment chamber Fig. 4b to that measured during flight Fig. 4c).

We reiterate that the only reason we implement this background noise subtraction approach is due to the referee's original comment that background subtraction should be applied to the energy spectrum prior to fitting the -5/3 inertial subrange slope used for the dissipation rate calculation. However, such background subtraction is only truly possible in the pressure transducer spectra (e.g. Fig. 4c) since that is the only point in the process where the noise is easily identified. Once the pressure signals from the different transducers are convoluted with the directional calibration and aircraft kinematics, it is much harder to confidently discriminate between useful signal and noise.

Regardless its source, the actual influence of the background noise on the measurement results is quite small. Prior to its implementation we compared statistics with and without the filtering applied and found that it has very little impact on the derived statistics, with the most noticeable impact being reduced scatter in some derived quantities, such as the turbulent kinetic energy dissipation rate and Reynolds stresses. This can be expected, as such statistics (e.g. mean, variance, power

spectra) will average out the phase distortion that will be introduced when the signal and noise have similar amplitude, while still reflecting the subtraction of the magnitude of the noise contribution to the signal.

However, we agree that this process is not ideal, and there will likely be phase distortions introduced into the time-dependent signal which may not be reflected in the derived statistics. We therefore have added additional text to the manuscript which acknowledges the likelihood of phase distortion being introduced through this process. If this is deemed an insufficient response to the referee's concerns, we can easily revert back to the statistics calculated from the unfiltered signal without any impact on the results and discussion within the body of the paper. However, in such a case we will not be able to address the referee's initial comment regarding background noise subtraction.

– Lines 340-341: "uncertainty in wind magnitude was found to be most dependent on the yaw angle" How? Why? Do the wind excursions in Figure 4 correlate with yaw angle?

The uncertainty dependence of wind magnitude on aircraft attitude is introduced into the wind estimate during the coordinate transformation between body-fixed and inertial coordinate systems. During flight, the yaw angle can vary from 0 to 360 degrees, whereas the sideslip, angle of attack, pitch and roll angles are near zero. The result is that the greatest contribution to the $u/v$ wind components during transformation is the true airspeed being multiplied by the sine/cosine of yaw and cosine of the pitch. The result is that during an orbit, the horizontal velocity components will have high sensitivity to yaw error at both $0°$ degrees and $90°$. For a similar reason, the vertical component of velocity is most sensitive to error in pitch.

This argument oversimplifies the wind estimation problem. The vector conversion of relative wind from body to inertial coordinates is equally sensitive to yaw errors at all yaw angles. It is the vector combination of relative wind and inertial velocity in the wind triangle that produces sensitivity variation on the orbit, since when the plane has the highest inertial velocity (flying downwind), its rate of change of attitude is greatest on the circle, and this can expose errors due to timing mismatches in the various sensors. The paper uses a complicated timing recovery scheme, since the data is not time stamped with a common reference, so this may be where to look for periodic wind estimate excursions. Do they occur around this point on the circle? Or do they occur where the orbit nears the jet? Or are they at 0 and 90 deg? (I can't tell from the paper).

The statement referenced above directly addresses the question as to why uncertainty in yaw can specifically lead to periodicity in the wind estimate and why it correlates to the yaw angle of the aircraft. It is also consistent with (van den Kroonenberg, A., Martin, T., Buschmann, M., Bange, J., and Vörsmann, P.: Measuring the Wind Vector Using the Autonomous Mini Aerial Vehicle M2AV, J. Atmos. Oceanic Technol., 25, 1969–1982, 2008) as noted in the manuscript. As noted in the statement, the sensitivity the referee is describing above is manifested as long-wavelengths in the wind estimate through the coordinate transformation, most notably the contribution from yaw, as we note in the text.

As to the influence of systematic errors, we are aware of these issues, and in the current study these were addressed by applying the referenced correction procedure of (Al-Ghussain, L. and Bailey, S. C. C.: An approach to minimize aircraft motion bias in multi-hole probe wind measurements made by small unmanned aerial systems, Atmospheric Measurement Techniques, 14, 173–184, https://doi.org/10.5194/amt-14-173-2021, 2021.) this optimization procedure is specifically designed to minimize the type of errors being referenced by identifying and corrects systematic bias in yaw, pitch, roll, as well as any residual timing mismatches and undetected airframe distortion. Use of this approach is already referenced in the manuscript. As for specific timing mismatches, recall that we have redundant measurement of the relative airspeed through the pitot probe (logged by the autopilot) and the five hole probe (logged by the on-board data acquisition system). These redundant measurements allow for precise intercomparison of the time lags between the two systems, which was examined carefully during our initial vetting of the results prior to preparation of the original submission. The importance of timing mismatches was also brought up in review several revisions ago and, as noted in the response to that comment, we detected a very slight mismatch in the system clock rates and re-processed the data accordingly. Note that correcting this mismatch did not have a noticable impact on the measured wind profiles, indicating that it was not a significant source of error.

Of course, it is impossible to remove all systematic bias error, and therefore we also note that our error estimation includes the effects of bias error in Equation 11 (and as discussed in the corresponding discussion) through the term $E_B$. This term is intended to characterize the impact of unaccounted for bias error.

As illustrated by the uncertainty bounds shown in Figure 4, the wind excursions cannot be explained by error in yaw. Note also that prior to preparation of this manuscript, we conducted an informal perturbation study to determine if these excursions could be explained by measurement error and found that for them to be removed from the wind profile, the yaw error would not only have to be non-monotonically dependent on altitude (i.e. not attributable to sensor drift), but it would have to be so high that it could only be explained by failure of the system.

I think you mean Figure 5.

Yes, you are correct, the response was written before adding the figure addressing the electrical noise filter. It's inclusion shifted the figure numbers resulting in the aforementioned typo.

Your uncertainty analysis only models the random forms of error, not the systematic ones such as that mentioned in the response above. So it is premature to say that these are covered by the doubling mentioned in the text.

As mentioned above, the reviewer is incorrect and the uncertainty analysis does indeed include the systematic (i.e. bias) error. Of course, the magnitude of the undetected bias errors are truly unknowable, however we believe that we have exercised all due diligence to eliminate the known biases, and estimate the uncertainty introduced by the unknown biases on the wind estimates.

The real test would be to plot the winds as a function of azimuth on the circle, as well as noting the mean wind direction on each circle and the location of the jet, to see if aircraft motions are correlated with the measurements.

Indeed, and during our vetting of the data prior to preparing the original submission (and at many points between then and now) we have indeed examined the results in such a manner to ensure that we were not misrepresenting the results. We include one such figure below as figure 1, which illustrates that the wind excursions/long wavelengths which the referee is referencing are not dependent on the yaw angle and vary in spatial location with altitude. We have not included this figure in the revised manuscript as it implies that these wind excursions are due to spatial heterogeneity. Discussion of any observations of spatial heterogeneity in previous versions of this manuscript was not well received by another reviewer. Given that our only evidence that the spatial heterogeneity was present are the results from the aircraft measurements we cannot provide any evidence that such spatial heterogeneity was present during our measurements.

We do note that the Flight 1 results in figure 1 does show lower winds at $\pm 90°$, which may be a signuature of systematic bias having an influence on the estimate due to the high winds present during that flight. We acknowledge in the manuscript that the wind excursions observed in the profiles, particularly for this flight, could be due to these types of errors.

We have added the first statement to the manuscript, the second statement is left out due to the informal nature of the perturbation analysis that was conducted (the uncertainty analysis presented in the manuscript being the more rigorous approach).

More rigorous, yes, given the assumptions, but perhaps leaving out the largest effects!

As noted above, we believe that we have exercised all due diligence to identify and eliminate the effect of biases, and to estimate the uncertainty of the unknown biases, on the wind estimates. We also acknowledge in the text that the wind excursions observed in the profiles could be due to these types of errors. Given this, it is not clear what additional revisions we can make to the manuscript to address the reviewers concerns.

However, should the editor feel that a version of figure 1 is important enough to include in the manuscript, we are happy to include it, but have not added it to the revised manuscript for the reasons noted above.

– Lines 400-401: "and we assume that the characteristics of the atmosphere within these segments are horizontally homogeneous (i.e. they are a function of z only)". A very loaded assumption to make with no justification!

[Figure]

**Figure 1.** Isocontours of wind magnitude for each flight shown as a function of azimuthal angle of aircraft position relative to the center of the aircraft's flight path. Also shown is the triangular mesh used for determination of the contours. The isocontours illustrate that the majority of the long wavelength periodicity shown in the profiles is not correlated to aircraft yaw angle.

This statement was not thought to be as loaded as as the reviewer is implying, due existing consensus that the intrinsic stability in the stratosphere and upper troposphere will promote horizontal homogeneity (as exemplified by one of the other reviewers being insistent that variability in the measured profiles must be due to measurement error or vertical stratification rather than any horizontal heterogeneity). As the aircraft approaches the surface and the radius of the orbits becomes smaller, this assumption also becomes less restrictive.

We have added our rationale for this assumption in the revised manuscript.

"Promote horizontal homogeneity" does not mean that the layer structures are perfectly horizontal and homogeneous over arbitrary distances. Does this extend over hundreds of m, km, tens of km? Gravity wave activity, for one, can upset this ideal situation, as intimated later from the data in lines 537-539, possibly contradicting this assumption. Also, nearer the surface, does the orbit radius decrease faster than the scale of horizontal variations? Basically, there is still no quantitative support for this assumption, and it should not be treated as common or obvious.

We agree with the referee that there are many potential sources of horizontal heterogeneity, and as evident in Figure 1 above, this assumption is not strictly supported by our measurements. It's inclusion was intended to satisfy concerns of another referee regarding the Richardson number estimation.

Upon reflection, we now realize that the assumption referenced above is neither justified, nor necessary, and therefore the simplest way to address the referee's concerns is to remove this statement from the revised manuscript.

– Line 444: "These low frequency waves may be bias in the wind estimate introduced by the orbital path". Indeed, it seems like they may correspond to one turn on the helix, so may be an anomaly in the wind retrieval. Comparison to the vertical period of the helix would be important here.

We have added some quantification and discussion of these values to the text.

The only "quantification" I see is the statement "that the periodicity is shorter than the pitch of the helical flight path". How much shorter? Does this hold for all orbits?

As the period/wavelength of these waves is altitude and flight dependent, their quantification can only be provided in an altitude-dependent analysis (e.g. as done with the wavelet analysis presented later). the statement we introduced at this point in the manuscript was intended to specifically address the referee's concern that the periodicity was correlated to the pitch of the helix.

In the revised manuscript, we have added a statement at this point in the manuscript referencing the wavelet analysis that appears later. The wavelet analysis allows the reader to see the energy content of the wind as a function of altitude and wavelength.

– Line 478: "which will reach a minimum at the frequency where the noise has a greater contribution to the integration than the signal". Why?

Compensated, or "pre-multiplied," spectra are commonly employed in turbulent boundary layer studies because they facilitate the visualization of energy spectra on semi-logarithmic axes. Specifically because they allow for a clearer representation of the frequency/wavenumber dependence of the relative contribution of each frequency/wavenumber to the overall energy content. This is because, for example,

$$d\langle u^2 \rangle = F_{uu}df = fF_{uu}d(\log f). \tag{1}$$

Hence when $fFd(\log f)$ begins to increase on a semi-logarithmic plot at high frequencies, this indicates a frequency range where the energy content increases with $f$. Given that universal equilibrium range turbulence will decrease in energy content with $f$, the minimum in $fFd(\log f)$ indicates a frequency at which the noise begins to have a greater contribution to the variance than the turbulence content.

However, the use of the compensated spectrum is a procedural detail and in hindsight distracts from the overall point of this processing step, which is to determine at frequency the contribution to overall variance increases with increasing $f$, rather than decreases. The same result could have been achieved using the un-compensated frequency spectrum, and we only used the compensated spectrum to simplify visualizing the frequency-dependence of $\langle u^2 \rangle$, $\langle v^2 \rangle$ and $\langle w^2 \rangle$.

As including the detailed exposition above would distract from this point, while adding little value to the overall intent of its inclusion, we have simplified the discussion to simply state that the upper bound of this range was determined by identifying the frequency where the noise has a greater contribution to the integration than the signal than the velocity fluctuations.

I thought the explanation above was fine. But why make the detail vague by leaving it out?

As we mention in the original response, including the detailed exposition above would distract from this point, while adding little value to the overall intent of its inclusion. There are approaches which could have been used that achieve the same outcome and hence the exposition provided above is unnecessary for repeatability or understanding of the paper. We thought this addressed the referee's 'why?' question above.

As we recognize the importance of precision in scientific writing, and that the referee has identified this as a point where such precision is necessary, we have added the above description to the revised manuscript.

– Line 478-479: "filter frequency was consistent with the probe's frequency response in the boundary layer and varied between 1 Hz and 20 Hz above the boundary layer". Does "filter frequency" correspond with the "frequency range" in line 473? By "probe's frequency response" hear do you mean "probe bandwidth"? How did this vary above the boundary layer?

By filter frequency, we meant the upper bound of the range described in the previous statement. We have modified 'filter frequency' to read upper bound of the frequency range.

Yes we do mean probe's bandwidth. These terms are interchangeable and, as we use frequency response at numerous points throughout the paper prior to using it in this sentence, would prefer not to change it to 'probe bandwidth'. We have clarified that we are referring to the maximum frequency response, as frequency response in general can also refer to the Bode plot of the probe's response to excitation.

A frequency response (amplitude curve as a function of frequency) does not have a "maximum frequency of response". This is a smoothly varying function with no lower bound, so some standard point on this curve is picked to indicate "bandwidth" of response. This is typically the -3dB roll-off frequency. This bandwidth (particularly if you say 3dB bandwidth) would be widely understood, whereas your "maximum frequency response" could be confused with other things, such as the frequency where the response is maximum.

We had incorrectly made the assumption that our use of the phrase 'maximum frequency response' implied that we were referring to the -3dB roll-off frequency. We have added a clarification in the revised manuscript at the location where we first use this term to indicate what it is referring to.

As noted in the text, the frequency at which the noise exceeded the signal varied between 1 Hz and 20 Hz. We do not describe a trend above the boundary layer since no trends were evident, being dependent on the presence of low-frequency energy content. We have altered the text to now read 'with the higher upper frequency bounds corresponding to instances where there was increased low frequency content in $F_{uu}$, $F_{vv}$ and $F_{ww}$.'

The frequency where the noise exceeds the signal is entirely dependent on the turbulent energy in the flow, so no trend (say with altitude) would be expected.

The phrase being referenced above was specifically introduced to answer the referee's original question 'How did this vary above the boundary layer?' However, on re-reading the original comment, we think that it's possible that the referee was referring to how the maximum frequency response varied above the boundary layer, instead of how the filter frequency varied above the boundary layer? If the former then, as already described in the manuscript, we only conducted frequency response measurement tests at ambient conditions and in an environment chamber at conditions close to those experienced in the stratosphere. Given that the maximum frequency response change between these two conditions was very similar (20 Hz and 10 Hz respectively) we can only infer that the maximum frequency response variation was small enough that it can be assumed constant above the boundary layer.

– Line 569: Do you mean $[\sigma]_{LF}$ instead of $[\sigma_f]$? Also, I don't understand why these two would increase at the same rate if high frequency energy is primarily increasing, as supposed as being "most likely".

Yes, we meant $[\sigma]_{LF}$ and have fixed this typo.

Rate is probably the wrong word for what we are trying to describe. We have reworded this statement to try to more clearly describe our rationale for why the ratio $[\sigma^2]_{LF}/[\sigma^2]$ will remain constant in the boundary layer.

Lines 713-715: garbled revised sentence.

We have revised the sentence spanning Lines 713-715.

**Response to Referee 4**

We appreciate the time taken by the referee to review our manuscript and provide comments and suggestions for its improvement.

Below, we respond to the comment made by the referee on this version of the manuscript.

325

I suggest that the authors amend the title of the article, for better clarity, to "High-altitude atmospheric turbulence measurements and qualitative infrasound observations using a balloon-launched small uncrewed aircraft system"

We have updated the title following this suggestion. However, we note that while the comparison between infrasonic energy and turbulent kinetic energy was qualitative, the infrasound measurements themselves were not qualitative. We therefore have

330 modified the suggestion to 'High-altitude balloon-launched uncrewed aircraft system measurements of atmospheric turbulence and qualitative comparison with infrasound microphone response' which we believe better captures the intent of the suggested title change.